# Extracellular matrix remodeling through endocytosis and resurfacing of Tenascin-R

Tal M. Dankovich [1,2✉], Rahul Kaushik[3,4], Linda H. M. Olsthoorn[2,5], Gabriel Cassinelli Petersen [1], Philipp Emanuel Giro[1], Verena Kluever [1], Paola Agüi-Gonzalez [1], Katharina Grewe[1], Guobin Bao[1,6], Sabine Beuermann[7], Hannah Abdul Hadi[1], Jose Doeren [1], Simon Klöppner[1], Benjamin H. Cooper [7], Alexander Dityatev [3,4,8] & Silvio O. Rizzoli [1,9✉]

The brain extracellular matrix (ECM) consists of extremely long-lived proteins that assemble around neurons and synapses, to stabilize them. The ECM is thought to change only rarely, in relation to neuronal plasticity, through ECM proteolysis and renewed protein synthesis. We report here an alternative ECM remodeling mechanism, based on the recycling of ECM molecules. Using multiple ECM labeling and imaging assays, from super-resolution optical imaging to nanoscale secondary ion mass spectrometry, both in culture and in brain slices, we find that a key ECM protein, Tenascin-R, is frequently endocytosed, and later resurfaces, preferentially near synapses. The TNR molecules complete this cycle within ~3 days, in an activity-dependent fashion. Interfering with the recycling process perturbs severely neuronal function, strongly reducing synaptic vesicle exo- and endocytosis. We conclude that the neuronal ECM can be remodeled frequently through mechanisms that involve endocytosis and recycling of ECM proteins.

[1] University Medical Center Göttingen, Institute for Neuro- and Sensory Physiology, Excellence Cluster Multiscale Bioimaging, Göttingen, Germany. [2] International Max Planck Research School for Neuroscience, Göttingen, Germany. [3] Molecular Neuroplasticity, German Center for Neurodegenerative Diseases (DZNE), Magdeburg, Germany. [4] Center for Behavioral Brain Sciences (CBBS), Magdeburg, Germany. [5] Max Planck Institute for Biophysical Chemistry, Göttingen, Germany. [6] University Medical Center Göttingen, Institute of Pharmacology and Toxicology, Göttingen, Germany. [7] Max Planck Institute for Experimental Medicine, Göttingen, Germany. [8] Medical Faculty, Otto von Guericke University, Magdeburg, Germany. [9] Biostructural Imaging of Neurodegeneration (BIN) Center, Göttingen, Germany. ✉email: tal.dankovich@med.uni-goettingen.de; srizzol@gwdg.de

The extracellular matrix (ECM) molecules of the brain form lattices that enwrap neurons and fill perisynaptic spaces[1]. These structures appear to be particularly durable, owing to the exceptional longevity of the ECM molecules[2,3], and they are deemed to stabilize neurons and synapses[4,5]. Nevertheless, the ECM must remain alterable to allow for the structural changes that occur in response to activity and plasticity. Such changes have long been thought to be infrequent in the adult brain, but this notion has been recently challenged by a series of super-resolution imaging studies. Synapses were found to change shape and location continually, on a time scale of minutes, both in acute brain slices[6] and in the adult brain[7,8]. These findings therefore suggest that the ECM may be remodeled relatively frequently.

This notion is difficult to accommodate with the best-known mechanisms for ECM remodeling, which involve ECM cleavage by proteolytic enzymes such as matrix metalloproteinases, followed by the secretion and incorporation of newly-synthesized ECM molecules[9,10]. An alternative solution to ECM remodeling seems therefore necessary, through mechanisms that reuse existing molecules, rather than relying on de novo synthesis. To search for such a mechanism, we focused on Tenascin-R (TNR), a matrix glycoprotein that is predominantly expressed in the central nervous system[11]. TNR is highly enriched in perineuronal nets (PNNs), a condensed ECM lattice surrounding a subset of inhibitory interneurons, and is essential for PNN formation. TNR is also expressed in the more diffuse perisynaptic ECM associated in the neuropil with both inhibitory and excitatory synapses on a broad range of neuronal cell types[12].

We targeted TNR by several advanced imaging assays, and we found that a subset of the TNR molecules, found especially in the vicinity of synapses, cycled between the ECM and neuronal organelles. We termed these TNR molecules the "recycling pool," in analogy to the recycling pool of synaptic vesicles[13,14], which also cycle between the cell surface (exocytosed) and internal (endocytosed) states. The recycling TNR molecules were secreted mainly at synapses, in a process that was dependent not only on the overall network activity levels, but also on the activity levels of the particular synapses. These molecules were then endocytosed by the neurons and were recycled back to the perisynaptic ECM, over ~3 days. Most molecules appeared to undergo several cycles without being degraded. Finally, perturbing the recycling TNR pool disrupted synaptic function, which suggests that these molecules are intimately linked to synaptic activity. We conclude that neurons maintain a pool of TNR that continually recycles in and out of perisynaptic ECM, allowing for frequent ECM remodeling without the need to synthesize new molecules, and thereby explaining how synaptic fluctuations could be dealt with without compromising the exceptionally long lifetime of ECM molecules.

## Results

**A classical biotinylation assay suggests that TNR molecules are recycled**. To test whether ECM molecules can indeed be reused, we first used a classical biotinylation-based assay that has been instrumental in determining the reuse (recycling) of neuronal surface proteins, and especially of neurotransmitter receptors[15]. We employed the same system used previously for determining the recycling of neurotransmitter receptors, rat cultured hippocampal neurons.

TNR is well detectable in these cultures, as observed by immunostaining, and is also found in PNNs (Supplementary Fig. 1a). Moreover, it is present at both excitatory and inhibitory synapses, as indicated in the Introduction (Supplementary Fig. 1b). Therefore, a biotinylation assay should be able to detect the dynamics of TNR in culture.

We treated the neurons with a cleavable, membrane-impermeable biotin derivative, to tag all proteins at the cell surface. We allowed the neurons to internalize molecules for 6 h, and we then stripped biotin from the cell surface, thus leaving this label only on the endocytosed proteins (Fig. 1a). To measure the potential recycling of the internalized molecules, we incubated the neurons a further 18 h, to allow for protein resurfacing, and then performed a second round of biotin stripping (Fig. 1a). We collected the biotin-tagged proteins on streptavidin beads, and we then tested the recycling of TNR and other proteins by immunolabeling the beads and imaging them in confocal microscopy (Fig. 1b). This is a particularly sensitive technique, enabling the detection of proteins in minute sample volumes[16]. We found that TNR is indeed endocytosed during the initial 6 h of incubation, and that a significant proportion of the molecules resurfaces during the subsequent 18 h. This behavior was similar to that of a positive control, the synaptic vesicle protein synaptotagmin 1, Syt1, which participates in the well-known recycling of synaptic vesicles (Fig. 1c). In addition, we tested a membrane protein that is endocytosed, but is not expected to recycle (the lysosomal protein LAMP1), and a membrane-resident protein, which is very rarely endocytosed (myelin basic protein, MBP). Both behaved in the expected fashion (Fig. 1d, e), suggesting that this assay reports accurately the protein behavior.

In principle, the recycling TNR molecules could have as a source neurons or glia cells. To verify this, we first characterized our cell cultures (Supplementary Fig. 2), noting that astrocytes make up ~60% of all cells, with neurons making up ~38%, and the MBP-containing oligodendrocytes ~2%. PNN-containing neurons made up 11% of all neurons, in agreement to the literature[17]. In spite of their large number, astrocytes internalized very little TNR (around 5%, Supplementary Fig. 3), implying that most of the TNR recycling observed in the biotinylation assay must be in neurons. The original source of TNR (the original location of TNR synthesis), however, cannot be determined by these experiments, and could potentially also be found in astrocytes, at least to some level, since astrocyte cultures can produce the molecule[18].

Finally, the loss of biotinylated TNR cannot be ascribed to its degradation, since this molecule is extremely stable (see Supplementary Fig. 4, including a characterization of TNR degradation via imaging and Western blotting), and since we blocked lysosomal degradation during these experiments.

**An imaging assay suggests that TNR molecules emerge preferentially near synapses**. To verify these findings by an imaging assay, we turned to a knock-out validated TNR antibody[19] (Supplementary Fig. 5a, b). Application of the antibody to neuronal cultures did not appear to cause any changes in their behavior, as was verified by electrophysiological measurements (Supplementary Fig. 5c, d). We applied fluorophore-conjugated TNR antibodies to the cultures, and imaged them at 37 °C, for several hours. We observed the accumulation of TNR antibodies in the neuronal somas, indicative of endocytosis (Supplementary Fig. 6). However, many other cellular regions remained virtually unchanged, which suggests that not all TNR molecules are dynamic.

To focus on the dynamic TNR molecules, we relied on an assay used extensively for synaptic vesicle proteins, a 'blocking-labeling' assay (see[20] and references therein). The surface TNR epitopes were blocked using unconjugated antibodies. Fluorophore-conjugated TNR antibodies were then applied at different intervals, to label newly-emerged epitopes (Fig. 2a). This assay enabled us to detect a slow but steady appearance of such epitopes (Fig. 2b, c). A super-resolution investigation using STED microscopy revealed that these

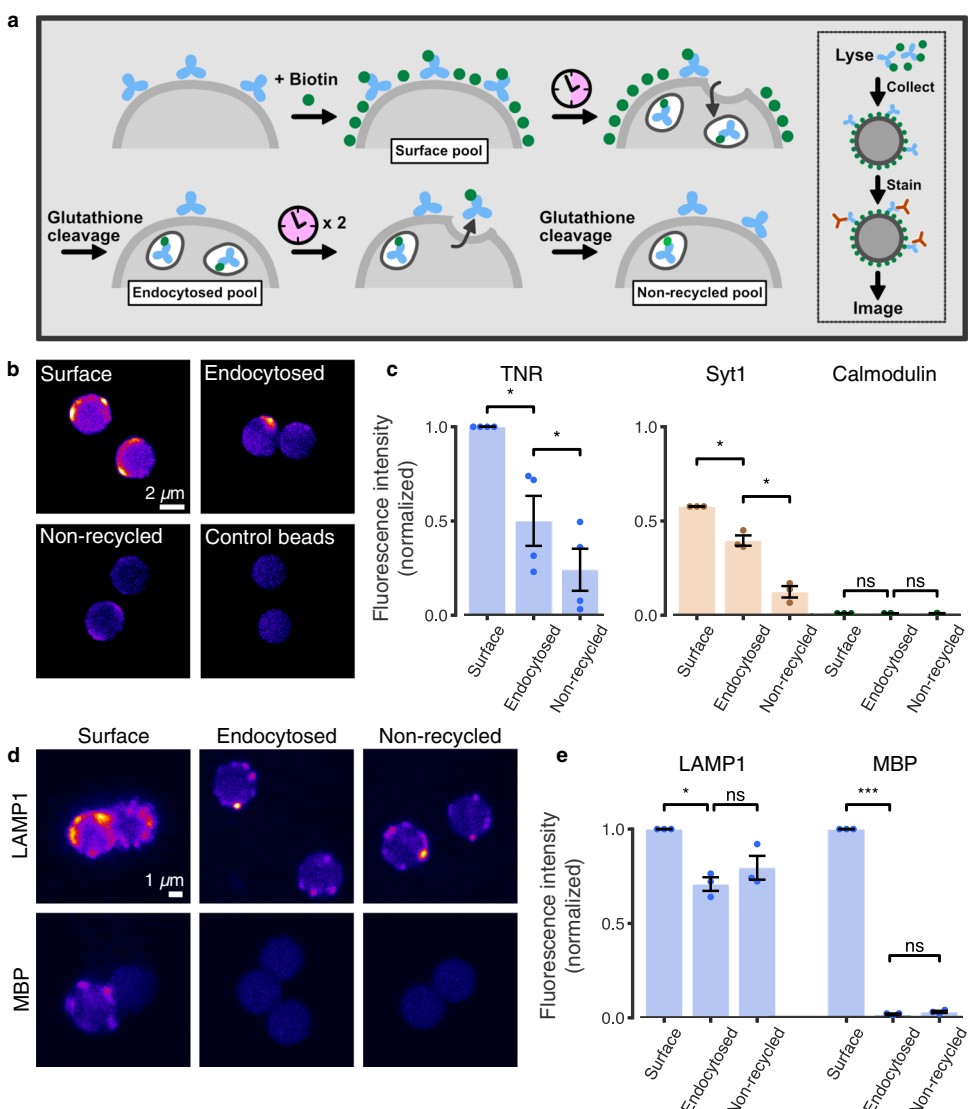

**Fig. 1 TNR molecules endocytose and subsequently resurface in neurons. a** Neurons were treated with sulfo-NHS-S-S-biotin to label cell-surface proteins. Following a 6-h incubation, allowing for internalization, remaining cell-surface proteins were stripped of their labels with glutathione. After a further 18 h of incubation, allowing for recycling, the neurons were again treated with glutathione. Lysates representing surface, endocytosed or non-recycled pools, were collected on streptavidin-coupled beads, immunostained for TNR, immobilized on glass slides and imaged with confocal microscopy. **b** Example beads collecting TNR pools, or controls incubated without primary antibodies. Scale bar = 2 μm. **c** A quantification of TNR fluorescence intensity normalized to the 'surface' mean in the corresponding experiment, indicates that a large fraction of TNR molecules endocytose within 6 h, and many subsequently resurface within 24 h. As positive/negative controls, the synaptic vesicle protein Syt1, well-known to recycle, and the intracellular protein calmodulin were tested. The plots are scaled by the ratio between the 'surface' mean for these proteins and that of TNR. $N = 4$ (TNR) and 3 (Syt1/calmodulin) independent experiments with >100 (TNR) and >50 (Syt/calmodulin) beads. **d**, **e** As additional controls, the lysosomal marker LAMP1, known to endocytose but scarcely recycle, and myelin basic protein (MBP), which should not endocytose, were also tested. Scale bar = 1 μm. **e** Quantification of LAMP1/MBP fluorescence intensity, normalized to the 'surface' mean of the corresponding experiment. $N = 3$ independent experiments with > 50 beads. Statistical significance was evaluated using repeated-measures one-way ANOVA, (**c** TNR: $F_{1.153,3.458} = 28.29$, **$p = 0.009$; Syt1: $F_{1.007,2.014} = 62.98$, *$p = 0.015$; calmodulin: $F_{1,2} = 0.016$, $p = 0.912$; **e** LAMP1: $F_{1.293,2.585} = 19.6$, *$p = 0.028$; MBP: $F_{1.52,3.041} = 28337$, ***$p < 0.001$), followed by the Holm-Sidak multiple comparisons test comparing 'surface'/'endocytosed' and 'endocytosed'/'non-recycled' (**c** TNR: *$p = 0.032$, *$p = 0.021$; Syt1: *$p = 0.044$, *$p = 0.044$; calmodulin: $p = 0.933$, $p = 0.993$; **e** LAMP1: *$p = 0.045$, $p = 0.162$; MBP: ***$p < 0.001$, $p = 0.068$). Data represent the mean ± SEM, dots indicate individual experiments. Source data are provided in Source Data file.

TNR molecules were enriched near synapses (Fig. 2d), where they also emerged more rapidly than for the cell as a whole (compare Fig. 2c, e). Importantly, their appearance was potentiated by enhancing neuronal activity using the GABA$_A$ blocker bicuculline, and was inhibited by reducing neuronal activity using the glutamate receptor blockers CNQX/AP5 (Fig. 2f).

We analyzed the distribution of the TNR signals along dendrites or axons (labeled using the lipophilic tracer DiO),

moving from synapses (identified by immunostaining the synaptic vesicle marker VGlut1) along the neurites. A substantial enrichment can be found at synapses for newly-emerged epitopes, which is much higher than that observed for all TNR epitopes (Fig. 2g). This indicates that newly-emerged TNR epitopes are preferentially found in the vicinity of synapses.

To test whether this preference is accidental, or whether it depends on the local synaptic characteristics, we compared the

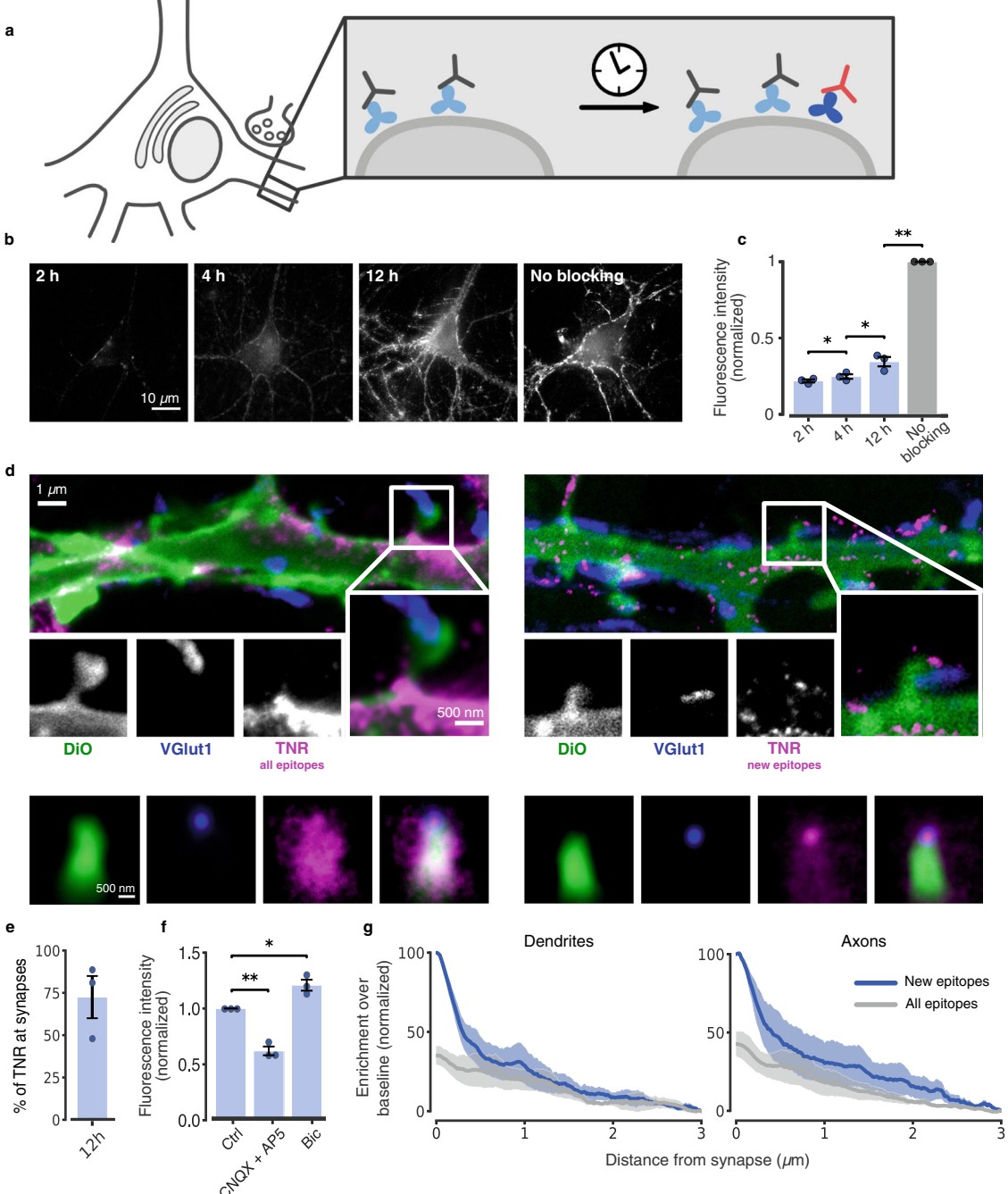

TNR labeling to synaptic strength. To estimate this on the presynaptic side, we labeled the active vesicle pools using an antibody directed to the intra-vesicular domain of synaptotagmin 1 (Syt1), which has been employed for this type of assay for more than two decades[20–23]. This antibody enters actively-recycling vesicles during exo- and endocytosis, and marks them fluorescently, thereby providing a measure of the amount of vesicle activity at the particular synapse. We found a remarkably strong correlation between the amount of TNR epitopes present at a particular synapse and its estimated activity (its estimated synaptic strength; Fig. 3a). To perform a similar measurement for the postsynaptic side, we took the spine head size as a proxy for synaptic strength[24]. Again, the amounts of newly-emerged epitopes were found in higher amounts at synapses with larger spines (Fig. 3b). This suggests that the emergence of these TNR molecules on the surface correlates to synaptic weight.

To further confirm this issue, we turned to an analysis of the TNR levels in the neurites (axonal or dendritic) of individual DiO-labeled neurons. The literature suggests that the total TNR levels in individual neurons should correlate among all neurites, with specific neurons exhibiting high or low TNR levels[11,25]. We indeed found this to be the case (Supplementary Fig. 7a). However, this should *not* be the case for newly-emerged TNR epitopes, if their emergence is dependent on the local synapse strength. This hypothesis was confirmed by our measurements (Supplementary Fig. 7b), as the dendrites and axonal branches of individual neurons had widely variable fluorescence levels for the newly-emerged epitopes.

Overall, these experiments suggest that this assay is able to detect TNR molecules appearing on the surface, from an intracellular TNR population. However, new TNR epitopes could also emerge through the un-binding of unconjugated antibodies

**Fig. 2 Dynamic TNR molecules emerge at synapses, in an activity-dependent fashion. a** To monitor dynamic TNR molecules, surface epitopes were blocked by incubating with non-fluorescent TNR antibodies (gray). After some time, fluorophore-conjugated antibodies were applied (red) to reveal newly-emerged epitopes (dark blue). **b** Newly-emerged epitopes 2/4/12 h post-blocking, (epifluorescence). Scale bar = 10 μm. **c** Fluorescence intensity, normalized to non-blocked neurons. $N = 3$ independent experiments, ≥10 neurons per datapoint. Repeated-measures one-way ANOVA: $F_{1.089,2.179} = 790.8$, ***$p < 0.001$, followed by Fisher's LSD: '2 h'/'4 h':*$p = 0.041$; '4 h'/'12 h':*$p = 0.032$; '12 h'/'no blocking': **$p = 0.002$. **d** All TNR epitopes (left) or newly-emerged epitopes 12 h post-blocking (right) were revealed (magenta, STED imaging). Membranes of a subset of neurons were labeled using sparse DiO labeling (green, confocal imaging). Presynapses were identified by VGlut1 (blue, STED imaging). Scale bars: 1 μm (full images), 500 nm (insets). Bottom: hundreds of synapses were averaged by centering synapse images on the VGlut1 puncta and orienting the dendritic DiO signals vertically. "All" TNR epitopes cover the entire bouton-dendrite area (left), while newly-emerged epitopes are enriched in the bouton region (right). **e** Quantification of TNR exchange at synapses (as in **c**, measuring exclusively TNR at VGlut1-labeled synapses). $N = 3$ independent experiments with >100 synapses. **f** Comparison of newly-emerged TNR epitopes 12 h post-blocking in cultures treated with bicuculline (40 μM), or CNQX (10 μM) and AP5 (50 μM). Intensity is normalized to the corresponding control (DMSO). $N = 3$ experiments, ≥10 neurons per datapoint. One-way ANOVA on rank: $F_{2,6} = 42$, ***$p < 0.001$, followed by Dunn's multiple comparisons test: 'ctrl'/'CNQX + AP5': **$p = 0.003$; 'ctrl'/'bic': *$p = 0.042$. Data represent the mean ± SEM, dots indicate individual experiments. **g**, Analysis of 2-color-STED images (as shown in **d**). Synaptic enrichment is substantially higher for newly-emerged epitopes. $N = 3$ independent experiments with >100 synapses. Repeated-measures one-way ANOVA on log-transformed data: $F_{1.977,3.954} = 24.13$, **$p = 0.006$, followed by Fisher's LSD: 'new'/'all' epitopes: *$p = 0.024$ (dendrites); *$p = 0.036$ (axons). Data represent the mean (line) ± SEM (shaded region). Source data are provided in Source Data file.

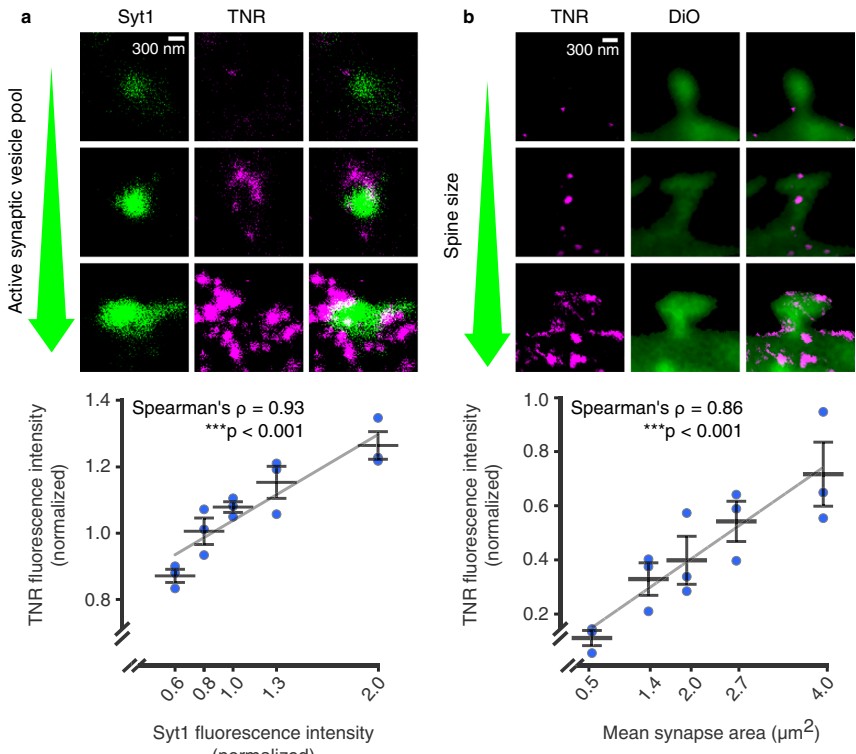

**Fig. 3 The emergence of TNR epitopes is dependent on synaptic weight. a** The TNR epitopes in the ECM were blocked as in the previous experiments, and 12 h later the cultures were incubated with fluorophore-conjugated TNR antibodies (magenta) and with fluorophore-conjugated antibodies for the intra-vesicular domain of Syt1 (green), which reveal the synaptic vesicle pool that undergoes exo- and endocytosis (the active pool). The size of this pool is a measure of the activity of the respective boutons. The panels show example synapses with different active vesicle pools, imaged in STED (TNR) and confocal (Syt1). Scale bar = 300 nm. The graph shows the mean fluorescence intensities normalized to the median intensity of the respective experiment. The Syt1 intensities are binned to include an equal number of synapse images. An analysis of the correlation of the TNR signal at Syt1-labeled synapses indicates that the TNR signals correlate strongly with the size of the active vesicle pool in the respective boutons. $N = 3$ independent experiments, with >1100 synapses per datapoint, Spearman's ρ = 0.927, ***$p = 6.489 \times 10^{-7}$ (two-sided). **b** Newly-emerged TNR epitopes (magenta) were labeled after 12 h as in panel a, and the neuronal plasma membrane was visualized with DiO (green). The panels show example spines with different sizes, imaged in STED (TNR) and confocal (DiO). Scale bar = 300 nm. The graph shows the mean fluorescence intensity of TNR and the mean synapse area, normalized to the median values in the respective experiment. The synapse area values are binned to include an equal number of synapse images. An analysis of the correlation of the TNR signal at DiO-labeled spines indicates that the TNR signals correlate strongly with the size of the dendritic spine for newly-emerged TNR epitopes. $N = 3$ independent experiments, with >280 synapses per datapoint, Spearman's ρ = 0.862, ***$p = 3.601 \times 10^{-5}$ (two-sided). All data represent the mean ± SEM, with dots indicating individual experiments. Source data are provided in Source Data file.

from their epitopes, which would allow the fluorophore-conjugated ones to take their place. In control experiments, we found no evidence for such un-binding, either from the surface of fixed cells at 37 °C (Supplementary Fig. 8) or in live cells at 4 °C (Supplementary Fig. 9). A second possible source for such epitopes would be a pre-existing population of molecules that were present on the cell surface, but were previously unavailable to antibody binding, due to effects such as steric hindrance. Such molecules would be revealed by antibodies when the steric hindrance is eliminated by treatments that change the neuronal surface profoundly. To test this, we subjected the neurons to treatments that modify the membrane proteins (aldehyde treatment), or that remove glycan chains (chondroitinase ABC). We found that such treatments cause no changes in TNR epitope availability (Supplementary Fig. 10a). A third potential source of new TNR epitopes is the proteolytic cleavage of pre-existing surface TNR molecules or their binding partners, which would reduce steric hindrance and make TNR epitopes available for antibody binding. We found no evidence for this, since the TNR epitopes appeared in the same fashion after blocking the activity of matrix metalloproteinases that might cleave existing ECM (Supplementary Fig. 10b).

We therefore conclude that most neurons contain a dynamic pool of TNR molecules, which appear on the surface preferentially near synapses. PNN-containing neurons also exhibit similar TNR dynamics, having higher levels of both total TNR and newly-emerged TNR (albeit somewhat lower ratios of new TNR to total TNR; Supplementary Fig. 11). Moreover, as expected from the limited involvement of astrocytes in TNR endocytosis (Supplementary Fig. 3), we observed ample emergence of new TNR epitopes also in hippocampal neurons that were grown at a large distance above an astrocyte feeder layer (Banker cultures[26]; Supplementary Fig. 12).

**The newly-emerged TNR epitopes recycle over 3 days in culture**. We imaged these epitopes in living cells, and found that after surfacing they are endocytosed on a time scale of hours (Fig. 4a). To visualize the location of the internalized molecules in neurites, we allowed endocytosis to proceed for several hours, and then eliminated all surface molecules by a proteinase K treatment (Fig. 4b). We found internalized TNR to be present not only in the cell body (where it is prominent, Fig. 4a), but also in both axons and dendrites (Fig. 4b). To then verify whether these molecules resurface, we designed an assay in which we tested the amounts of antibody-labeled TNR present on the surface at different timepoints (Fig. 4c). Immediately after antibody labeling of TNR, the antibodies are found mostly on the surface, as expected, and neurites are fully visible. A day or two later, the antibodies are no longer on the surface, as they have been endocytosed, and, since many of the organelles have already reached the cell body (Fig. 4a), neurites are virtually invisible. Remarkably, this situation changes at 3 days after labeling, and a high proportion of the antibodies are again on the surface, especially in neurites (Fig. 4c). These antibodies are later again endocytosed, and will return to the surface after another 3 days (Fig. 4c).

To verify the recycling assay, we also targeted Syt1, as a molecule that is known to recycle, and the EGF receptor, as a molecule that is endocytosed, but does not recycle readily. Both molecules were on the surface at "time 0", and were endocytosed after 15 min. A proportion of the Syt1 molecules returned to the surface after another 45 min, while no EGF receptors returned (Supplementary Fig. 13).

This assay therefore suggests that TNR molecules endocytose and are then repeatedly recycled. Importantly, this conclusion is independent of any problems with, for example, antibodies leaving their epitopes. Random un-binding of blocking antibodies

would allow fluorophore-conjugated TNR antibodies to take their place, therefore resulting in immobile, non-dynamic spots on the surface. The un-binding of fluorophore-conjugated TNR antibodies would simply make the respective TNR molecules invisible. Therefore, none of these scenarios would report either endocytosis or recycling of TNR, implying that our interpretation is independent of such problems (which anyway appear to be negligible, Supplementary Figs. 8–10).

However, a more significant problem is that antibodies may cross-link the TNR molecules, and may thereby change their behavior. To control for this, we repeated our key experiments using monovalent Fab fragments. They could be successfully used for our 'blocking-labeling' assay (Supplementary Fig. 14a–d), and showed the recycling of TNR in the same fashion as the antibodies (Supplementary Fig. 14e). The use of Fab fragments also enabled a more elaborate labeling experiment, based on the detection of Fab fragments with unconjugated or fluorophore-conjugated anti-mouse nanobodies (Fig. 5a). This strategy only reveals TNR epitopes that have completed an entire cycle of endocytosis and resurfacing, and again showed ample signals at ~3 days after the initiation of the experiment (Fig. 5b).

To verify this important conclusion by an experiment not involving the live application of affinity tags, like antibodies or Fab fragments, we turned to the use of purified, recombinant TNR containing a His-tag. Exogenous TNR applied to the culture medium has been previously demonstrated to integrate into the ECM and promote assembly of aggrecan in PNNs[27]. Recombinant TNR was identified, at any desired timepoint, by immunostaining for the His-tag. The recombinant TNR incorporated well into the cultures, as expected, and was more prominently seen on PNN-exhibiting cells, again as expected (Fig. 5c). We then analyzed its location over time. Immediately after application it was found mostly on the cell surface, and it was mostly internalized after 1 day in culture. The recombinant TNR returned to the cell surface after 3 days, in the same fashion as we observed for the experiments involving antibodies or Fab fragments (Fig. 5d, e).

We therefore conclude that a dynamic population of TNR molecules surfaces regularly, preferentially near synapses, and is then endocytosed and recycled over the course of a few days.

**Endocytosed TNR molecules reach the Golgi apparatus**. To first validate the endocytosis of TNR molecules, we marked the newly-emerged TNR molecules using fluorescently-conjugated antibodies, we allowed them to endocytose, and we then applied LysoTracker to the neurons, which labels virtually all acidic organelles of the neurons, including synaptic vesicles[28]. In the live cells, we observed that ~70% of the TNR spots colocalized with the organelle marker, which provides ample evidence that these molecules had been endocytosed (Fig. 6a).

To identify the compartments to which the TNR molecules were internalized, we immunostained the cells for an assortment of intracellular targets, and searched for a colocalization with internalized TNR (Fig. 6b–g). We observed that only a small quantity of TNR molecules was found in Rab5-positive early endosomes and Rab7- or Rab11-positive late or recycling endosomes (Fig. 6g). This, however, does not demonstrate that these organelles do not participate in TNR dynamics, since their slow recycling kinetics (days) implies that only a handful of molecules will be found, at any given time, in compartments involved in rapid molecule sorting, as these endosomes. More importantly, we found that a significant number of molecules colocalized with the Golgi apparatus, including dendritic Golgi outposts (Fig. 6e, h), and with the endoplasmic reticulum (ER). A

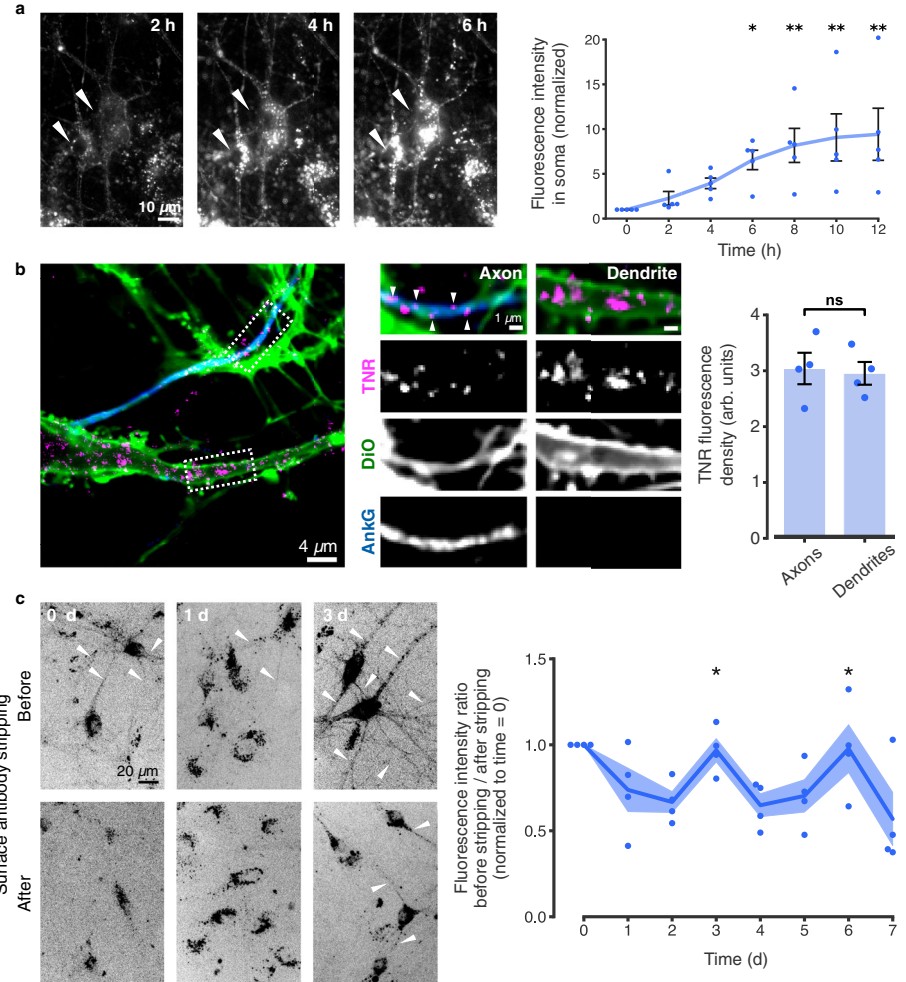

**Fig. 4 Dynamic TNR molecules are endocytosed in neurons over hours, and recycle with a periodicity of ~3 days. a** Newly-emerged TNR epitopes were labeled 4 h post-blocking, and monitored by live epifluorescence imaging. Arrowheads indicate neuronal somas. Scale bar = 10 μm. Quantification of the intensity in somas (normalized to $t_0$ timepoint) indicates significant internalization. N = 5 independent experiments, 1-4 neurons per datapoint. Friedman test ($\chi^2_6 = 25.46$, ***$p < 0.001$), followed by two-sided Dunn's multiple comparisons test ('6 h': *$p = 0.033$; '8 h': **$p = 0.005$; '10 h': **$p = 0.005$; '12 h': **$p = 0.002$). Data represent mean ± SEM, dots indicate individual experiments. **b** Internalized TNR in axons vs. dendrites. Epitopes were allowed to internalize for 12 h, followed by surface stripping with proteinase K. The remaining signal was imaged with confocal microscopy in neurites visualized using DiO (green), with axons identified by immunostaining AnkyrinG (blue). Scale bar = 4 μm (1 μm zoom). Quantification of the signal density reveals no differences between dendrites and axons. N = 4 experiments, ≥10 neurons per datapoint. Two-sided paired t-test (t = 0.741, $p = 0.513$). Data represent mean ± SEM, dots indicate individual experiments. **c** Newly-emerged TNR epitopes were labeled 6 h post-blocking. The fraction present on the surface of neurites was measured at different intervals by imaging neurons in epifluorescence before and after stripping with proteinase K. Quantification of the fluorescence ratio before/after stripping (normalized to the '0d' timepoint) reveals peaks of TNR resurfacing at 3 and 6 days post-labeling (~3 day periodicity). Amounts stripped at '3d' and '6d' are significantly higher than at '1d' and '2d', or '5d' and '7d'. N = 4 independent experiments, 5 before/after images per datapoint. Kruskal-Wallis followed by Fisher's LSD (Days 2, 3, 4: $H_2 = 8.29$, *$p = 0.016$, '3d'/'2d'; *$p = 0.046$; '3d'/'4d': **$p = 0.005$; Days 4, 5, 6: $H_2 = 6.74$, *$p = 0.036$, '6d'/'5d': *$p = 0.022$, '6d'/'7d': *$p = 0.028$). Scale bar = 20 μm. Data represent the mean (lines) ± SEM (shaded regions); dots indicate individual experiments. Source data are provided in Source Data file.

proportion of the TNR molecules could also be found in lysosomes, pointing to some degree of degradation. These observations also explain why a measurable proportion of TNR spots are not found in acidified organelles, since the ER tubules have a neutral pH.

To further verify that TNR molecules rely on intracellular trafficking for their recycling pathway, we blocked dynamin, which is thought to be involved in most endocytosis reactions, and we also perturbed cellular trafficking with monensin or brefeldin. Dynamin inhibition with Dyngo reduced substantially TNR endocytosis (Supplementary Fig. 15a). Monensin and brefeldin also reduced the appearance of newly-emerged TNR epitopes (Supplementary Fig. 15b). Inhibition of dynamin could

also be demonstrated to prevent the recycling of TNR (Supplementary Fig. 15c). Moreover, the involvement of dynamin in TNR dynamics could be verified by a dynamin knock-down (Supplementary Fig. 15d–g). Finally, dynamin could be co-immunoprecipitated from synaptosomal material together with TNR (Supplementary Fig. 15h), suggesting that the two molecules interact either directly or indirectly, thereby implying again that TNR spends a substantial proportion of its lifetime in intracellular compartments that rely on dynamin function.

**Integrins are involved in TNR recycling.** In addition to operating as a scaffold, the ECM actively regulates neuronal function

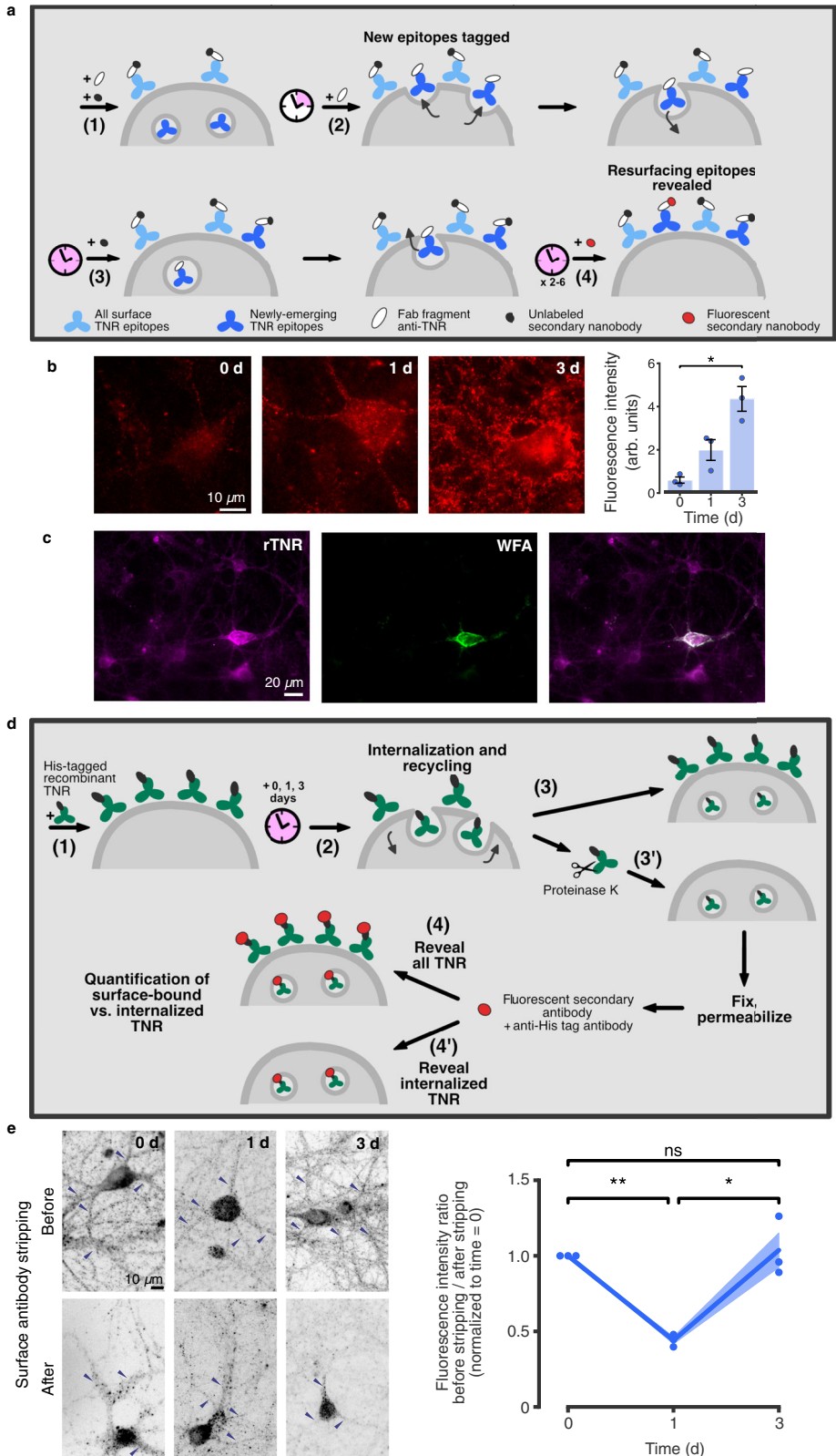

by interacting with ECM receptors on the plasma membrane, such as the integrins, which link the ECM to the cell cytoskeleton[29]. A class of integrin receptors containing the β1 subunit has been shown to functionally interact with TNR[30] and to be particularly enriched at hippocampal synapses, where they mediate outcomes on dendritic spine motility and LTP development[31]. Since the endocytosis and recycling of integrins are well-established phenomena[32], we wondered whether the trafficking of the recycling TNR pool might be related to β1-integrins.

We examined the colocalization of TNR with β1-integrin at two stages of its recycling pathway, by labeling surface-bound β1-integrins together with the newly-emerged TNR epitopes (Fig. 7a, b). We found that the recycling TNR molecules often colocalized

**Fig. 5 The 3 day-long recycling observed by labeling with Fab fragments or His-tagged TNR. a** Assay to label molecules completing a full endocytosis/ resurfacing cycle. (1) Surface TNR epitopes are blocked with TNR Fab fragments and non-fluorescent secondary nanobodies. (2) 4 h later, newly-emerged epitopes are tagged with new Fab fragments, without secondary nanobodies. (3) Following a 12-h incubation, allowing for internalization, newly-emerged epitopes remaining at the surface are blocked with non-fluorescent nanobodies. (4) Immediately afterwards, or 1–3 days later, the newly-emerged and then internalized epitopes that resurfaced are revealed with fluorophore-conjugated secondary nanobodies. **b** Neurons were imaged in epifluorescence microscopy. Substantial fluorescence is visile at both the 1- and 3-day time points. $N = 3$ independent experiments, ≥15 neurons per datapoint. Kruskal-Wallis ($H_2 = 7.2$, *$p = 0.0273$), followed by two-sided Dunn's multiple comparisons test: '0 d'/'3 d': *$p = 0.0199$. Scale bar = 10 µm. Data represent mean ± SEM, dots indicate individual experiments. **c–e** Recycling of recombinant His-tagged TNR (rTNR). **c** rTNR distributes similarly to endogenous TNR, after pulsing neurons with rTNR for 1 h and staining with WFA to label PNNs (epifluorescence). Scale bar = 20 µm. $N = 3$ independent experiments. **d** rTNR recycling assay: (1) Neurons were pulsed with rTNR for 1 h, and then incubated for 0–3 days, allowing for internalization and recycling (2). Neurons were fixed immediately (3), or first incubated with proteinase K to remove surface-bound rTNR (3'). Neurons were permeabilized and immunostained with anti-His tag antibodies to reveal all rTNR (4), or internalized rTNR (4'). **e** At time = 0, rTNR staining was strongly reduced by stripping. At 1d, similar staining was observed in stripped/non-stripped cultures. At 3d, staining was again reduced after stripping. Scale bar = 10 µm. $N = 3$ independent experiments, 5 before/after images per datapoint. Repeated-measures one-way ANOVA ($F_{1.044, 2.088} = 28,6$, *$p = 0.03$), followed by Fisher's LSD ('0 d'/'1 d': **$p = 0.002$; '1 d'/'3 d': *$p = 0.027$; '0 d'/'3 d': $p = 0.775$). Data represent mean (lines) ± SEM (shaded regions), dots indicate individual experiments. Source data are provided in Source Data file.

with surface-bound β1-integrins immediately after their emergence, as well as with internalized β1-integrins 12 h later (Fig. 7a, b). This implies that TNR recycling relies on organelles involved in β1-integrin dynamics. To verify this hypothesis in a more direct fashion, we used an antibody that blocks β1-integrin[33]. This reduced profoundly the TNR internalization (Fig. 7c), suggesting that this molecule indeed serves as a receptor involved in the recycling of TNR-containing ECM.

**The purpose of TNR recycling may be to renew the glycosylation of these molecules**. While recycling processes through different endosomal systems are well understood, and have long been discussed in synapses[13,34], TNR recycling through the Golgi/ER appears rather unusual. Such a pathway would be needed, however, if the surface-exposed TNR molecules suffer modifications to their sugar moieties, and therefore require re-glycosylation, and/or a specific glycosylation pattern is needed to recycle perisynaptic TNR. This type of Golgi/ER recycling and re-glycosylation pathway has been less investigated than many other trafficking reactions, but has been demonstrated for several cell surface glycoproteins, especially in liver cells[35]. To test whether TNR follows such a pathway, we labeled newly O-glycosylated proteins by feeding the neurons with azide-modified galactosamine (GalNAz) and glucosamine (GlcNAz), which were then revealed by tagging with a fluorophore, using a click chemistry reaction[36]. We found that the recycling TNR pool colocalized to a significant extent with GalNAz, and less with GlcNAz (Fig. 8). This in agreement with previous studies that showed that GalNAc (but not GlcNAc) is a dominant component of O-linked glycosylations on TNR[37–39]. As a control, we performed the same experiment by feeding the neurons with the methionine analog azidohomoalanine (AHA), which incorporates into de novo synthesized proteins and is then similarly tagged using click chemistry[40], and we found little colocalization between TNR and AHA (Fig. 8), in agreement with the expectation that this is a very long-lived protein[3].

The resolution of the imaging approach we used (two-color STED microscopy) is insufficient to demonstrate whether the recycling TNR molecules are themselves newly glycosylated, or whether their presence in the ER/Golgi compartments simply places them near newly-glycosylated proteins, resulting in the colocalization we measured optically. However, if the latter were true, the TNR molecules would also colocalize with newly-secreted proteins (labeled by AHA), which are abundant in the ER/Golgi compartments. As this was not the case, the overall interpretation of these experiments is that the recycling pool of TNRs consists of molecules that are not metabolically young, and

that their trafficking to the ER/Golgi might function as means of re-glycosylation.

**Perturbing newly-emerged TNR epitopes blocks synaptic function and modifies synaptic structure**. To test whether the dynamic pool of TNR is relevant for synaptic transmission, we performed a crude experiment in which these molecules were bound by large aggregates of antibodies[41], and we then analyzed synaptic vesicle exo-/endocytosis. In brief, we tagged the newly-emerged TNR epitopes with biotin-coupled antibodies (Fig. 9a). These were then immediately bound by large aggregates formed by goat anti-biotin antibodies and donkey anti-goat antibodies (Fig. 9b, c). In principle, such aggregates should block TNR endocytosis, and should perturb severely the bound molecules (Fig. 9a). We then stimulated the synapses in the presence of Syt1 antibodies (as used in Fig. 3), to determine the overall degrees of exo-/endocytosis. Adding the antibody aggregates for just 30 min eliminated presynaptic vesicle release in response to stimulation (Fig. 9d, e). This was only true for the newly-emerged TNR epitopes, as the antibody aggregates had no effect when they were bound specifically to the surface-resident, non-recycling TNR epitopes (Fig. 9e).

The elimination of stimulation-induced vesicle release that we observed was not due to perturbing $Ca^{2+}$ influx in synapses, since the stimulus-induced rise in intracellular $Ca^{2+}$ was not affected by the treatment (Supplementary Fig. 16), which leaves the mechanism linking synaptic vesicle dynamics to TNR open. Hypotheses involving the interactions between TNR, integrins and the presynaptic active zone could be verified in the future[42,43], including the possibility of multipartite interactions between synaptic vesicle proteins, integrins, laminins and TNR[44,45]. An additional perturbation of neuronal activity became evident when we examined the spontaneous $Ca^{2+}$ activity in the absence of stimulation. The addition of antibody aggregates raised substantially the spontaneous firing rates (Supplementary Fig. 16), despite the inhibition of vesicular trafficking. As TNR has previously been shown to interact with voltage-gated $Na^+$ channels ($Na_v s$)[46], this effect may be due to the TNR perturbation affecting $Na_v$ activity, which may, in turn, increase the excitability of the neurons.

To also analyze the effects of these aggregates on structural changes in synapses, we applied them for 12 h, and we then analyzed spine morphology both in dissociated cultures and in organotypic slices (Fig. 9f, g). A significant reduction of spine head sizes was observed both in culture and in slices. As spine head size is an important reporter for synaptic strength[24], this implies that TNR dynamics are closely linked

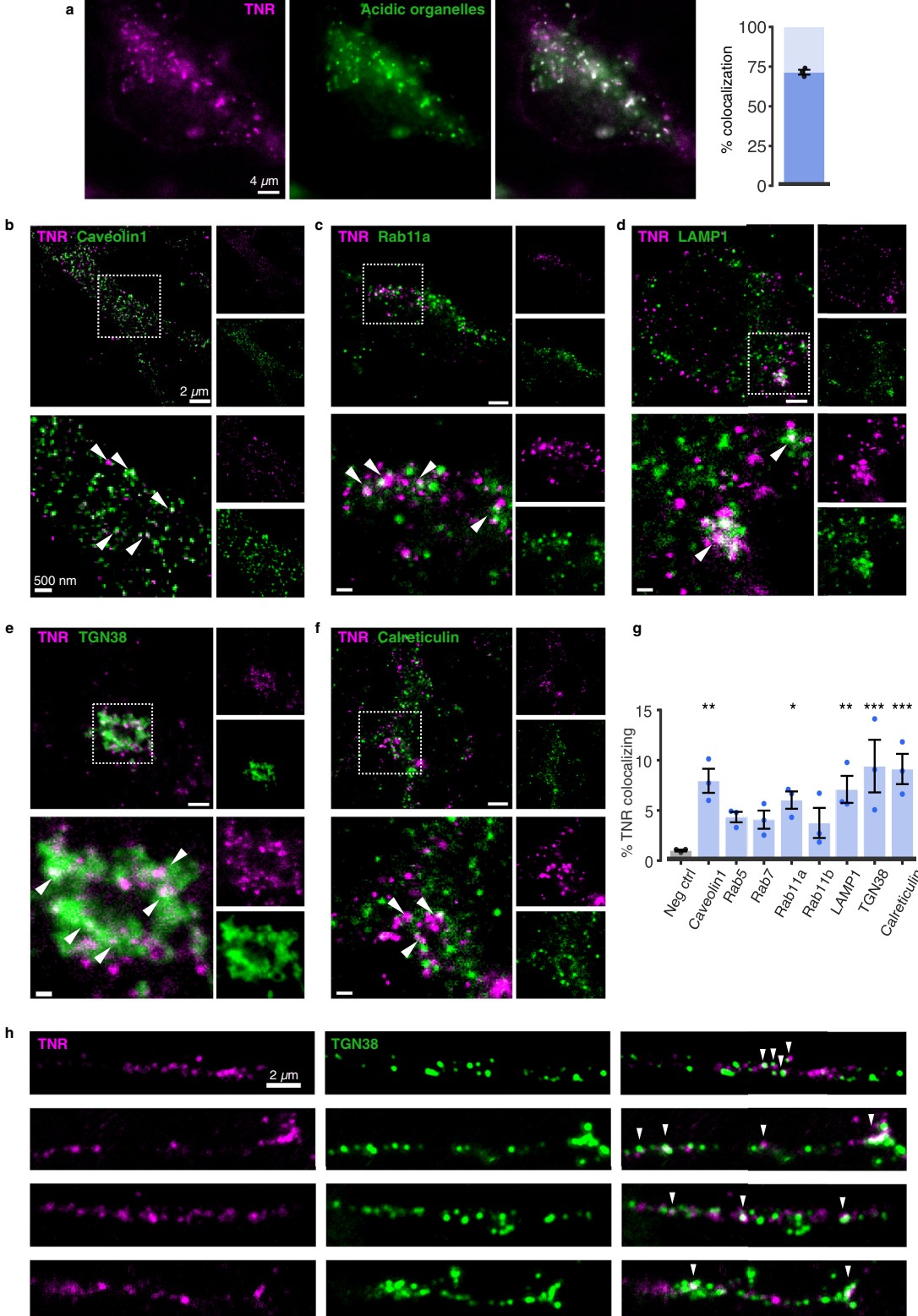

to synaptic plasticity. Importantly, the antibody aggregates had no effect when they were bound to the surface-resident, non-recycling TNR epitopes, as also observed above for the presynaptic dynamics.

At the moment it is difficult to pursue the mechanisms behind this type of manipulation further, without a major effort in developing new tools for interference with TNR recycling. Newly-emerged TNR integrates into the ECM at synapses, along with older TNR molecules (Supplementary Fig. 17), and its localization does not suggest a special positioning within synapses, which could be used to derive interpretations on its function relating to synapse activity and plasticity.

**Fig. 6 An overview of organelles involved in the trafficking of newly-emerged TNR epitopes. a** Newly-emerged TNR epitopes were labeled 12 h post-blocking, concurrently with the application of LysoTracker™ Green, to label acidic organelles. After a 6-h incubation, allowing for internalization, surface TNR was stripped with proteinase K. Neurons were imaged live (epifluorescence). Scale bar = 4 µm. >70% of internalized TNR is present in acidic organelles. N = 3 independent experiments, ≥4 neurons per datapoint. Data represent mean ± SEM, dot indicate individual experiments. **b–g** To identify the compartments containing internalized TNR, newly-emerged TNR epitopes were labeled 12 h post-blocking and allowed to internalize for 6 h, after which remaining surface-bound TNR was stripped with proteinase K. The neurons were fixed and immunostained with organelle markers. Shown are 2-color-STED images of TNR (magenta) and organelle markers (green): caveolin1, Rab11a (recycling endosomes), LAMP1 (lysosomes), TGN38 (trans-Golgi network) and calreticulin (ER). The right side of each panel shows zoomed views of the dashed boxes. Arrowheads indicate colocalizing signals. Scale bar = 2 µm (full images), 500 nm (zoomed images). **g** Quantification of % TNR spots colocalizing with organelle markers, compared to a negative control (using non-specific primary antibodies). TNR colocalizes significantly with ER, TGN, LAMP1, Rab11a and caveolin. N = 3 independent experiments, ≥10 neurons per datapoint. One-way ANOVA ($F_{8,18}$ = 4.284, **$p$ = 0.005), followed by Fisher's LSD to compare all markers with 'neg ctrl' (Caveolin1: **$p$ = 0.002; Rab5: $p$ = 0.099; Rab7: $p$ = 0.126; Rab11a: *$p$ = 0.017; Rab11b: $p$ = 0.169; LAMP1: **$p$ = 0.005; TGN38: ***$p$ < 0.001; Calreticulin: ***$p$ < 0.001). Data represent mean ± SEM, dots indicate individual experiments. **h** A fraction of newly-emerged TNR localizes to dendritic Golgi outposts following endocytosis. Newly-emerged TNR epitopes (magenta) were labeled 4 h post-blocking, and allowed to internalize over 12 h. The neurons were fixed and immunostained with TGN38 to identify dendritic Golgi outposts[78,79]. Representative images, taken with confocal microscopy, are shown. Arrowheads indicate colocalizing signals. Scale bar = 2 µm. N = 4 independent experiments. Source data are provided in Source Data file.

---

**TNR dynamics are also observed in organotypic and in acute hippocampal slices**. Having already observed that TNR manipulations affect synapse structure in cultured organotypic slices (Fig. 9g), we turned to testing the 'blocking-labeling' assay in this model. We observed a similar behavior to the dissociated cultures (Supplementary Fig. 18a). The slices showed higher levels of newly-emerged epitopes after activation by the addition of bicuculline, and lower levels after silencing via CNQX and AP5 (Supplementary Fig. 18b), as observed in Fig. 2 for the dissociated cultures. Moreover, the application of recombinant, His-tagged TNR resulted in ample endocytosis, which was stimulated by bicuculline (Supplementary Fig. 18c).

To come closer to the in vivo situation, we first verified whether substantial TNR amounts could be found within the cell bodies of neurons in the hippocampi of adult mice. This could be indeed observed, using conventional immunostaining techniques (Fig. 10a). Moreover, intracellular TNR was subject to changes according to the functional state of the neurons. To increase the activity rate of the neurons, they were stimulated by in vivo kainic acid administration, in what constitutes a well-studied mouse model of epilepsy[47]. This enhanced the levels of intracellular somatic TNR, while leaving TNR unaffected in other locations (Fig. 10a). Finally, the accumulation of intracellular somatic TNR was not simply an effect of neuronal damage, since a mouse model in which neuronal damage is prominent (5xFAD mice as a model of familial Alzheimer disease), showed no effects on intracellular TNR accumulation (Fig. 10b).

Second, we turned to the question of whether the intracellular TNR molecules are newly synthesized, or are older molecules that the cells have endocytosed from the ECM. To analyze this, we used a technique we introduced in the past, correlated optical and isotopic nanoscopy (COIN[48]). Wild-type mice were pulsed with the essential amino acid lysine containing 6 stable $^{13}$C isotopes, for 14 or 21 days, and hippocampal slices were then immunostained, as above. After imaging the slices, we analyzed them using nanoscale secondary ion mass spectrometry (nanoSIMS). In nanoSIMS a primary $Cs^+$ beam irradiates the sample and causes the sputtering of secondary particles from the sample surface. These particles are partly ionized and are then identified by mass spectrometry. This reveals the $^{13}$C isotopes, and enables us to test whether the TNR-containing spots consisted of newly-synthesized proteins (*i.e.* rich in $^{13}$C isotopes), or whether they contained older proteins (lacking $^{13}$C isotopes; Fig. 10c, d). We found intracellular TNR objects were substantially older than the rest of the cell (Fig. 10e). This implies that they are not newly synthesized, and therefore need to be molecules that the cell has endocytosed from the ECM, in line with our model.

Third, we also sought to verify that acute slices from adult mice can internalize TNR, and that increased neuronal activity enhances TNR internalization. To test this, we applied the His-tagged version of TNR to the slices, and followed its internalization by fluorescence imaging (Supplementary Fig. 19). His-tagged TNR was indeed taken up by the cells, and its uptake was enhanced by stimulation of neuronal activity with bicuculline (Supplementary Fig. 19).

We conclude that our model of TNR recycling is plausible both in brain slices and in vivo.

**Other ECM molecules also show similar dynamics to TNR**. To test whether our observations extend to other ECM molecules, we used the 'blocking-labeling' assay to assess neurocan, chondroitin-sulfate (CS)-bearing proteoglycans (predominantly aggrecan) labeled by *Wisteria floribunda* agglutin (WFA), and hyaluronic acid (HA)[10]. In a similar fashion to TNR, we observed that the amount of newly-emerged epitopes was far larger than would be predicted from their exceptionally long half-lives[2,3], and that the epitope emergence increased after culture activation using bicuculline (Supplementary Fig. 20a–c).

To verify the turnover dynamics of the ECM by a completely different approach, we used a fluorescence recovery after photobleaching (FRAP)-based assay to observe the hyaluronan-binding protein HAPLN1, which is substantially easier to express and monitor than all other molecules tested here (Supplementary Fig. 20d–f, Supplementary Movie 1). HAPLN1 dynamics were far higher than expected according to its lifetime[2,3] and were also significantly faster in synaptic regions, supporting our previous observations for TNR. Moreover, organelle transport of HAPLN1 appeared to take place, as observed in long-term imaging of HALPN1-expressing cultures (Supplementary Movie 1).

While these last experiments do not constitute a direct proof of endocytosis or recycling for these ECM molecules, they complement the more direct assays used for TNR, and suggest that the potential for remodeling by recycling should be analyzed for these molecules as well.

## Discussion

ECM remodeling in the adult brain is thought to occur in sparse, isolated events that take a high metabolic toll on the cells. Based on the current dogma, existing ECM structures are cleaved by secreted matrix proteases to enable synaptic remodeling, and are then re-stabilized by the addition of freshly-synthesized proteins[9,10]. Indeed, numerous studies have demonstrated that synaptic plasticity events (*e.g.* learning) can induce the release of ECM-cleaving enzymes, and

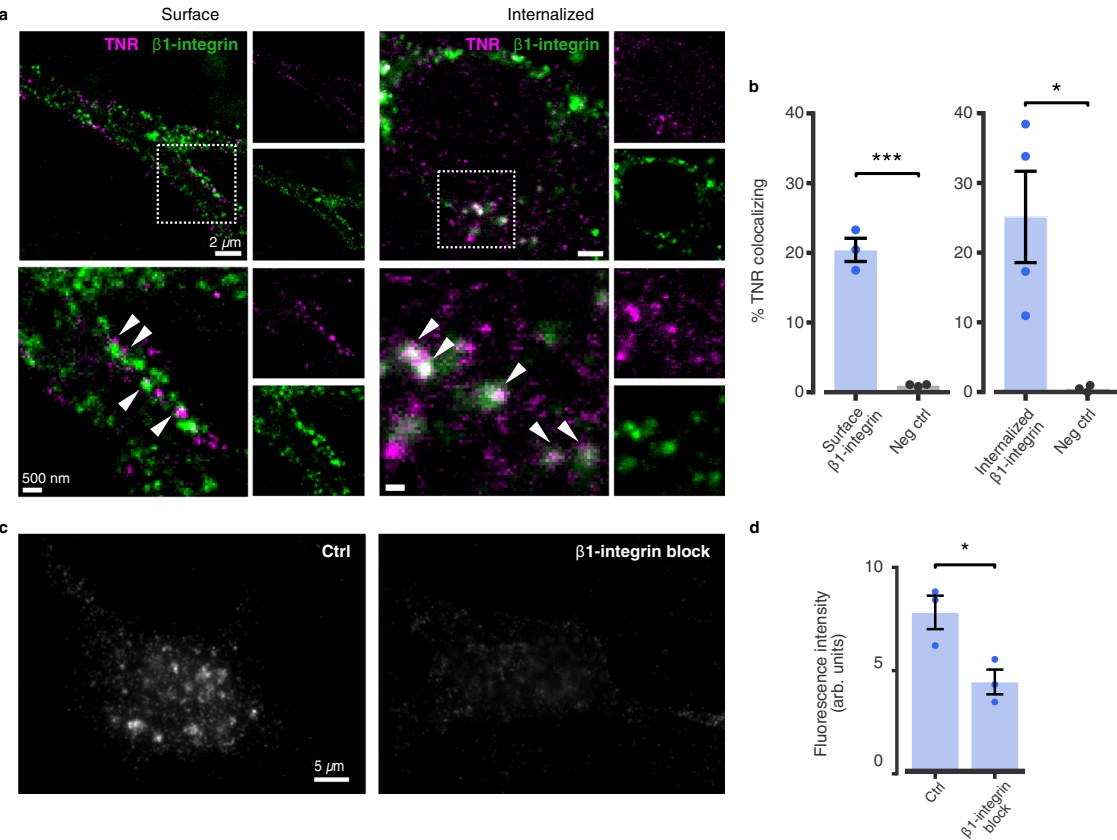

**Fig. 7 TNR recycling is mediated by integrins. a, b** Assessment of colocalization between recycling TNR molecules and β1-integrin. **a** Left: newly-emerged TNR epitopes were labeled 12 h post-blocking concurrently with a labeling of surface-bound β1-integrins, by applying fluorophore-conjugated antibodies directed against the extracellular domain of the receptors. The neurons were fixed and imaged with 2-color-STED. Right: newly-emerged TNR epitopes were labeled 4 h post-blocking, concurrently with β1-integrin. Neurons were incubated a further 12 h to allow for internalization, and remaining surface-bound molecules were stripped with proteinase K. The neurons were fixed and imaged with confocal microscopy. Images on the right of each panel show zoomed views of the dashed boxed. Scale bars = 2 μm (full images), 500 nm (zoomed images). **b** Quantification of % colocalizing TNR signal (for **a**) shows newly-emerged TNR epitopes colocalize with both cell surface-bound and internalized β1-integrins. The values are significantly higher than negatives controls, relying on non-specific primary antibodies. Controls were imaged in STED/confocal for comparison to images in the left/right panels, respectively. $N = 3$ ('surface β1-integrin' experiments and negative controls), and 4 ('internalized β1-integrin') independent experiments, ≥10 neurons per datapoint. Two-sided Student's $t$-test ('surface integrin' vs. 'neg ctrl': $t = 11.61$, ***$p = 0.0003$; 'internalized integrin' vs. 'neg ctrl': $t = 3.177$, *$p = 0.025$. Data represent mean ± SEM, dots indicate individual experiments. **c** To assess whether β1-integrin receptors are required for TNR endocytosis, newly-emerged TNR epitopes were labeled 12 h post-blocking, after which the neurons were immediately incubated with function-blocking anti-β1-integrin antibodies[33] for 6 h. Neurons were then incubated with proteinase K, to remove remaining surface-bound TNR, and imaged with epifluorescence microscopy. A reduction in fluorescence signal is evident in integrin-blocked cultures. Scale bar = 5 μm. **d** Quantification of the fluorescence intensity confirms that the amount of internalized TNR is significantly reduced following the blocking of β1-integrin receptors. $N = 3$ independent experiments, ≥15 neurons per datapoint. Two-sided Student's $t$-test ($t = 3.343$, *$p = 0.029$). Data represent mean ± SEM, dots indicate individual experiments. Source data are provided in Source Data file.

result in a transient upregulation of ECM protein synthesis[49–51]. While this notion of ECM remodeling can account for infrequent events such as synaptic plasticity, it is at odds with the observation that synaptic morphology changes continually, even at rest. For example, long-term imaging of synapses in slices of rat hippocampi revealed that dendritic spines continue to be generated and eliminated, and undergo significant volume changes, even when synaptic plasticity is suppressed[52]. Such severe changes to synapses are likely to necessitate the reorganization of peri-synaptic ECM[6–8,53–55]. We would therefore expect that ECM remodeling does not solely occur during infrequent plasticity events[56], but is rather a constitutive process in the brain. In this case, the components of the peri-synaptic ECM would need to turn over frequently, at a significantly higher rate than would be expected from their extremely long lifetimes[2,3]. The discrepancy between the high frequency of remodeling events and the slow rate of ECM protein synthesis can be reconciled by the existence of a mobile, recycling pool of ECM molecules that can be continually incorporated and re-internalized at synapses, without the need for novel protein secretion, as we found here.

Overall, our results demonstrate that the neural ECM is significantly more plastic than previously assumed. As this mechanism may not be limited to TNR, our observations open a new field of investigation that should prove important in understanding not only ECM regulation in the brain, but also brain plasticity and stability in general. Finally, as ECM changes are known to accompany a plethora of brain diseases, these findings should also prove relevant for clinical research in the future.

## Methods
In sections where the model is not indicated (or indicated as 'neuronal cultures'), the experiments were performed on dissociated primary hippocampal cultures beginning at DIV14-16.

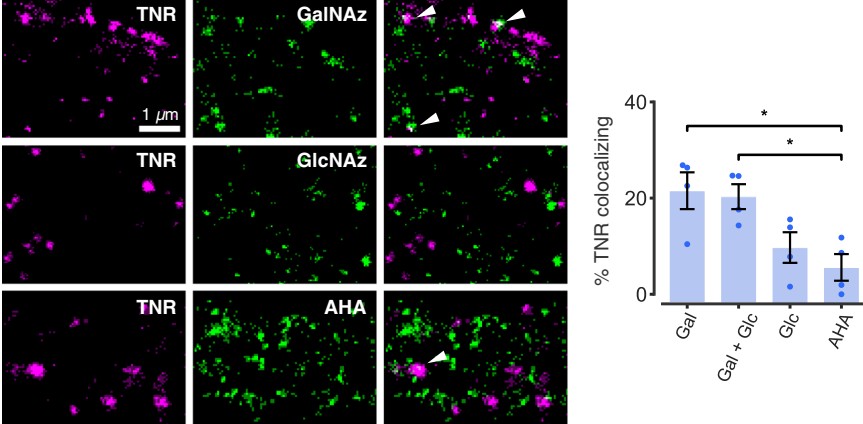

**Fig. 8 TNR recycling possibly relates to TNR re-glycosylation.** Newly O-glycosylated proteins were labeled by feeding neurons with azide-modified galactosamine (GalNAz) and/or glucosamine (GlcNAz), which were then revealed by click chemistry. Alternatively, newly-synthesized proteins were labeled by feeding neurons with azidohomoalanine (AHA), which was also tagged using click chemistry. Newly-emerged TNR epitopes were labeled 6 h post-blocking and visualized at the surface. The neurons were imaged with 2-color-STED. Scale bar = 1 μm. Quantification of the colocalization of the signals confirmed that internalized TNR epitopes colocalize significantly with GalNAz or GalNAz+GlcNAz, at levels substantially higher than the minimal AHA colocalization, which is not significantly different from negative controls (relying on non-specific primary antibodies). N = 4 independent experiments, ≥10 neurons per datapoint. Kruskal-Wallis test (H₃ = 9.022, *p = 0.029), followed by two-sided Fisher's LSD (*p = 0.014 and *p = 0.021 for 'GalNAz' and 'GalNAz+GlcNAz' respectively). Data represent mean ± SEM, dots indicate individual experiments. Source data are provided in Source Data file.

**Animals**. All animals were handled according to the specifications of the University of Göttingen or DZNE Magdeburg and of the local authorities, the State of Lower Saxony (Landesamt für Verbraucherschutz, LAVES, Braunschweig, Germany) and State of Saxony-Anhalt (Landesverwaltungsamt, Halle, Germany). All animal experiments and tissue collection were performed in accordance with the European Communities Council Directive (2010/63/EU) and approved by the local authority, the Lower Saxony State Office for Consumer Protection and Food Safety (Niedersächsisches Landesamt für Verbraucherschutz und Lebensmittelsicherheit) or by the Ethical Committee on Animal Health and Care and the local authority of the State of Saxony-Anhalt, Germany (license numbers: 42502-2-1316 DZNE and 42502-2-1322 DZNE).

**Preparation of rat dissociated hippocampal cultures**. Dissociated primary hippocampal cultures were prepared from newborn rats (*Rattus norvegicus*) as previously described[26,57]. Briefly, hippocampi of newborn Wistar rat pups were dissected in HBSS (140 mM NaCl, 5 mM KCl, 6 mM glucose, 4 mM NaHCO₃, 0.3 mM Na₂HPO₄ and 0.4 mM KH₂PO₄) and incubated for one hour in enzyme solution (DMEM containing 0.5 mg/mL cysteine, 100 mM CaCl₂, 50 mM EDTA and 2.5 U/mL papain, bubbled with carbogen for 10 min). Dissected hippocampi were then incubated for 15 min in a deactivating solution (DMEM containing 0.2 mg/mL bovine serum albumin, 0.2 mg/mL trypsin inhibitor and 5% fetal calf serum). The cells were triturated and seeded on circular glass coverslips (18 mm diameter) at a density of ~ 80,000 cells per coverslip. Before seeding, the coverslips were treated with nitric acid, sterilized, and coated overnight with 1 mg/mL poly-L-lysine. The neurons were allowed to adhere to the coverslips for 1–4 h at 37 °C in plating medium (DMEM containing 3.3 mM glucose, 2 mM glutamine, and 10% horse serum), after which they were switched to Neurobasal-A medium (Life Technologies, Carlsbad, CA, USA) containing 2% B27 (Gibco, Thermo Fisher Scientific, USA) supplement, 1% GlutaMax (Gibco, Thermo Fisher Scientific, USA) and 0.2% penicillin/streptomycin mixture (Biozym Scientific, Germany). The cultures were maintained in a cell incubator at 37 °C, and 5% CO₂ for 14–16 days before use, and the culture medium was replaced at most once per week, if the pH indicator suggested a loss of medium quality. Percentages represent volume/volume.

**Preparation of organotypic hippocampal slice cultures**. Organotypic hippocampal slice cultures were prepared as previously described[58], with the modifications described in[59]. In brief, hippocampi of postnatal day 3 (P3) C57BL/6 J mice (*Mus musculus*) were isolated, and 300-μm thick transverse slices were cut and placed on support membranes (Millicell-CM Inserts, PICMORG50; Millipore). The surface of the slices was covered with culture medium consisting of 50% MEM with Earle's salts (#M4655; Merck, Germany), 25 mM HEPES, 6.5 mg/ml glucose, 25% horse serum, 25% Hanks solution buffered with 5 mM Tris and 4 mM NaHCO₃, pH 7.3. The slices were maintained in a cell incubator at 37 °C and 5% CO₂ for 14 days before use, and the culture medium was replaced every other day. Percentages represent volume/volume.

**Cell-surface biotinylation assay**

*Biotinylation and glutathione treatment*. The assay was adapted from[15]. Briefly, neurons were incubated with 100 μM Leupeptin (#L2884; Merck, Germany-Aldrich,

Germany) for 1 h at 37 °C, to inhibit lysosomal protein degradation. Leupeptin was also present in the cell media throughout the remainder of the experiment. The neurons were incubated with 1.5 mg/mL EZ-Link™ Sulfo-NHS-S-S-Biotin (#21331; Thermo Fisher Scientific, USA) in PBS for 30 min at 37 °C. The neurons were subsequently washed in PBS containing 10 mM glycine to quench the unreacted biotin. The neurons were either immediately scraped into lysis buffer, to detect the entire surface pool (50 mM Tris-HCl, 150 mM NaCl, 2 mM EDTA, 0.5% IGEPAL, 0.5% sodium deoxycholate, 0125 mM PMSF protease inhibitor, and 1x protease inhibitor cocktail: #87786; Thermo Fisher Scientific, US), or were returned to their original culture media and incubated at 37 °C, to allow for endocytosis of cell-surface proteins. After a further 6 h, the neurons were incubated with glutathione cleavage buffer containing 50 mM glutathione (#G6013; Merck, Germany), 75 mM NaCl, 10 mM EDTA, 75 mM NaOH and 1% BSA in H₂O, for 20 min at 4 °C. Subsequently, the neurons were either returned to their original cell media supplemented with 10 mM glutathione, or quenched in iodoacetamide buffer containing 50 mM iodoacetamide (#A1666; Applichem GmbH, Germany) and 1% BSA in PBS, for 30 min at 4 °C, and were immediately scraped into lysis buffer, to reveal the endocytosed pool of molecules. The remaining neurons were incubated a further 18 h at 37 °C, to allow for the resurfacing of endocytosed proteins, and were then subjected to a second glutathione cleavage reaction to cleave, the newly-surfacing biotinylated proteins. The neurons were then quenched in iodoacetamide buffer and scraped in lysis buffer, thereby revealing the non-recycling pool.

*Precipitation of biotinylated proteins and immunostaining*. Biotinylated cell-surface proteins were pulled down with streptavidin-coupled magnetic beads (#11205D; Thermo Fisher Scientific, US). The beads were isolated, were washed, and were then blocked with PBS containing 2.5%, bovine serum albumin (BSA) (A1391-0250; Applichem, Germany) and 0.1% Tween20 (9005-64-5; Merck, Germany) for 1 h. They were then immunostained with 1:500 monoclonal mouse anti-TNR (#217 011; Synaptic Systems, Göttingen, Germany), 1:100 monoclonal mouse anti-Syt1 (#105 311; Synaptic Systems, Göttingen, Germany), 1:100 monoclonal mouse anti-calmodulin (#MA3-917; Thermo Fisher Scientific, US), 10 μg/mL monoclonal mouse anti-LAMP1 (#MA1-164; Thermofisher Scientific, USA) or 1:1000 monoclonal mouse anti-myelin basic protein (#NBP1-05203, Novus Biologicals, Germany), together with 1:100 STAR635P-conjugated anti-mouse secondary nanobodies (NanoTag, Göttingen, Germany), overnight at 4 °C in. The beads were subsequently washed and mounted on glass slides in Mowiol for imaging.

*Imaging*. Confocal imaging was performed on a Leica TCS SP5 microscope (Leica, Wetzlar, Germany) equipped with an HCX Plan Apochromat 63×1.4 NA oil objective. The 561 nm or 633 nm lines of a Helium-Neon laser were utilized for excitation, using acousto-optic tunable filters to select appropriate emission wavelengths. The images were acquired with photomultiplier tubes. For each channel the pinhole was set to 1 Airy unit.

**Blocking-labeling assay and live treatments**

*TNR blocking-labeling with antibodies*. To block surface epitopes of TNR, neurons were incubated with knock-out-validated antibodies[19] (#217 011; clone 619;

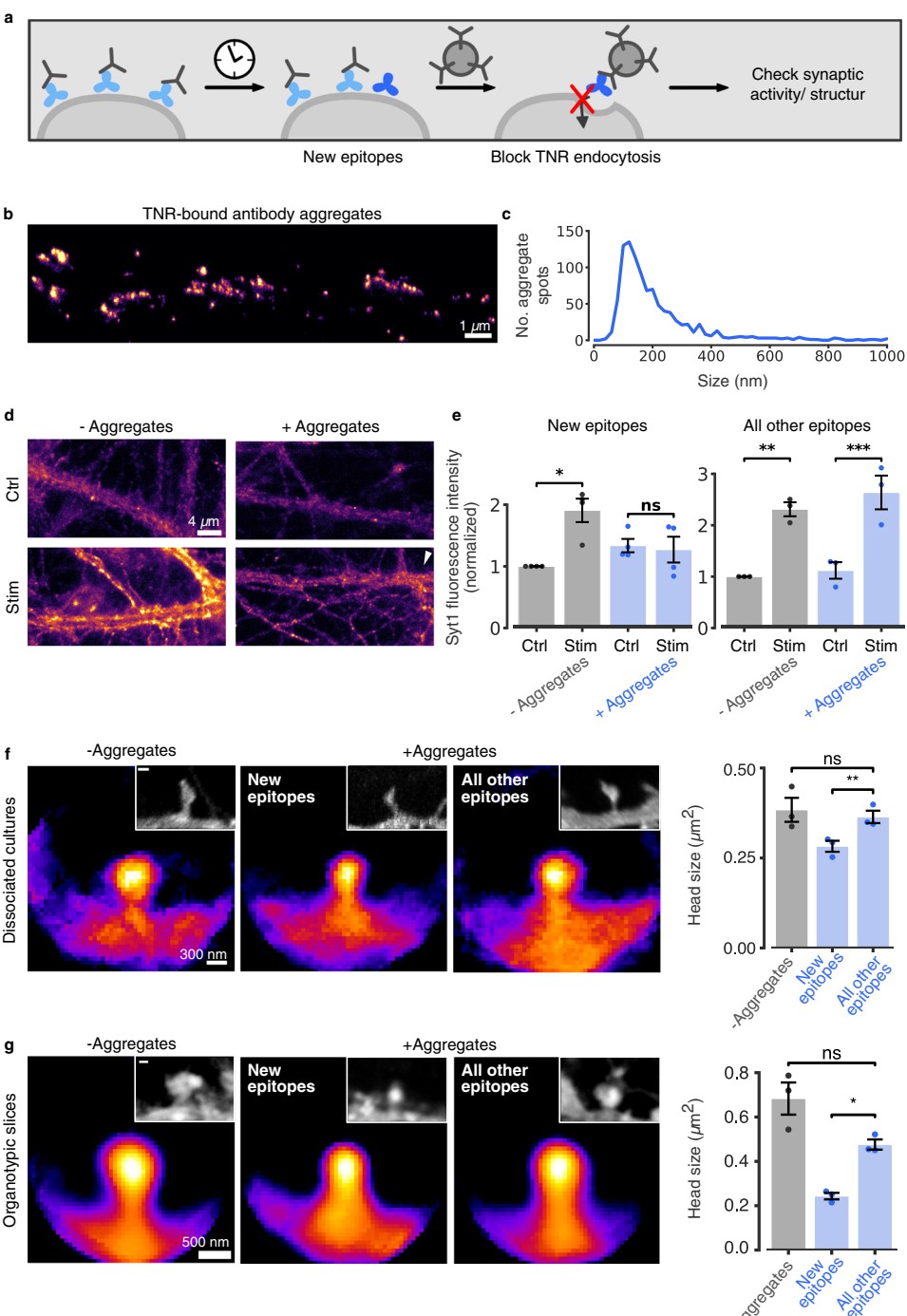

Synaptic Systems, Göttingen, Germany) diluted 1:100 in their own cell media, for 2 h. The neurons were subsequently washed in Tyrode's solution (124 mM NaCl, 30 mM glucose, 25 mM HEPES, 5 mM KCl, 2 mM CaCl$_2$, 1 mM MgCl$_2$, pH 7.4) and returned to their original conditioned media. For labeling of newly-emerged TNR epitopes, neurons were incubated with antibodies conjugated to the fluorescent dye Atto647N, Atto550 or STAR580 (custom-made; clone 619; Synaptic Systems, Göttingen, Germany) diluted 1:500 in their own cell media, for 1 h. The labeling was performed after either 0, 2, 4, 6 or 12 h post-blocking. After labeling, the neurons were fixed either immediately, or after a further incubation of 6 or 12 h, in their original culture media allowing for epitope internalization. In some experiments, the surface-bound antibodies were stripped at the end of the blocking-labeling assay. These details are denoted in the figure legends. For the 2-color STED experiment comparing old and newly-emerged TNR epitopes (Supplementary Fig. 17), the blocking step was performed with STAR580-conjugated TNR antibodies. As a control for unspecific uptake or binding of antibodies, neurons were incubated with 1:500 Atto647N-conjugated goat anti-mouse IgG (#610-156-121 S; Rockland, USA).

*Experiments with TNR Fab fragments.* The blocking-labeling assay using Fab fragments (Fig. 5a, b and Supplementary Figs. 14 and 18b), surface TNR epitopes were blocked with 10 µg/mL (cultured neurons) or 50 µg/mL (organotypic slices) unlabeled Fab fragments directed against TNR (custom-made from the same TNR antibody used in the rest of the work; Synaptic Systems, Göttingen, Germany), applied together with 1 mg/mL unlabeled FluoTag-X2 anti-mouse secondary nanobody (custom-made; NanoTag, Göttingen, Germany), diluted in their own cell media for 2 h, and newly-emerged TNR epitopes were labeled with 2 µg/mL Fab fragments directed against TNR, applied together with 1:500 FluoTag-X2 STAR635P or STAR580-conjugated anti-mouse secondary nanobodies (#N1202-Ab635P, #N1202-Ab580; NanoTag, Göttingen, Germany).

For the experiment described in Supplementary Fig. 14c, d, newly-emerged epitopes were labeled 12 h post-blocking. The neurons were mounted for live imaging 4 h after labeling, and the Fab fragments bound to surface TNR molecules were stripped by incubation with proteinase K (see 'Surface antibody stripping with proteinase K'). For the experiment described in Supplementary Fig. 14e, newly-emerged TNR epitopes were labeled 4 h post-blocking. The neurons were mounted

**Fig. 9 Perturbing the recycling TNR pool modulates synaptic function. a** Assay to perturb TNR recycling: newly-emerged TNR epitopes were labeled 12 h post-blocking with biotinylated antibodies, and bound to large aggregates of antibodies. As control, all other epitopes (non-recycling) were labeled. **b** STED images of aggregates. Scale bar = 1 μm. **c** Histogram of aggregate size (FWHM). N = 4 independent experiments, 995 aggregates. **d** Neurons were incubated with aggregates for 30 min. Synaptic activity was assessed by uptake of Syt1 antibodies (as in Fig. 3). Without stimulation, Syt1 antibodies detect the surface vesicle population (40–50% of actively-recycling vesicles[20]). Stimulation results in signal increase (exo-/endocytosis of new vesicles) in controls, but not in aggregate-treated cultures (epifluorescence). Scale bar = 4 μm. **e** Quantification of Syt1 fluorescence intensity confirms this observation and indicates that tagging all other epitopes has no effects. N = 4 ('new epitopes')/3 ('all other') independent experiments, ≥15 neurons per datapoint. Repeated-measures ANOVA on rank ('new epitopes': $F_{1,6}$ = 12.54, *p = 0.012; 'all other epitopes': $F_{1,4}$ = 1.5, p = 0.288) for the interaction Stim/ctrl x +/− Aggregates), followed by Sidak's multiple comparisons test ('new epitopes': *p = 0.02, p = 0.419; 'all other epitopes': **p = 0.002, ***p < 0.001 for 'stim' vs. 'ctrl' for untreated and treated neurons, respectively). **f, g** Effect of recycling perturbation on synapse structure. Dissociated cultures (**f**) and organotypic hippocampal slices (**g**) were treated with aggregates for 12 h. Plasma membranes were visualized with DiO (**f**) or by infection with AAV9-Syn-eGFP (**g**), in averaged spines or individual examples (insets). Scale bar = 300 nm (**f**), 500 nm (**g**). N = 3 independent experiments, >80 (**f**), >60 (**g**) synapses per condition. One-way ANOVA (**f**: $F_{2, 6}$ = 5.269, *p = 0.05) or repeated-measures one-way ANOVA (**g**: $F_{1.041, 2.083}$ = 20.76, *p = 0.042), followed by Fisher's LSD (**f**:** p = 0.005, p = 0.418; **g**: *p = 0.025, p = 0.16), to compare 'all other epitopes'/'new epitopes' and 'all other epitopes'/'Tyrode', respectively. Data represent mean ± SEM, dots indicate individual experiments (**d**–**g**). Source data are provided in Source Data file.

for live imaging 1, 2 or 3 days after labeling and the Fab fragments bound to surface TNR molecules were stripped by incubation with proteinase K. For the experiment described in Fig. 5a, b, newly-emerged TNR epitopes were tagged with unlabeled Fab fragments directed against TNR 4 h later post-blocking. The tagged epitopes that remained exposed on the surface 12 h later were blocked by an additional incubation with 1 mg/mL unlabeled FluoTag-X2 anti-mouse nanobodies for 2 h. Subsequently, the remaining Fab-tagged epitopes that had internalized were revealed at the surface with 1:500 STAR635P-conjugated anti-mouse secondary nanobodies immediately after the second blocking step, or following an additional incubation of 1–3 days. To visualize the active synaptic vesicle pool, neurons were incubated with 1:500 polyclonal rabbit antibodies directed against the lumenal domain of Syt1 conjugated to the fluorescent dye Oyster488 (#105 103C2; Synaptic Systems, Göttingen, Germany) during the TNR labeling step.

*Additional ECM molecules and integrins.* For the experiments with alternative ECM components, surface epitopes were blocked with 1:100 Wisteria floribunda agglutinin WFA (#L8258; Merck, Germany), 1:50 Hyaluronan binding protein HABP (#H0161; Merck, Germany) or 1:100 mouse anti-Neurocan (#N0913; clone 650.24; Merck, Germany) together with 1 mg/mL unlabeled FluoTag-X2 secondary anti-mouse nanobodies (custom-made; NanoTag, Göttingen, Germany). Labeling was performed with 1:500 biotinylated WFA or HABP followed by 1:500 streptavidin-Atto647N (#AD 647-61; ATTO-TEC GmbH, Germany), or 1:500 anti-Neurocan and 1:500 FluoTag-X2 anti-mouse secondary nanobodies conjugated to STAR635P (#N1202-Ab635P; NanoTag, Göttingen, Germany). For experiments with β1-integrin (Fig. 7), its labeling was performed concurrently with TNR labeling (4 or 12 h after the initial TNR surface epitope blocking, as denoted in the figure legend), with 1:250 FITC hamster anti-CD29 (#561796; clone Ha2/5; BD Biosciences, CA, USA) from a 0.5 mg/mL stock solution. The neurons were fixed immediately or after a further incubation of 12 h, followed by surface antibody stripping with proteinase K (see 'Surface antibody stripping with proteinase K'). For the blocking of β1-integrins, the neurons were treated with 1:25 hamster anti-CD29 (#555003; clone Ha2/5; BD Biosciences, CA, USA) immediately after labeling the newly-emerged TNR epitopes, and remained for an additional 6 h.

*Live labeling of acidic organelles.* To label acidic organelles, LysoTracker™ Green DND-26 (#L7526, Thermofisher Scientific, USA) was added throughout the labeling of the newly-emerged TNR epitopes, at a concentration of 75 nm.

*Drug treatments.* Unless otherwise specifed, the drug applications in these experiments began after the TNR blocking step and lasted until fixation. To enhance culture activity, the neurons/slices were treated with 40 μM bicuculline (#485-49-4; Merck, Germany) or 0.1% volume/volume DMSO (#67-68-5; Merck, Germany) as a control. To reduce culture activity by inhibiting AMPA and NMDA receptors, neurons were treated with 10 μM CNQX (#0190; Tocris Bioscience, Germany) and 50 μM AP5 (#0106; Tocris Bioscience, Germany). To block the activity of matrix metalloproteinases, neurons were treated with 10 μM GM6001 (#CC1010, Merck, Germany). To digest glycosaminoglycans, neurons were treated with 0.5 units/mL Chondroitinase ABC from *Proteus vulgaris* (#C3667, Merck, Germany) for 30 min following the blocking step. To perturb dynamin-dependent endocytosis, neurons were treated with 30 μM Dyngo® 4a (#ab120689; Abcam, United Kingdom) for 2 h following the labeling step (for the experiment shown in Supplementary Fig. 15a) or throughout the experiment (for the experiment shown in Supplementary Fig. 15c). To perturb Golgi trafficking, neurons were treated with 5 μg/mL brefeldin (#B7651; Merck, Germany) or 1 μM monensin (#M5273; Merck, Germany) for 4 h, added from the onset of blocking.

*Live labeling of synaptotagmin 1 and EGF receptors.* For the surface antibody stripping experiments of with Syt1 and EGF, neurons were incubated with 1:100

monoclonal mouse antibodies directed against the lumenal domain of Syt1 conjugated to Atto647N (custom made, Synaptic Systems, Göttingen, Germany), or 1:100 epidermal growth factor (EGF), complexed to Alexa Fluor® 647 (#E35351; Thermo Fisher Scientific, USA). The incubations were performed for 5 min at 4 °C. The neurons were imaged before and after stripping with proteinase K (see 'Surface antibody stripping incubation with proteinase K') immediately after labeling, following a short incubation of 15 min, or after a longer incubation of 60 min (for Syt1) or 4 h (for EGF).

Antibodies were diluted from 1 mg/ml stocks, unless specified otherwise. Live-cell incubations were performed at 37 °C, and live washing steps were performed in pre-warmed Tyrode.

**Surface antibody stripping with proteinase K**. For surface antibody stripping, neurons were incubated with 8 units/ml Proteinase K from *Tritirachium album* (#P2308, Merck, Germany) in Tyrode for 5 min at room temperature. The neurons were then washed and mounted for live imaging or immediately fixed and post-immunostained, as described in the figure legends.

**Fixation and post-fixation immunostaining**. Neurons/slices were fixed in 4% PFA in PBS (137 mM NaCl, 10 mM $Na_2HPO_4$, 2 mM $KH_2PO_4$, 2.7 mM KCl, pH 7.4) for 20 min on ice followed by 20 min at room temperature. The fixation reaction was quenched with 100 mM $NH_4Cl$ in PBS for 30 min. For subsequent immunostainings, neurons were permeabilized and blocked with PBS containing 2.5%, bovine serum albumin (BSA) (#A1391-0250; Applichem, Germany) and 0.1% Tween20 (#9005-64-5; Merck, Germany) or Triton X (#9005-64-5, Merck, Germany) for 1 h. In addition to the labels mentioned under the corresponding experiments, the following labels were used for post-fixation immunostainings. Primary antibodies and labels: to identify excitatory glutamatergic synapses and visualize the synaptic vesicle pool: FluoTag-X2 anti-VGlut1 nanobodies directly conjugated to STAR580 (#N1602; NanoTag, Göttingen, Germany); to identify inhibitory synapses: 1:200 rabbit polyclonal anti-VGAT (#131 103; Synaptic Systems, Göttingen, Germany); to identify perineuronal nets: 1:500 biotinylated *Wisteria floribunda* agglutinin WFA (#L8258; Merck, Germany); to identify the neuronal axons (for the experiment described in Fig. 4b): 1:100 mouse monoclonal directed against Ankyrin G (#75-146; NeuroMab, USA); to identify organelles: 1:100 rabbit polyclonal anti-LAMP1 (#ab24170; Abcam, United Kingdom), 1:200 rabbit monoclonal anti-Rab5 (#C8B1; Cell Signaling, Germany), 1:100 rabbit monoclonal anti-Rab7 (#9367; Cell Signaling, Germany), 1:100 rabbit polyclonal anti-TGN38 (#T9826; Merck, Germany), 1:100 rabbit polyclonal anti-calreticulin (#12238 S; Cell Signaling, Germany), 1:100 rabbit polyclonal anti-Caveolin1 (#ab2910; Abcam, United Kingdom), rabbit polyclonal anti-Rab11a (#2413; Cell Signaling, Germany) and 1:100 rabbit polyclonal anti-Rab11b (#ab3612; Abcam, United Kingdom); to identify neurons: 1:100 mouse monoclonal or guinea pig polyclonal anti-NeuN (#266 011 and #266 004; Synaptic Systems, Göttingen, Germany); to identify astrocytes: 1:1000 rabbit polyclonal anti-GFAP (#173 002; Synaptic Systems, Göttingen, Germany); to identify oligodendrocytes: rabbit monoclonal anti-myelin basic protein (#78896; Cell Signaling, Germany), to identify microglia: 1:500 guinea pig polyclonal anti-Iba1 (#234 004; Synaptic Systems, Göttingen, Germany). Secondary antibodies and labels were used,: 1:200 Cy3-conjugated goat anti-mouse IgG (#115-035-146; Dianova, Germany); 1:500 Atto647N-conjugated goat anti-mouse IgG (#610-156-121 S; Rockland, USA); 1:500 STAR580-conjugated FluoTag-X2 anti-mouse secondary nanobodies (#N1202-Ab580; NanoTag, Göttingen, Germany); 1:500 STAR580-conjugated goat anti-mouse IgG (#ST580-1001; Abberior GmbH, Göttingen, Germany); 1:200 Cy3-conjugated goat anti-rabbit IgG (#111-165-144; Dianova, Germany); 1:200 Cy5-conjugated goat anti-rabbit IgG (#111-175-144; Dianova, Germany); 1:200 STAR580-conjugated goat anti-rabbit IgG (#ST580-1002; Abberior GmbH, Göttingen, Germany); 1:200 Cy3-conjugated goat a anti-guineapig IgG (#706-165-148; Dianova, Germany); 1:200 STAR635P-conjugated streptavidin (#ST635P-0120;

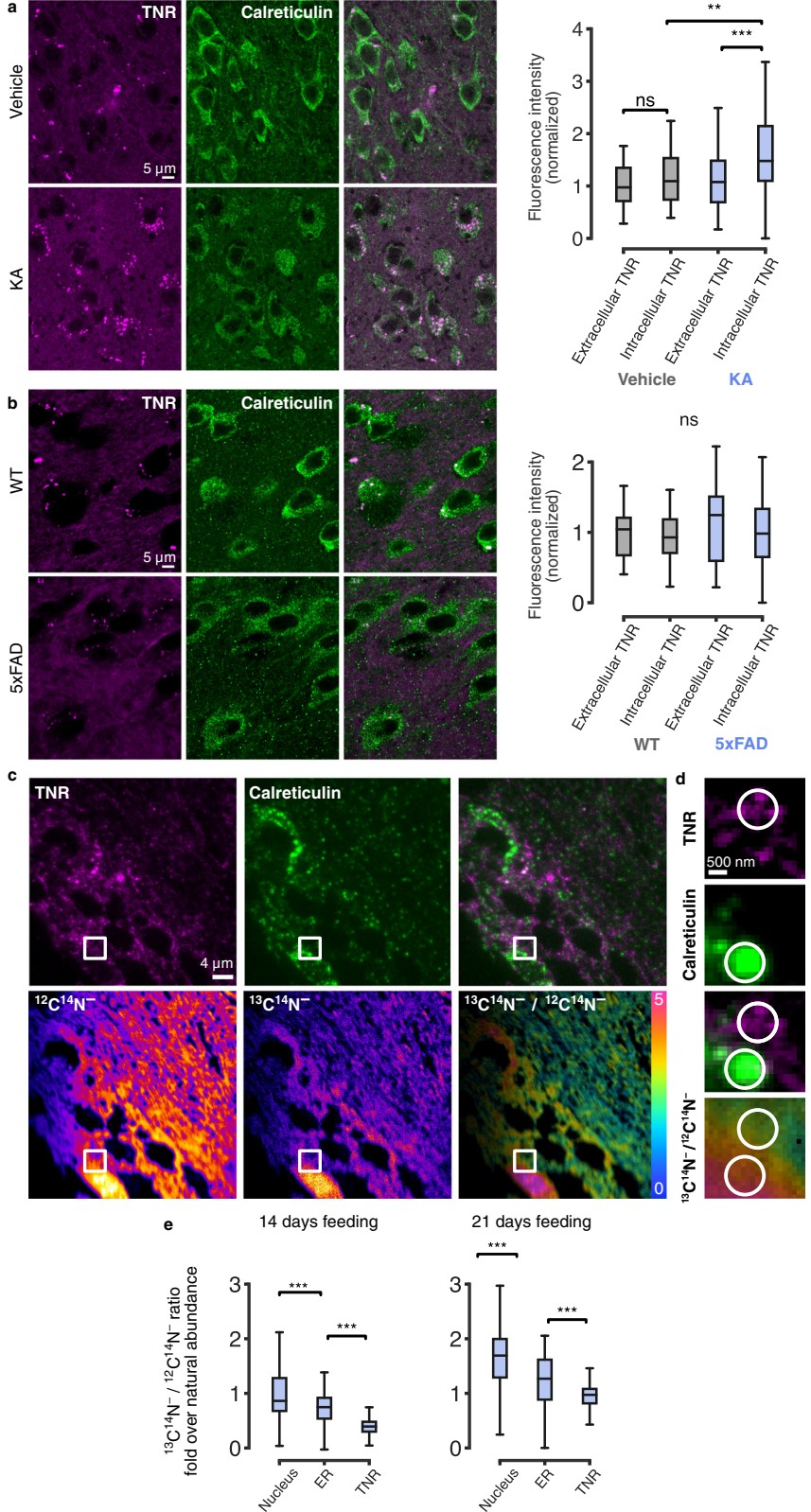

Abberior GmbH, Göttingen, Germany); 1:200 Cy3-conjugated streptavidin (#016-160-084; Dianova, Germany). Primary and secondary incubations were performed for 1 h at room temperature, also in blocking solution. Percentages represent volume/volume. Antibodies were diluted from 1 mg/ml stocks, unless specified otherwise. For visualizing neuronal membranes, fixed and immunostained cover-slips were incubated with DiO (#D275; Molecular probes, Thermofisher, USA). In brief, DiO crystals were diluted 20 µg/mL in PBS and sonicated for 30 min, and then diluted further to 2 µg/mL. Neurons were incubated with DiO for 20 min at

37 °C, washed once and left overnight. Neurons were subsequently washed twice and embedded in Mowiol (Calbiochem, Billerica, MA, USA). To label nuclei, neurons were incubated with Hoechst 33342 (#62249, Thermofisher Scientific, USA) for 10 min before mounting.

**4 day-long live imaging of TNR-labeled neuronal cultures.** Neurons were plated on 24 well glass-bottomed cell culture plates (#P24-1.5H-N, Cellvis, USA) at a

**Fig. 10 TNR dynamics are observed in brain slices from adult mice, and are altered in an epilepsy model. a, b** Intracellular TNR in disease models. Hippocampal slices from kainic acid (KA)-induced epilepsy model mice and 5xFAD familial Alzheimer's disease model mice were immunostained for TNR and the ER marker calreticulin, to enable identification of intracellular (somatic) TNR. All other TNR was presumed extracellular. **a** Imaged regions (confocal) from mice pre-treated with vehicle or KA. The proportion of intracellular TNR is increased in KA-treated mice. $N = 3$ mice per treatment, 60 (vehicle) and 67 (KA) regions analyzed. Kruskal-Wallis ($H_3 = 27.93$, ***$p < 0.001$), followed by two-sided Dunn's multiple comparisons test ('vehicle; extracellular TNR'/'vehicle; intracellular TNR': '$p = 0.932$; 'KA; extracellular TNR'/'KA; intracellular TNR': ***$p < 0.001$; 'vehicle; intracellular TNR'/'KA; intracellular TNR': **$p = 0.004$). **b** Similar analysis for 5xFAD mice. No significant differences are observed. $N = 3$ mice per treatment, 68 (WT) and 29 (5xFAD) regions analyzed. Kruskal-Wallis ($H_3 = 3.233$, $p = 0.357$). Scale bar = 5 μm. **c–e** Isotopic imaging in adult mice suggests intracellular TNR is not newly synthesized. TNR turnover in vivo was measured with correlative fluorescence and isotopic imaging (COIN[20,48,72,80,81]) in brain slices of mice pulsed with isotopically stable $^{13}C_6$-lysine for 14 or 21 days (previously characterized in[2]). **c** Top: section stained for TNR and calreticulin (epifluorescence). Bottom: nanoSIMS images of $^{12}C^{14}N^-$ (left) and $^{13}C^{14}N^-$ (middle) secondary ions. The $^{13}C^{14}N^-/^{12}C^{14}N^-$ ratio image (right) indicates the enrichment of $^{13}C$. Scale bar = 4 μm. **d** Zoom of square regions in **c**. Scale bar = 500 nm. **e** Quantification of $^{13}C^{14}N^-/^{12}C^{14}N$ ratio as fold over the natural abundance level. TNR-enriched areas exhibit the lowest $^{13}C$ enrichment in these cells (lowest newly synthesized protein levels). $N = 6$ sections from 3 mice per condition. Kruskal-Wallis ($H_2 = 167.2$, ***$p < 0.001$), followed by two-sided Dunn's multiple comparisons test (***$p < 0.001$ for all comparisons). For all panels: boxes show median (mid-line) and quartiles, whiskers show minimum/maximum values. Outliers were omitted according to inter-quartile range (IQR) proximity (exceeding 1.5*IQR). Source data provided in Source Data file.

density of ~50,000 cells per well as described (see 'Preparation of rat dissociated hippocampal cultures'). At DIV14, all or newly-emerged TNR molecules were labeled with Atto550-conjugated TNR antibodies, as described above (see 'Blocking-labeling assay and live treatments'), and in the figure legend. The neurons were transferred to an automated live-cell incubator/imaging system (BioSpa™ 8 Automated Incubator coupled with a Cytation™ 5 Cell Imaging Multi-Mode Reader, BioTek, USA). The plates were incubated in the BioSpa at 37 °C and 5% CO₂ for 4 days. Every 4 h, the plates were automatically transferred to the Cytation 5, also set to 37 °C and 5% CO₂, and imaged using a 20x Plan Fluorite, 0.45 NA (#1320517, BioTek PN) objective in the RFP imaging channel, in addition to a phase-contrast channel overlay. For each well, 16 fields of view were acquired (a total $6.3 \times 4.7$ mm imaging area per well).

**TNR/brevican KO mouse brain slices**. TNR/brevican KO mice were first deeply anesthetized by ketamine (90 mg/kg of body weight) and xylazine (18 mg/kg of body weight) in a 0.9% NaCl solution and perfused transcardially with 4% paraformaldehyde (PFA) in 0.1 M PBS (PH 7.2) for 10 min. The dissected brains were incubated in 4% PFA containing PBS at 4 °C for 2 h. Fifty-micrometer thin sagittal sections were cut using a microslicer (HM650V, MICROM). The sections were first washed with PBS and then permeabilized with 0.1% Triton X-100 (#T9284, Merck, Germany-Aldrich) in PBS for 10 min at room temperature. Next, the sections were incubated for 1 h (at room temperature with gentle shaking) in a blocking solution containing 10% normal goat serum (16210064, Life Technologies), 0.4% Triton X and 0.1% glycine in PBS. Then, the sections were incubated for 24 h (at room temperature with gentle shaking) with a mixture of 1:500 *Wisteria floribunda* agglutinin (WFA) (#B-1355; Vector Laboratories, California, USA) and 1:200 monoclonal mouse anti-TNR (#217 011, Synaptic Systems, Göttingen, Germany) for 24 h. The sections were then washed in PBS and incubated with 1:200 streptavidin Alexa Fluor® 405 (#S32351; Life Technologies, Thermofisher Scientific, USA) and 1:200 goat anti-mouse Alexa Fluor® 647 (#A21236; Life Technologies, Thermofisher Scientific, USA), on a shaker for 3 h at room temperature. Following the incubation, the sections were washed with PBS and then mounted on Superfrost glass slides (#J1800AMNZ, Thermofisher Scientific, USA) with Fluoromount medium (#F4680, Merck, Germany). Images were acquired with confocal laser-scanning microscopy (LSM 700, Zeiss) with the same acquisition parameters for the comparison of samples.

**shRNA-mediated TNR knockdown**

*Generation of shRNA viral vectors*. The shRNA plasmid to knockdown mouse TNR (GeneID:21960) was cloned by the insertion of the siRNA's sequence (siRNA ID: SASI_Mm01_00073137, Rosetta Predictions from Merck, Germany Aldrich, Merck) targeting the open reading frame of mouse Tenascin-R into adeno-associated viral (AAV) vector U6 GFP (Cell Biolabs Inc., San Diego, CA 92126, USA), using BamH1 and EcoR1 restriction sites[60]. A scrambled siRNA sequence was used as a non-targeting negative control. The sequences for TNR siRNA and control siRNA are 5′-gcgttcactctcctccctg, and 5′-cggctgaaacaagagttgg, respectively. Clones were verified by sequencing analysis and used for the production of adeno-associated particles as described previously[61]. Briefly, HEK 293 T cells (Thermo Scientificl #HCL4517) were transfected with the calcium phosphate method with an equimolar mixture of the expression plasmid, pHelper plasmid, and Rep/Cap plasmid pAAV/DJ (for the production of the shRNA, pAAV2/1 and pAAV2/2 were combined to obtain a mixed capsid of AAV1/2). After 48 h of transfection, cells were lysed using freeze-thaw and treated with benzonase at a final concentration of 50 units/ml for 1 h at 37 °C. The lysate was centrifuged at 8000 g at 4 °C. The supernatant was collected and filtered with a 0.2-micron filter. Filtered supernatant was passed through pre-equilibrated Hitrap Heparin columns (#17-0406-01; Ge HealthCare Life science, USA), followed by a wash with Wash Buffer 1

(20 mM Tris, 100 mM NaCl, pH 8.0; filtered sterile). Columns were additionally washed with Wash Buffer 2 (20 mM Tris 250 mM NaCl, pH 8.0; filtered sterile). Viral particles were eluted with elution buffer (20 mM Tris 500 mM NaCl, pH 8.0; filtered sterile). Amicon ultra-4 centrifugal filters with a 100,000 molecular weight cutoff were used to exchange the elution buffer with sterile PBS. Finally, viral particles were filtered through 0.22 μm syringe filters (Nalgene®; # Z741696; Merck, Germany), aliquoted and stored at −80 °C until usage.

*Culture preparation and infection with shRNA*. Mouse hippocampal cultures were prepared from newborn C57BL/6 N mice as previously described[62] with minor modifications. For the glial feeder layer, the cortices were dissected in HBSS, and enzymatically digested for 15 min at 37 °C with 0.05% (w/v) trypsin/EDTA solution (Gibco, Thermo Fisher Scientific, USA). The dissected cortices were then triturated and the glia were cultured for 7 days in T-75 flasks in DMEM containing 10% FBS and penicillin/streptomycin (100 U/ml; 100 mg/ml) mixture and 0.1% MITO + serum extender (Biozym Scientific, Germany). After 7 days, the cells were seeded onto 18 mm coverslips at a density of (50,000/cm³), and 1 μM FUDR (Merck, Germany) was added to the medium 6 days later. For the neuronal culture, hippocampi were dissected in HBSS, and digested for 1 h at 37 °C in DMEM supplemented with 25 U/ml papain (Worthington Biomedical Corp., USA), 0.2 mg/ml cysteine (Merck, Germany), 1 mM CaCl₂, and 0.5 mM EDTA. The digestion was arrested by a 15-min incubation in DMEM supplemented with 2.5 mg/ml bovine serum albumin (Merck, Germany), 2.5 mg/ml trypsin inhibitor (Merck, Germany), and 10% (v/v) FBS. Neurons were triturated and seeded onto the astrocyte feeder layers at a density of ~ 75,000/cm³. Prior to the addition of the hippocampal neurons, the culture medium was switched to Neurobasal-A medium (Life Technologies, Carlsbad, CA, USA) containing B27 supplement (Gibco, Thermo Fisher Scientific, USA), 2 mM GlutaMax (Gibco, Thermo Fisher Scientific, USA) and penicillin/streptomycin (100 U/ml; 100 mg/ml) mixture (Biozym Scientific, Germany). At DIV7, the neurons were infected with AAV1/2-GFP-U6-shTNR (shRNA for TNR) or AAV1/2-GFP-U6-shScr (scramble shRNA) (1 μl/ml with a titer of $3.35 \times 10^{10}$). At DIV14, infected neurons were washed extensively in PBS and fixed by immersion in a solution of 4% PFA in 0.1 M PBS.

*Immunostaining*. The neurons were blocked with PBS containing 2.5% bovine serum albumin (BSA; #A1391-0250; Applichem, Germany) and 0.1% Triton X (#9005-64-5, Merck, Germany) for 1 h at room temperature. Then, the neurons were incubated with 1:500 Atto647N-conjugated mouse monoclonal anti-TNR (custom made, Synaptic Systems, Göttingen, Germany), overnight at 4 °C. The neurons were washed in PBS and then mounted in Mowiol.

**Electrophysiology**. Na⁺, K⁺ currents and mEPSCs were recorded by conventional whole-cell patch-clamping under a physiological temperature of 37 °C, with continuous perfusion of ACSF (pre-saturated with 95% O₂ and 5% CO₂). ACSF (artificial cerebrospinal fluid) contains (in mmol/L): 119 NaCl, 2.5 KCl, 2 CaCl₂, 2 MgSO₄, 1.25 NaH₂PO₄, 26.2 NaHCO₃, 11 Glucose, with pH ~7.4 and osmolarity 305-315 mOsm. Patch pipettes (2–4MΩ) were pulled from borosilicate glass (1.6 mm outside diameter) and filled with intracellular solution containing (in mmol/L): 130 HMeSO₄, 5 KCl, 1 EGTA, 10 HEPES, 4 MgCl₂, 2 ATP-Na₂, 0.4 GTP-Na, 7 Phosphocreatine-Na₂, with pH adjusted to 7.2–7.4 by KOH, and osmolarity adjusted to 290–295 mOsm. Final concentration (in mmol/L): K⁺: 135; Na⁺: 17.4; Cl⁻: 9. Whole-cell configuration was first made under normal ACSF. Na⁺ and K⁺ currents were elicited by a series of depolarizing pulses between −60 mV and +50 mV, in 10 mV increments, from a holding potential of −70 mV. Miniature EPSCs were recorded at a holding potential of −70 mV under ACSF, with 1 μM TTX and 100 μM picrotoxin, after Na⁺ currents have disappeared. An EPC-9 patch-clamp amplifier equipped with Patchmaster software (HEKA Electronics,

Germany) was used for data acquisition. No leak currents compensation was used, but fast and slow capacitances[63,64], and series resistances were corrected online. The data were sampled at 40 kHz and filtered at 10 kHz (four-pole Bessel) and 5.9 kHz (three-pole Bessel). Data were stored and exported to Matlab for further analysis, which was performed by thresholding the curves using an empirically derived threshold, to detect the mEPSCs. Control cultures were exposed to boiled TNR antibodies, while test cultures were incubated with normal TNR antibodies (#217 011; Synaptic Systems, Göttingen, Germany).

**TNR 'blocking-labeling' experiment in rat dissociated hippocampal 'sandwich' cultures**

*Culture preparation.* Dissociated primary hippocampal sandwich cultures, in which the neurons are physically separated from a glial feeding layer, were prepared from E18 Wistar rats as previously described[26], and maintained in N2 medium. The following modifications were made to the original protocol. For the feeding layer, the glia were seeded 3 days before the dissection of the E18 rats in 12-well plates, at a density of ~ 10,000 cells per well. The neurons were then seeded on circular glass coverslips (18 mm diameter) at a density of ~ 30,000 cells per coverslip. Paraffin dots were added to the coverslips as the spacers between the neurons and the glia. The blocking-labeling assay in sandwich cultures was performed as for the regular neuronal cultures (see 'Blocking-labeling assay and live treatments', above).

**Application of recombinant His-tagged TNR to neuronal cultures/slices**

*Neuronal cultures.* For the experiment shown in Fig. 5c–e, the neurons were incubated with 5 μg/mL recombinant TNR (# 3865-TR; Biotechne GmbH, Germany) diluted in their own cell media, for 1 h at 37 °C. The neurons were then washed in Tyrode's solution (124 mM NaCl, 30 mM glucose, 25 mM HEPES, 5 mM KCl, 2 mM CaCl$_2$, 1 mM MgCl$_2$, pH 7.4) and returned to their original conditioned media. Immediately after, 1 or 3 days later the neurons were stripped with proteinase K (see 'Surface antibody stripping with proteinase K') and then fixed with 4% PFA in PBS (or immediately fixed, without stripping), as described in detail in the figure legend. The fixation and subsequent immunostaining were performed as in other experiments (see 'Fixation and post-fixation immunostaining'), using 1:250 rabbit monoclonal anti-His-Tag (#12698; Cell Signaling, Germany), followed by Cy3-conjugated goat anti-mouse IgG (#111-165-144; Dianova, Germany).

To assess the extent of rTNR degradation in neuronal cultures, they were incubated with the protein (5 μg/mL) for 2 h, 3 days and 6 days. Samples were then collected and lysed in SDS sample buffer, and were later boiled at 95 °C for 5 min, before SDS-PAGE procedures. To ensure even loading of material, we used as an internal control a monoclonal mouse anti-calmodulin (#MA3-917; Thermo Fisher Scientific, US) antibody that was also added to the cultures, and whose levels reflected the general protein concentrations in the cultures. The samples were loaded on 10% SDS polyacrylamide gels and were then transferred to nitrocellulose membranes. The membranes were blocked for 1 h at room temperature in blocking buffer consisting of PBS with 0.1% Tween-20 and 5% skimmed milk powder, and then incubated with 1:1000 rabbit monoclonal anti-His-Tag for 1 h at room temperature. After washing, the membranes were incubated with IRDye® 800CW donkey anti-mouse IgG and IRDye® 680RD donkey anti-rabbit IgG (#926-32212 and #926-68073; LI-COR Biotechnology, USA), for 1 h at room temperature. The primary and secondary incubations were performed in blocking solution. The membranes were then washed in PBS and scanned using Odyssey CLx (LI-COR).

*Organotypic slices.* The slices were incubated with 5 μg/mL recombinant TNR diluted in their own cell media, for 2 h at 37 °C. The slices were then washed in Tyrode's solution and returned to their original conditioned media. The slices were incubated a further 6 h to allow for internalization, and then stripped with proteinase K and then fixed with 4% PFA, as described in detail in the figure legend. Stimulated and control slices were incubated with 40 μM bicuculline (#485-49-4; Merck, Germany) or 0.1% volume/volume DMSO (#67-68-5; Merck, Germany) added throughout the experiment. As additional controls, some of the slices were not incubated with the recombinant TNR (to measure background signal) or not stripped with proteinase K (to visualize the full population of molecules). The fixation and post-fixation immunostaining were performed as described above for the neuronal cultures.

*Acute slices.* The hippocampi of 3 week-old Wistar rats were extracted and cut into ~700 μm-thick transverse slices. The slices were placed in artificial cerebrospinal fluid at 37 °C (aCSF: 125 mM NaCl, 2.5 mM KCl, 1.25 mM NaHPO$_4$, 25 mM NaHCO$_3$, 25 mM D-Glucose, 2 mM CaCl$_2$ and 1 mM MgCl$_2$, bubbled with carbogen for 30 min). After a short period of recovery, 5 μg/mL recombinant was pipetted directly into the aCSF and the slices were incubated for 2 h. The slices were then incubated with proteinase K (see 'Surface antibody stripping with proteinase K') for a further 30 min, to remove any protein that remained at the surface. Stimulated and control slices were incubated with 40 μM bicuculline or 0.1% volume/volume DMSO added throughout the experiment. As additional controls, some of the slices were not incubated with the recombinant TNR (to measure background signal) or not stripped with proteinase K (to visualize the full population of molecules). After washing, the slices were fixed with 4% PFA in PBS for 1 h at room temperature, and the fixation reaction was quenched with 100 mM

NH$_4$Cl in PBS for 30 min. The slices were then blocked in 1 h with PBS containing 2.5% bovine serum albumin and 0.3% Triton X for 1 h at room temperature, and then incubated with 1:250 rabbit monoclonal anti-His-Tag and FluoTag-X2 anti-VGlut1 nanobodies directly conjugated to Atto 488 (#N1602; NanoTag, Göttingen, Germany), overnight at 4 °C. The slices were washed, and then incubated with 1:200 Cy3-conjugated goat anti-mouse IgG (#115-035-146; Dianova, Germany) for 2 h, at room temperature. All antibody incubation steps were performed in blocking solution. The slices were washed in PBS and embedded in Mowiol.

**siRNA-mediated dynamin knockdown**

*siRNA transfection.* The neurons were co-transfected with 50 nM dynamin 1, 2, and 3 siRNA constructs (previously described in[65]) or with non-targeting control siRNA (ON-TARGETplus Non-targeting Control Pool; # D-001810-10, Horizon Discovery) at DIV 7, using Lipofectamine RNAiMAX transfection reagent (#13778030; Thermo Fisher Scientific, USA). At DIV 14, labeling of newly-emerged TNR epitopes was performed as in the other experiments (see 'Blocking-labeling assay and live treatments'). Fixation and immunostaining were performed as in other experiments (see 'Fixation and post-fixation immunostaining'), using 1:100 rabbit anti-dynamin 1, 2, 3 (#115 002, Synaptic Systems, Göttingen, Germany), followed by 1:200 Cy3-conjugated goat anti-rabbit IgG (#111-165-144; Dianova, Germany).

**Co-immunoprecipitation from rat synaptosomes and Western blotting.** Synaptosomes were prepared from 5- to 6-week-old rats as previously described[66]. In brief, brain homogenates were were centrifuged at 5000 rpm for 2 min. The resulting supernatants were then centrifuged at 11,000 rpm for 12 min. The pellets were resuspended and then loaded on Ficoll gradients (6, 9 and 13% w/v), and centrifuged at 22,500 rpm for 35 min. The synaptosomes (enriched between 9-13%) were collected. Precipitation of TNR was performed by incubating the synaptosomes with the same antibodies used in the rest of the study (#217 011; Synaptic Systems, Göttingen, Germany) for 1 h at 4 °C, rotating. The synaptosomes were then incubated with 4% CHAPS (1 h at 4 °C, rotating), to permeabilize the membranes (as previously described[67]). The supernatants were incubated with Dynabeads™ Protein G (#10003D; Thermofisher Scientific, USA) prewashed using PBS with 0.01% Tween-20 (#9005-64-5; Merck, Germany)), for 2 h at 4 °C, rotating. The beads were then washed 3 times again, and the samples were eluted using SDS-sample buffer and boiled at 95 °C for 5 min. Equal amounts (~15% of the starting synaptosome homogenate) were loaded on 10% SDS polyacrylamide gels, alongside 0.05% of synaptosome homogenate. The gels were transferred to nitrocellulose membranes, which were blocked for 1 h at room temperature, in blocking buffer consisting of PBS with 0.1% Tween-20 and 5% skimmed milk powder. The membranes were then incubated with 1:1000 mouse monoclonal anti-TNR antibodies or 1:1000 rabbit polyclonal anti-Dynamin 1,2,3 antibodies (#115 002, Synaptic Systems, Göttingen, Germany), overnight at 4 °C. After washing, the membranes were incubated with IRDye® 800CW donkey anti-mouse IgG or IRDye® 680RD donkey anti-rabbit IgG (#926-32212 or #926-68073; LI-COR Biotechnology, USA), for 1 h at room temperature. The primary and secondary incubations were performed in blocking solution. The membranes were then washed in PBS and scanned using Odyssey CLx (LI-COR). Full blots can be found in the Source Data file.

**Metabolic labeling of newly synthesized proteins and glycans.** For visualization of newly-synthesized proteins on the cell surface, neurons were incubated in methionine-free media subsequent to TNR epitope blocking: DMEM (4.5 mg/ml glucose, lacking pyruvate, methionine, glutamine, and cysteine; Life Technologies, Carlsbad, CA, USA) supplemented with 50 μM L-azidohomoalanine (AHA; Life Technologies), 812 μM MgCl2, 6.5 mM HEPES, 260 μM cysteine, 1:50 B27 (Gibco, Life Technologies), and 1:100 GlutaMAX (Gibco, Life Technologies). For visualization of surface glycans, 50 μM Click-IT™ GalNAz and/or GlcNAz (#C33365, C33367, Thermo Fisher Scientific, USA) were diluted directly into the neurons' media subsequent to blocking, and remained throughout the experiment. Live, copper-free click labeling of surface AHA or glycans was performed after TNR labeling using strain-promoted azide-alkyne cycloaddition (SPAAC)[68]. Neurons were incubated with 1:1000 dibenzocyclooctyne (DBCO) conjugated to Alexa-Fluor647 (#CLK-1302, Jena Bioscience, Germany) diluted in Ca$^{2+}$/Mg$^{2+}$-free Tyrode's solution with 1 mM EGTA (#67-42-5; Merck, Germany).

**Sequestration of TNR on the plasma membrane using large antibody aggregates.** Sequestration of TNR epitopes on the plasma membrane was done as previously described[41]. To perturb the newly-emerged TNR epitopes, cultured neurons or organotypic hippocampal slices cultures were blocked for 2 h with 1:100 unlabeled TNR antibodies (neurons) or 50 μg/mL unlabeled Fab fragments directed against TNR (organotypic slices), together with 1 mg/mL unlabeled FluoTag-X2 anti-mouse secondary nanobodies. The cultures were then labeled 12 h after blocking with 1:500 TNR antibodies (neurons) or 2 μg/mL Fab fragments directed against TNR (organotypic slices), together with 1 mg/mL biotinylated FluoTag-X2 secondary anti-mouse secondary nanobodies (custom-made; NanoTag, Göttingen, Germany). This ensures that only the newly-emerged epitopes, which surfaced during the 12 h after blocking, are tagged by biotin. Alternatively, to perturb

specifically the epitopes that do not recycle ("all other epitopes"), the blocking step was performed using unlabeled TNR antibodies/Fab fragments, but without unlabeled anti-mouse secondary nanobodies. The cultures were incubated for 12 h, to ensure that the recycling epitopes are endocytosed, and then the biotinylated secondary nanobodies alone were applied. These now cannot detect the endocytosed molecules, and cannot detect the newly-emerged TNR epitopes, since these molecules are not bound by the TNR antibodies. The biotinylated secondary nanobodies only detect the TNR antibodies on the surface molecules that were present at the time of labeling (12 h previously) and did not endocytose in the meanwhile – meaning the stable, non-recycling pool. Subsequently, the cultures were incubated with antibody aggregates diluted 1:20 in Tyrode (untreated cultures were incubated in plain Tyrode). The antibody aggregates consisted of goat anti-biotin (#B3640; Merck, Germany) and STAR635P-conjugated donkey anti-goat (#ST635P; Abberior GmbH, Göttingen, Germany). Donkey anti-goat and goat anti-biotin antibodies were diluted 1:10 in PBS, and the mixture was left to rotate overnight at 4 °C. For the analysis of stimulus-induced vesicle release in cultured neurons, the aggregates were added for 30 min, after which the neurons were incubated for 5 min with 1:100 polyclonal rabbit antibodies directed against the lumenal domain of Syt1 conjugated to the fluorescent dye Oyster488 (#105 103C2; Synaptic Systems, Göttingen, Germany) in either plain Tyrode (stimulated condition) or $Ca^{2+}$-free Tyrode containing 1 mM EGTA and 1 μM TTX (#1069; Tocris Bioscience, Germany) (unstimulated condition). To reveal the releasable vesicles, neurons were stimulated electrically with field pulses with a frequency of 20-Hz and an intensity of 100 mA, for 10 s. This stimulus causes the exo- and endocytosis of 50% of the active vesicle pool (the other ~50% are already present on the surface membrane, waiting for endocytosis[20,69]). The stimulations were performed with a 385 Stimulus Isolator and an A310 Accupulser Stimulator (World Precision Instruments, Sarasota, FL, USA), using a custom-made platinum plate field stimulator (8 mm-distanced plates). Following the stimulation, the neurons were incubated a further 5 min with the Syt1 antibodies and then briefly washed and fixed. For the analysis of structural changes to synapses in organotypic slices, the aggregates were added for 12 h, after which the cultures were washed and fixed (see 'Fixation and post-fixation immunostaining'), and then immunostained with FluoTag-X2 anti-VGlut1 nanobodies directly conjugated to STAR-580 (#N1602; NanoTag, Göttingen, Germany), to identify synapses. To visualize the synaptic membranes in neuronal cultures, the immunostained neurons were incubated with DiO (see 'Fixation and post-fixation immunostaining'). To visualize synaptic membranes in organotypic slices, the slices were infected with AAV9 Syn-eGFP (eGFP under the synapsin 1 (hSyn) promoter; Penn Vector Core, Pennsylvania, USA), at 1ul per well, with a titer of $1.32×10^{13}$, at DIV 7, to ensure sufficient expression at DIV 14. Following the post-fixation labeling, the neuronal cultures/organotypic slices were mounted in Mowiol.

Live-cell incubations were performed at 37 °C, and live washing steps were performed in pre-warmed Tyrode.

**$Ca^{2+}$ imaging.** For measuring $Ca^{2+}$ responses, neurons were infected with IncuCyte® NeuroBurst Orange Reagent (#4736; Sartorius, Germany) at DIV7. After 7 days, the neurons were treated with antibody aggregates, (see 'Sequestration of TNR on the plasma membrane using large antibody aggregates') and then mounted in a live-imaging chamber in pre-warmed Tyrode. The neurons were imaged under constant temperature (37 °C) with an inverted Nikon Ti microscope with a Plan Apochromat 20× 0.75 NA air objective (Nikon Corporation, Chiyoda, Tokyo, Japan), equipped with a cage incubator system (OKOLab, Ottaviano, Italy). For recording spontaneous bursting activity, the neurons were imaged at a frequency of 1 frame per second (fps) for a total of 1 min. For imaging the $Ca^{2+}$ response to stimulation, 1 μM TTX was added to inhibit the neurons' spontaneous activity, thus ensuring that the opening of presynaptic voltage-gated $Ca^{2+}$ channels (VGCCs) is solely the result of the external electrical stimulations. The neurons were imaged with a Plan Apochromat 100×/1.4 NA oil objective at 1 fps for 10 s, after which a brief electrical stimulation was applied (20-Hz, 100 mA, 1 s) with an Accupulser Stimulator. Following the stimulation, the neurons were imaged for an additional 30 s.

**Immunostaining of brain slices from kainic acid-treated mice and 5xFAD mice**
*Kainic acid treatment and preparation of sections.* Kainic acid or vehicle injections were performed in 3 month-old C57BL6/J mice as previously described[47]. In brief, the animals were implanted with cannulas positioned above the right hippocampal CA1 region. Two weeks later, 120 nL of 10 mM kainic acid in vehicle (Millipore distilled sterile water) was delivered through the cannula. As a control, 120 nL of vehicle was delivered. Seven days later, the mice were sacrificed in a $CO_2$ chamber and perfused with PBS and 4% PFA in PBS. Brains were kept in 4% PFA in PBS for post-fixation overnight at 4 °C. Then the brains were put in a 1 M sucrose solution in PBS at 4 °C for cryoprotection. Two days later, brains were frozen in 2-methyl-butane at −80 °C and kept frozen until sectioning. The frozen tissues were sectioned into ~30 μm-thick slices using a cryostat.

*Preparation of sections from 5xFAD mice.* The sections were prepared as previously described[70]. In brief, 3 male 5xFAD (familiar Alzheimer's disease mouse model[71]) or wildtype littermates with the C57BL6/J background were decapitated at the age of 6–9 months and brain tissue was isolated in ice-cold PBS. The left hemispheres

were fixed in 4% PFA in PBS overnight, at 4 °C. The tissue was then cryoprotected overnight in 30% sucrose in PBS, and frozen in 2-methyl-butane at −80 °C until sectioning. The frozen tissues were sectioned into ~40 μm-thick slices using a cryostat.

*Immunostaining.* The slices were blocked with PBS containing 2.5%, with PBS containing 2.5% bovine serum albumin (BSA; #A1391-0250; Applichem, Germany) and 0.3% Triton X (#9005-64-5, Merck, Germany) for 1 h at room temperature. Then, the slices were incubated with 1:500 Atto647N-conjugated mouse monoclonal TNR antibodies (custom made, Synaptic Systems, Göttingen, Germany) and 1:100 rabbit polyclonal anti-calreticulin (#12238 S; Cell Signaling, Germany), overnight at 4 °C. The slices were washed, and then incubated with 1:200 Alexa488 donkey anti-rabbit IgG secondary antibodies (#711-545-152; Dianova, Germany) for 1 h at room temperature. All antibody incubations were performed in blocking solution. Following the staining, the slices were washed in PBS, and then mounted in Mowiol.

For both the KA-treated and 5xFAD mice, images were acquired of the CA1 and dentate gyrus regions.

**Correlated fluorescence and nanoSIMS imaging of brain slices from $^{13}$C-lysine-pulsed mice**
*Slice preparation.* $^{13}C_6$-lysine pulsing of adult mice was performed as previously described[2]. The brains were snap-frozen immediately upon removal from the skull and stored at −80 °C. After thawing, a thick coronal slice was extracted from the center of each brain containing the hippocampal formation (~30-50 mm) and fixed in 4% PFA in PBS overnight, at 4 °C. The slices were then embedded in Tissue-Tek® O.C.T.™ Compound (Sakura, Finetek USA Inc., Torrance, CA, USA) and frozen at −80 °C. The frozen slices were sectioned into ~60 μm thick slices on a Leica CM1850 cryotome.

*Immunostaining and mounting.* The slices were blocked for 1 h with PBS containing 2.5% bovine serum albumin (BSA; #A1391-0250; Applichem, Germany) and 0.3% Triton X (#9005-64-5, Merck, Germany) for 1 h at room temperature. All sections were immunostained with 1:500 Atto647N-conjugated mouse monoclonal TNR antibodies (custom made, Synaptic Systems, Göttingen, Germany) and 1:100 rabbit polyclonal anti-calreticulin (#12238 S; Cell Signaling, Germany), overnight at 4 °C. The slices were washed, and then incubated with 1:100 Alexa546-conjugated goat anti-rabbit IgG (#A11035; Thermofisher Scientific, USA) for 1 h at room temperature. The incubations with the primary and secondary antibodies were performed in the same blocking solution. Following the immunostaining, the slices were embedded in medium grade LR white (London Resin Company, London, UK), as previously described[72]. In brief, the sections were dehydrated in increasing concentrations of ethanol, and then placed on glass coverslips and incubated for 1 h in a 1:1 mixture of LR white and 50% ethanol, and for a further hour with pure LR white (all at room temperature). The slices were then overlayed with Beem® capsules (BEEM Inc., West Chester, PA, USA), which were filled with a mixture of LR white and accelerator (1 drop of LR white accelerator was added to 10 mL LR white), and heated for 1.5 h at 60 °C to allow for polymerization. After cooling, the glass coverslips were broken off, and the samples were cut using an ultramicrotome (EM UC6, Leica Microsystems, Wetzlar, Germany) into thin sections of ~200 nm, and then placed on Silicon wafers (Siegert Wafer GmbH, Aachen, Germany).

*Imaging.* The samples were first imaged with epifluorescence microscopy, on an inverted Nikon Ti microscope (Nikon Corporation, Chiyoda, Tokyo, Japan) equipped with a Plan Apochromat 100×, 1.45 NA oil immersion objective, a a 1.5× optovar lens and an IXON X3897 Andor (Belfast, Northern Ireland, UK) camera. After fluorescence imaging, the same areas were imaged with a NanoSIMS 50 L instrument (Cameca, France) equipped with an 8 kV Cesium primary ion source. To reach the steady-state of the secondary ion yield, prior to each measurement, the area of interest was implanted with a current of 600 pA for 60 s (primary aperture D1:1). Subsequently, a primary ion current of ~2.5 pA (primary aperture D1:2) was applied during the imaging on areas of 55 × 55 μm, obtaining images of 512 × 512 pixels and resulting in a pixel size of 107.4 nm. The dwell time was 5 ms per pixel. The detectors were set to simultaneously collect the following ions: $^{12}C^-$, $^{13}C^-$, $^{12}C^{14}N^-$ and $^{13}C^{14}N$ and the mass resolving power of the instrument was adjusted to ensure the discrimination between isobaric peaks such as $^{13}C^{14}N^-$ from $^{12}C^{15}N^-$, or $^{13}C^-$ from $^{12}C^1H^-$. A single plane was analyzed to avoid loss of lateral resolution caused by drift correction. The images were exported using the OpenMIMS plugin from Fiji (http://nano.bwh.harvard.edu).

**Imaging of fluorescence recovery after photobleaching (FRAP)**
*Generation of viral vectors.* In order to specifically investigate the recovery of hyaluronic acid-based matrix around synapses, AAV expression vector were designed, carrying mDlg4 (Gene ID: 13385) fused with EGFP with a linker sequence 3xGGGGS in between the ORF to label postsynaptic density, which was custom made as a service by VectorBuilder (Chicago, IL, USA). To label hyaluronic acid-based ECM an AAV expression vector was designed, carrying link protein hyaluronan and proteoglycan link protein 1 (HAPLN1, Gene ID: 12950) fused with mScarlet subcloned from the plasmid pCytERM_mScarlet_N1 (Addgene plasmid #

85066). Viral particles were produced as described above (see 'shRNA-mediated TNR knockdown')

*Culture preparation and infection.* For the FRAP live imaging experiments, rat cortical cultures (from E18-E19 and P0-P3) were prepared as previously described[73–75]. The neurons were seeded at a density of ~ 80,000/cm$^3$ on circular glass coverslips (18 mm diameter), and maintained in Neurobasal medium (Gibco, Thermofisher Scientific, USA), supplemented with B27 (Gibco, Thermofisher Scientific, USA), L-Glutamine (Gibco, Thermofisher Scientific, USA) and penicillin/streptomycin (PAA Laboratories, Pasching, Austria). At DIV7-9, the neurons were infected with the AAV-PSD95-eGFP (1 μl per well with a titer of $4.5 \times 10^{12}$) and AAV-HAPLN1-Scarlet (1 μl per well with a titer of $5 \times 10^{12}$).

*FRAP imaging.* All live imaging experiments were performed on neurons at DIV 21–23 after transferring infected cultures to a Quick Change Chamber 18 mm Low Profile RC-41LP (Warner Instruments), and keeping them in the original culture media at 37 °C in the constant presence of humidified carbogen for the duration of the imaging. The experiments were performed using a Leica SP5 confocal microscope (Leica-Microsystems, Mannheim, Germany; LAS AF software, version 2.0.2) equipped with Argon (458, 476, 488, 496, 514 nm laser lines), Diode Pumped Solid State (DPSS, 561 nm) and HeNe (633 nm) lasers and acousto-optic tunable filters (AOTF) for selection and intensity adaptation of laser lines. Confocal images with parameters of $512 \times 512$ pixels display resolution, 8-bit dynamic range, 63× objective, NA 1.40, 3× optical zoom, voxel size approximately $0.16 \times 0.16 \times 0.3$ μm$^3$) were acquired. In all FRAP experiments, one baseline image was collected as a prebleach image before 15–16 circular ROIs of 12–15 μm$^2$ size around PSD-95 puncta were bleached by high-intensity 561-nm laser, resulting in bleaching of only HAPLN1-scarlet fluorescence. After bleaching, images were acquired at the rate of one image per 10 min up to 16–17 h to monitor the recovery of hyaluronic acid-based matrix. Images were acquired with the same settings for all samples before and after bleaching. Data were collected following bleaching at 7–16 independent bleaching spots in each experiment. 14 independent experiments (FRAP movies) were performed, using 14 coverslips from 3 independent culture preparations. The images were analyzed by manually selecting circular regions of interest (ROIs) centered on the bleached spots, and monitoring the signal in the ROIs. The signal was corrected for the loss of signal caused by the repeated imaging. This parameter was monitored in ROIs placed in regions from the same regions, which were not bleached initially. The fluorescence signals were normalized to the pre-bleaching intensity, and were then averaged, as shown in the respective figure. The synaptic enrichment values shown in Supplementary Fig. 20f represent the number of synapses identified, normalized to the total PSD95 fluorescence within the respective ROIs.

**Fluorescence imaging.** Fluorescence imaging was performed using the following setups, unless otherwise specified.

*2-color STED microscopy.* The imaging was performed on an Abberior easy3D STED microscope (Abberior GmbH, Göttingen, Germany) equipped with a UPlanSApo 100×/1.4 NA oil immersion objective (Olympus Corporation, Shinjuku, Tokyo, Japan). Pulsed 488 nm, 561 nm, and 640 nm lasers were used for excitation, and easy3D module 775 nm laser was used for depletion. For each channel, the pinhole was set to 1 Airy unit. Avalanche photodiodes (APDs) were used for detection.

*Confocal microscopy.* The imaging was performed either on an Abberior easy3D STED microscope (detailed above), or on a Leica TCS SP5 microscope (Leica, Wetzlar, Germany) equipped with HCX Plan Apochromat 100× and 63×/1.4 NA oil objectives. For the latter, the 488 nm line of an argon laser, the 561 nm and 633 nm lines of a Helium-Neon laser were utilized for excitation, using acousto-optic tunable filters to select appropriate emission wavelengths. The images were acquired with photomultiplier tubes or Hybrid detectors. For each channel the pinhole was set to 1 Airy unit.

*Epifluorescence microscopy.* The imaging was performed on either an inverted Nikon Ti microscope (Nikon Corporation, Chiyoda, Tokyo, Japan) equipped with 100×/1.45 NA (oil), 20×/0.8 NA (air) and 10×/0.5 NA (air) Plan Apochromat objectives, and an IXON X3897 Andor (Belfast, Northern Ireland, UK) camera, or with an inverted Olympus microscope (Olympus Corporation, Shinjuku, Tokyo, Japan), equipped with a UPlanFL 20×/0.5 NA air objective, a UPlanSApo 100×/1.4 NA objective and a charge-coupled device camera (F-View II; Olympus). The live imaging experiments described in Fig. 4a and Supplementary Fig. 6, were performed on the Nikon Ti microscope, fitted with a cage incubator system (OKOLab, Ottaviano, Italy). The temperature was maintained at 37 °C and CO$_2$ was maintained at 5%. The neurons were imaged in their own cell media at a rate of 1 frame every 2 h. For each frame, 5 z-slices were acquired resulting in a stitched mosaic image of 5 × 5 lateral fields of view. All other live imaging experiments were performed on the Olympus microscope. For the experiments described in Fig. 4c and Supplementary Figs. 13, 14e, and 15c, five images of different fields of view were acquired before and after treatment with Proteinase K. For the experiment

described in Supplementary Fig. 14c–d, an image was acquired in the same field of view before and after treatment with Proteinase K.

**Image and data analysis.** Data was acquired using Leica Application Suite Advanced Fluorescence 2.7.3.9723, Abberior Instruments Imspector v16.3, NIS Elements 5.02.03 and Olympus CellSens Dimension 2.3. Analyses were performed in Matlab (MathWorks, Natick, MA, USA) and Python (Python Software Foundation).

*Calculation of mean fluorescence intensity in epifluorescent images.* Background fluorescence was subtracted by selecting unlabeled regions in the images, or by using an empirically defined threshold. Regions of interest (ROIs) were selected manually and the mean fluorescence intensity was calculated within the ROIs. The ROIs were selected to include neurites (for the experiments described in Figs. 2b, c, f, 5a, b, 9d, e, and Supplementary Figs. 8, 9a, b, 10, 11, 12, 14a, b, 15b, d–g and 20a–c), both neurites and somas (Supplementary Figs. 5b and 18), or somas (Fig. 7c, d and Supplementary Fig. 15a). For analyzing the epifluorescent images from the experiments described in Supplementary Figs. 4a, b, c and 9c, d, the mean fluorescence intensity was calculated for entire images. For the live imaging experiment described in Fig. 4a and Supplementary Fig. 6, the mean intensity was calculated on the maximum intensity projection images acquired at each timepoint. ROIs were drawn around neuronal somas and the mean intensity for each individual neuron was normalized to the first timepoint. For the experiments described in Figs. 4c, 5d, e and Supplementary Figs. 13, 14e and 15c, ROIs containing neurites were selected within the images. For each experimental day, a ratio was calculated between the mean intensity before and after incubation with the proteinase. For the experiment described in Supplementary Fig. 14c, d, the images before and after proteinase incubation were aligned, and the mean fluorescence intensity was determined in selected ROIs in corresponding regions of the images.

*Analysis of confocal images of immunolabeled beads.* For analyzing the confocal images of immunolabeled beads (Fig. 1), the images were segmented based on an empirically defined threshold, to detect the individual beads, and the background signal was removed. The total fluorescence intensity was calculated for each bead, and a mean value was calculated per experiment.

*Quantification of TNR at synapses by STED and confocal microscopy.* The synapses in the images shown in Figs. 2d and 3 were identified by manual thresholding of the VGlut1 or Syt1 channels, and square regions of interest centered on the individual VGlut1 or Syt1 puncta were excised. For the visualization of synaptic enrichment of TNR, as depicted in Fig. 2d, the excised image segments were automatically rotated to maximally overlap in the DiO channel, and averaged to one single image illustrating TNR localization at an average synapse (following procedures previously described[22,76]). For the enrichment analysis at dendrites and axons, the mean TNR fluorescence intensity was calculated for all image segments in concentric circles of increasing radii. To quantify the amount of newly-emerged TNR epitopes after 12 h specifically at synapses, as shown in Fig. 2e, the mean fluorescence intensity of TNR was estimated in VGlut1-positive pixels. For the correlation of TNR fluorescence to the presynaptic recycling vesicle pool size, as depicted in Fig. 3, the image segments surrounding individual presynapses (determined by the Syt1 channel) were sorted by the mean Syt1 fluorescence intensity, and then binned in five ordinal groups, to include a similar number of synapses. For each experiment, the mean fluorescence was calculated in each bin, and was normalized to the median intensity of the respective experiment. A similar procedure was used to analyze the TNR fluorescence in relation to the postsynaptic spines: image segments were excised for synaptic regions based on the VGlut1 channel, and mushroom-shaped spines were identified manually in these segments through the DiO channel. The spine size (area) was determined by thresholding the DiO signal of each individual image segment and counting the pixels above the threshold. The image segments were sorted by the spine size, and then binned in five ordinal groups, to include a similar number of synapses. For each experiment, the mean fluorescence and size were calculated in each bin, and normalized to the median of the respective experiment. For colocalizing TNR with the excitatory and inhibitory synapse markers VGlut1 and VGAT, an image-wide pixel-by-pixel correlation was first calculated between the images. The pixels indicating a high correlation between the channels were selected, and the TNR amount corresponding to these pixels was calculated as % of the total TNR staining in the images (corrected for background). To analyse the percentage of VGlut1+ and VGAT + synapses that colocalize with TNR, ROIs were drawn around neurites within the images. The VGlut1/VGAT channels were thresholded to identify individual synapses, and the proportion of objects that overlapped with the signal in the TNR channel was quantified.

*Characterization of cell types in dissociated neuronal cultures.* To count the number of neurons, astrocytes, oligodendrocytes, and microglia, the images were manually thresholded in each channel to segment individual cells. The total number of cells was determined by counting the number of objects in the Hoechst channel. The fraction of each cell type was determined by calculating the percent of objects overlapping with the Hoechst signal, out of the total number of cells. To quantify

the proportion of PNN-associated neurons in dissociated hippocampal cultures at DIV14, the number of WFA + neurons per field of view was counted manually. The total number of neurons was determined automatically by counting the number of objects in the thresholded NeuN image.

*Quantification of TNR uptake in astrocytes.* ROIs were selected surrounding the GFAP + and NeuN+ cells, based on a manual thresholding procedure. The TNR signal overlapping with these ROIs was quantified, as a percentage of the total TNR signal in the images.

*Quantification of TNR at neurites by confocal microscopy.* For the comparison of TNR in axons and dendrites as depicted in Fig. 4b and Supplementary Fig. 7, ROIs were selected on the axons and dendrites separately, based on the DiO channel (and the AnkyrinG staining, for Fig. 4b). The mean TNR fluorescence was calculated within these ROIs, indicating the signal density.

*Colocalization analysis.* For colocalizing TNR and organelles or surface integrins in 2-color STED images, ROIs were drawn around cells to include TNR signal, and an image-wide pixel-by-pixel correlation was first calculated between the images. The pixels indicating a high correlation between each of the channels were selected, and the TNR amount corresponding to these pixels was calculated as % of the total TNR staining in the images. For colocalizing TNR with internalized integrins in confocal images, line scans were drawn through the images and Pearson's correlation was calculated between the two channels. ~400 lines were analyzed per experiment, and the mean correlation coefficient was calculated for each experiment. Afterward, the percentage of correlation coefficients equal to, or larger than 0.7 was calculated for each experiment and normalized to a positive control (two identical secondary antibodies conjugated to different fluorophores). A similar procedure was used for the colocalization of TNR-containing organelles to LysoTracker.

For colocalizing TNR with metabolically labeled glycans or proteins in 2-color STED images, a region of each image was sampled blindly, and the numbers of overlapping and separated spots were determined. The percentage of colocalizing spots was calculated, after which the colocalization percentage of a negative control (immunostaining with an unspecific antibody) was subtracted. To analyse the amount of recombinant TNR that is internalized into synaptic regions in acute rat brain slices, an image-wide pixel-by-pixel correlation was calculated between the VGlut1 and rTNR channels, and the mean correlation was determined.

*Analysis of dendritic spine head size following treatment with TNR antibody aggregates.* Individual synaptic images were cropped from full images and aligned based on the DiO or eGFP channel, as was described above (see 'Quantification of TNR at synapses by STED and confocal microscopy'). To quantify the average spine head area, the DiO channel was thresholded for each individual image segment, and the pixels above the threshold were counted.

*Analysis of $Ca^{2+}$ imaging experiments.* ROIs were specified surrounding individual neurons. For the stimulation experiments, the fluorescent signal in each neuron was normalized to the baseline observed in the first few imaging frames, before stimulation. For analyzing the spontaneous activity, spontaneous events were identified as peaks in fluorescence above an empirically determined threshold.

*Comparison of old /new TNR epitope localization with 2-color STED.* To create the images of the average dendritic spine, image segments containing individual synaptic puncta were manually aligned based on the DiO channel (see 'Quantification of TNR at synapses by STED and confocal microscopy'). To calculate the percentage of newly-emerged epitopes that colocalize with old epitopes, the images were thresholded in the channel corresponding to the 'new epitopes', to find individual spots. The % colocalization was determined by the number of new epitope spots that also contained signal from the 'old epitopes' channel.

*Comparison of extracellular and intracellular TNR in slices from 5xFAD mice and mice injected with kainic acid.* For each slice, ROIs were selected in intracellular somatic (identified as an envelope of the calreticulin immunostaining, which labels the ER) and adjacent, presumably extracellular regions. The mean fluorescence intensity was calculated in the ROIs following subtraction of the background signal. The overall mean was determined for the extracellular and intracellular TNR signals, for each experimental condition. These were then normalized to the mean of extracellular TNR in the control mice.

*Analysis of correlated fluorescence and nanoSIMS images.* The analysis was done as previously described[72]. Briefly, the nanoSIMS images and the fluorescence images were overlayed, and circular ROIs (1.2 μm diameter) were selected in TNR or calreticulin-rich areas, or within cell nuclei, based on the fluorescence images. The average values were then calculated in the corresponding areas in the nanoSIMS images, and the respective isotope ratios were determined. This ratio was then expressed as fold over the baseline isotopic ratio of $^{13}C/^{12}C$ (or $^{13}C^{14}N/^{12}C^{14}N$), measured in regions of the embedded sample, outside the biological specimens.

*Analysis of recombinant TNR (rTNR) Western blots.* For each blot, the fluorescence intensity of the bands was determined by manually selecting a rectangular ROI surrounding the band and correcting for local background (see[77]).

**Statistical analysis.** Statistical significance was calculated with two-tailed t-tests, Mann-Whitney U-tests, Wilcoxon signed rank test, ANOVA, Kruskal-Wallis, or Friedman tests, according to the type of data analyzed. Post-hoc analyses were calculated with the tests that were recommended and implemented in the software packages used for the respective tests (GraphPad Prism version 8, GraphPad Software, USA; Matlab version 2017b, the Mathworks Inc., Natick, MA, USA), as noted in the figure legends. For repeated-measures ANOVA, the Greenhouse-Geisser adjustment was applied to account for any departures from sphericity. Correlations were calculated with Pearson's R or Spearman's ρ. A p-value of < 0.05 was considered statistically significant.

**Reporting summary.** Further information on research design is available in the Nature Research Reporting Summary linked to this article.

## Data availability
Image data are available from the corresponding author on reasonable request. Source data are provided with this paper.

## Code availability
Code is available upon request from Tal M. Dankovich and Silvio O. Rizzoli.

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

## Acknowledgements

We thank Dr. Renato Frischknecht from the Leibniz Institute for Neurobiology (LIN) for providing mice deficient in TNR and brevican as well as control mice, and Dr. Weilun Sun from DZNE Magdeburg for performing TNR immunostaining and imaging of brain sections. We thank Shaobo Jia from DZNE and Armand Blondiaux from LIN for providing brain sections from kainic acid-treated mice. We thank Katrin Boehm, David Baidoe-Ansah and Hadi Mirzapourdelavar from the DZNE Magdeburg for providing brain sections from 5xFAD mice and AAV preparation, and Isabel Herbert from LIN for preparation of cultures for FRAP experiments. This work was supported by grants to S.O.R. from the German Research Foundation (Deutsche Forschungsgemeinschaft, DFG SFB1190/P09, DFG SFB1286/A03, RI 1967/10-1/NeuroNex, RI 1967/7-3, RI 1967/11-1), from the Nieders. Vorab (76251-12-6/19/ZN 3458) and from the German Ministry for Education and Research, 13N15328/NG-FLIM. Also supported by the DFG GRK SynAge 2413/1, TP6 to A.D, and under Germany's Excellence Strategy-EXC 2067/1-390729940. This project has received funding from the European Research Council (ERC) under the European Union's Horizon 2020 research and innovation programme (Grant agreement no. 835102).

## Author contributions

S.O.R., T.M.D. and A.D. designed the experiments and statistical analysis. T.M.D. performed all fluorescence imaging experiments and the cell-surface biotinylation experiments, with the following exceptions: the fluorescence imaging of neurocan, hyaluronan and WFA was performed by G.C.P. and the imaging of metabolically labeled proteins and glycans was performed by P.E.G., T.M.D. and L.H.M.O. performed the co-immunoprecipitation experiments and the Western blotting. H.A.H. and S.K. assisted with cellular experiments. J.D. performed the fluorescence imaging of the brain sections from kainic acid-treated as well as the sections from 5xFAD mice. The AAV-based TNR shRNA vectors were designed, produced and provided by R.K., and the viral infections were performed by S.B. and B.C. The FRAP experiments were performed by R.K. The electrophysiology experiments were performed by G.B. T.M.D. and V.K. prepared the samples for nanoSIMS. NanoSIMS imaging was performed by P.A.G. and K.G. The data analysis was performed by T.M.D. and S.O.R. The manuscript was written by T.M.D. and S.O.R. and edited by A.D.

## Competing interests

S.O.R. has received compensation as a consultant of NanoTag Biotechnologies GmbH and owns stock in the company. The remaining authors declare no competing interests.
