## [Peer Review File · Nature Communications]

REVIEWER COMMENTS

Reviewer #1 (Remarks to the Author):

The manuscript “Extracellular matrix remodeling through endocytosis and resurfacing of Tenascin-R” by T. Dankovich et al. investigates the activity-dependent remodeling of the extracellular matrix (ECM) protein tenascin-R (TNR), as well as other ECM molecules, near synapses. Using multiple assays to label and track surface and endocytosed TNR on neurons in vitro, the authors demonstrate that surface-localized TNR is endocytosed and recycled back to the surface, which can be modulated by neuronal activity. Furthermore, they suggest that this periodic ECM recycling is important for synapse function. These findings are very interesting and challenge existing assumptions about ECM plasticity and the role of ECM in synaptic plasticity. The data are generally quite strong and support the conclusions. There are concerns regarding overall significance in more physiological settings, which would increase the impact. This and other concerns are expanded upon below:

1. The author’s reliance on in vitro systems limits the broader impact of their findings. Although they performed some experiments in brain slice, it is unclear if these results translate to an in vivo setting. While appreciated that these experiments could be quite challenging, some in vivo relevance would significantly increase the impact.
2. The authors suggest that TNR recycling occurs at synaptic sites and perturbing recycling disrupts SYT1-mediated vesicular release. To increase the functional significance of their findings, more in vitro experiments to assess synapse function and/or plasticity would strengthen these conclusions and increase the impact. This could be done by electrophysiology, calcium imaging, and/or assessment of structural synapse changes upon manipulation of TNR recycling.
3. More controls are warranted. To rule out the possibility of incomplete glutathione cleavage, the authors should include as a control a surface protein which is stable on the surface and is not endocytosed. Although appreciated that some controls are included to rule out non-recycled TNR degradation, the inclusion of another protein control that is endocytosed but not recycled back would bolster these findings.
4. In Extended Fig 1b, the authors show the percentage of total TNR that co-localizes with VGLUT1 and VGAT. It would also be informative to indicate the percentage of VGLUT1+ and VGAT+ puncta that are TNR positive.
5. While there seems to be more recycled TNR at synapses compared to non-recycled, it doesn’t seem from the images that the recycled TNR is preferentially at synapses vs. other parts of the neuron in Fig 2d or in Fig 3. The authors should clarify this in the text and also perform measurements to show that the recycled TNR is enriched at dendrites with an associated bouton vs. the rest of the dendrite or neuron. If recycled TNR is not highly enriched in the synapse and also exists at other places along the neuron, what would this mean for the neuron?

6. Can the authors comment on the TNR shown outside the DiO-labeled cell (e.g. Fig 2d top right panel)? Is this secreted TNR?
7. Fig 3c is very intriguing and a highlight of this paper. Inclusion of a protein that does not recycle and a synaptic protein such as Syt1 would be useful comparisons.
8. The authors should test whether the cyclical surface recycling of TNR can be blocked by inhibition of dynamin with Dyngo (like in Extended Fig 7), either throughout the experiment or between days 2 and 3.
9. Could the authors speculate on why the TNR recycling is on a 3 day cycle in Fig. 3c and, perhaps, show some data supporting this? Could this give some functional insight?
10. The ordering of figure panels in Fig 2 is confusing. Panels E and F should follow D. Otherwise, it is easy to overlook these panels.

Reviewer #2 (Remarks to the Author):

Reviewer's comments relating to "Extracellular matrix remodelling through endocytosis and resurfacing of tenascin-R" by Dankovich et al.

The manuscript by Dankovich and colleagues claims that ECM remodelling may not only occur consequent to degradation by proteases, but also by endocytosis and recycling of the scaffolding protein of the extracellular matrix (ECM) tenascin-R. The idea is interesting and the claim is novel. However, I am not fully convinced that the work presented supports this statement. I have several comments, as listed below:

- 1) Tnr is a soluble, secreted ECM molecule; for endocytosis it clearly requires a receptor system. The mechanistic basis of endocytosis is not explained in this study.
- 2) The study does not distinguish the different modes of Tnr in the extracellular matrix. Thus, Tnr is released into the environment by secretion and thereby part of the interstitial matrix of the CNS. It may bind to axonal surfaces by interaction with the IgSf member F3/F11/contactin. Moreover, it may be associated with perineuronal nets by interaction with other ECM components. So it is unclear which association with the neuronal surface is studied in this work. In fact no proof is given that uptake and recycling of Tnr holds true for the PNN-associated Tnr. To this end co-labelling with a PNN marker such as aggrecan would be required, for example in Fig. 3.

- 3) The observation that Tnr can be documented in PNNs in culture confirms earlier reports. However, there is no direct evidence that the PNNs are remodelled in this system nor that Tnr is pinched off the PNNs. It could as well be taken up from the medium, or axonal surfaces.
- 4) What is the ratio of PNN-wearing neurons in the culture system used for the study?
- 5) Would the authors comment about the source of Tnr in their culture system? Is Tnr produced by the PNN-positive or –negative neurons, or both?
- 6) In their model, the mouse neurons are cultured in close contact on a monolayer composed of astrocytes. The rat hippocampus is prepared from postnatal animals, at a stage when a significant number of glial cells, in particular astrocytes are already present. The latter can potentially engulf membranes or endocytose macromolecules. In the approach detailed in Fig. 1 the authors rely on a combination of pulse chase using glutathione and subsequent lysis and affinity beads to distinguish surface-associated, endocytosed and non-recycled Tnr. How can they be certain that they are seeing effects in neurons? Glial cell types and astrocytes are probably also part of their culture system.
- 7) The culture system is not characterized using classical cell markers, such as GFAP and O4, the fraction of glial cells after the incubation period of up to 14 DIV was not determined.
- 8) The label in Fig 3a is interpreted as revealing intracellular Tnr; to prove that the label in fact targets intracellular vesicles electron microscopy combined with immunogold labelling would be required.
- 9) Alternatively, co-labelling with markers of intracellular vesicles could be attempted, for example markers of the endosome, or of circulating intracellular vesicles, e.g. selected rab proteins. .
- 10) Does Tnr evade degradation in the lysosomes? The authors should show by Western blot developed with avidin peroxidase that the biotinylated Tnr is not (or is?) degraded in their system (which would be required for recycling integral Tnr).
- 11) The study is based on the use of one monoclonal antibody to Tnr (clone 619 from synaptic Systems). In fact, they may see degradation fragments of Tnr that carry the epitope. There is no proof that the signal they obtain is due to integral Tnr (see point above).
- 12) The authors claim that they can achieve a 100% saturation of the epitope by adding the Mab clone 619 to Tnr and further that this saturation is stable over time. Therefore, they argue that label obtained with directly labelled 619 can only be explained by the resurfacing of heretofore hidden Tnr epitopes. This is very unlikely because it is highly improbable to achieve complete saturation of 100% of the epitope in the first place. Second, one could assume that the addition of directly labelled antibody would compete with bound antibody, simply because both probes recognize the same epitope. In that case, bound fluorescent probe would not necessarily reflect newly externalized Tnr, but rather the effect of competing off unlabelled antibody.
- 13) Along the same lines, each equilibrium reaction such as antibody binding has an off-rate. Therefore, one would expect a progressive release of antibody once the incubation with unlabelled antibody is terminated. The more so as in the culture system the medium represents a very large

space for off-diffusion of the primary bound unlabelled antibody. In consequence, the bound labelled antibody may reflect an increasing number of free binding sites due to off-rate diffusion.

14) Thus, the reduced intensity of label in the blocked situation could as well reflect the lower concentration of the labelled probe in relation to the blocking probe.

15) In some case, the authors use Fab fragments for blocking the access to Tnr. It is well known that Fab fragments are monovalent and therefore of lower affinity than the divalent antibodies they are derived from. Thus directly labelled antibody should the more efficiently compete off the blocking reagent (see point 8).

16) In my view, the evidence of Tnr recycling is based in indirect evidence. A direct way to prove the claims of the paper would be to couple purified Tnr with biotin and add it to the culture system. Because the avidin-biotin interaction is of much (up to 1000-fold, 10-14) higher affinity than average antibody binding I trust that the itinerary of Tnr could reliably be traced using avidin probes.

Reviewer #3 (Remarks to the Author):

In this short manuscript Dankovich et al report on the unexpected observation that the ECM molecule tenascin-R (TNR) and possibly other ECM factors are endocytosed over many years and recycling by primary CNS neurons, often near synapses. The very slow reappearance (days) of TNR molecules on the neuronal surface appeared to be facilitated by neuronal activity as suggested by pharmacological blockers or enhancers of GABAA or glutamate receptor function. Additional pharmacological experiments using the partially non-specific drugs Dyngo (to block dynamin), BFA (to block Golgi traffic) or monensin, a ionophore that alters Na/ H homeostasis, interfered with TNR recycling to the surface. Finally, the authors conduct experiments in which they add anti-TNR antibody aggregates to neurons and observe decreased activity-induced exo-endocytosis of the SV marker synaptotagmin 1. Based on these experiments in cultured neurons and a few experiments in slices, a model is proposed according to which the remodeling of synapses is accompanied by the endocytosis and recycling of ECM molecules.

The observation that neurons are capable of slowly internalizing parts of the ECM is quite interesting and novel. The authors use a number of technical controls and variations of their assays ascertain that they are truly looking at TNR endocytosis and recycling. I concur with this conclusion. However, the paper lacks compelling functional data and mechanistic insights into the endocytic recycling process. These would be expected for a publication at this level and to leave a lasting impact on the field. The present form of the Ms seems premature for publication.

Specific questions to be considered before submission elsewhere:

1. It remains unclear how neuronal activity controls TNR endocytosis and/ or recycling and at what level. Such information is required to understand the physiological relevance of the process. Especially, experiments aimed at addressing the claimed function of TNR recycling for synapse remodeling in vitro and, ideally, in functional networks in vivo would be required to raise this study to the level expected from a Nat Commun paper.
2. Antibody aggregates may elicit non-specific effects on neuronal signaling and, hence, as the authors admit themselves can only serve as a crude door-opener to address the physiological relevance of ECM exo-endocytosis and recycling. More specific manipulations that would require additional mechanistic insight into the uptake and recycling pathway, e.g. using endocytosis defective TNR mutants or its so far unidentified cellular receptor would be a way forward (see also #3).
3. No information is provided as to the molecular machinery that mediates TNR recycling, apart from a possible function for dynamin that would need to be substantiated by genetic tools such as available dynamin 1/3 KO mice and neurons or at the very least lentiviral KD of dynamin.
4. Equally important, it remains undisclosed if and how altered activity in the culture system affects the structure and composition of the ECM that surrounds synapses in the culture system. Such effects would be expected if network activity controls TNR endocytosis and/ or recycling. Such changes could also be looked at in organotypic slice cultures for example.

Reviewer #4 (Remarks to the Author):

In this study, Dankovich and collaborators investigated the trafficking of Tenascin-R protein in cultured hippocampal neurons. Using several approaches, the authors show that Tenascin-R recycle between the plasma membrane and intracellular pool. Such a cycle will take up to 3 days and is regulated in an activity-dependent manner. This is an interesting topic and a well-designed study, which shed new light on the dynamics of extracellular matrix components. Yet, the study appears to be somehow preliminary and additional experiments are needed to strengthen the conclusion.

i) The authors state that recycled TNR molecules are enriched near synapses (e.g. lines 80-84). Yet, the STED illustrations show that TNR are indeed in the proximity of synapses as well as all along the dendritic shaft. The main claim that TNR are specifically trafficked near synapses appear thus overstated. How the authors exactly measured the % of TNR at synapses (Fig. 2e)? In addition, as the authors use a cultured system in which astrocytes and neurons are intermingled the exact relevance of the dendritic/spine TNR staining is unclear. Is this influenced by the presence of a nearby glial cells? Additional experiments and analysis describing the whole distribution of TNR along dendrites should be provided.

ii) Interpretation of the data from cultured slices is difficult and thus not convincing. What is exactly the staining observed and how specific it is remain not entirely clear. In addition, one may eventually expect difference between hippocampal layer. The illustrations show different locations within the hippocampus, which further complicate an understanding of the experiment. Clarifications and additional controls would be appreciated.

iii) Beside these experimental caveats, it remains unclear whether the recycled TNRs after days within neurons are still biologically active or, at least, have the capacity to embed with the ECM. This piece of information will be of great interest for the study.

Replies to Reviewers

Reviewer #1:

The manuscript “Extracellular matrix remodeling through endocytosis and resurfacing of Tenascin-R” by T. Dankovich et al. investigates the activity-dependent remodeling of the extracellular matrix (ECM) protein tenascin-R (TNR), as well as other ECM molecules, near synapses. Using multiple assays to label and track surface and endocytosed TNR on neurons in vitro, the authors demonstrate that surface-localized TNR is endocytosed and recycled back to the surface, which can be modulated by neuronal activity. Furthermore, they suggest that this periodic ECM recycling is important for synapse function. These findings are very interesting and challenge existing assumptions about ECM plasticity and the role of ECM in synaptic plasticity. The data are generally quite strong and support the conclusions.

We thank the Reviewer for the comments.

We replied to all of the Reviewer’s comments, as detailed below. We performed new experiments to address every one of the aspects raised by the Reviewer. Importantly, we would like to thank the Reviewer for pointing out the need for more experiments relating to physiological settings. We added several new figures dealing with this issue, which have improved our manuscript substantially.

There are concerns regarding overall significance in more physiological settings, which would increase the impact. This and other concerns are expanded upon below:

1. The author’s reliance on in vitro systems limits the broader impact of their findings. Although they performed some experiments in brain slice, it is unclear if these results translate to an in vivo setting. While appreciated that these experiments could be quite challenging, some in vivo relevance would significantly increase the impact.

The Reviewer raises a very important point here, which we addressed by a series of experiments involving adult animals.

First, we verified whether substantial TNR amounts could be found within the cell bodies of neurons in the hippocampi of adult mice. As shown in **Figure for Reviewers 1**, intracellular (somatic) TNR can indeed be found, and it appears to be quite abundant. Moreover, this TNR pool is subject to changes according to the functional state of the neurons. To increase the activity rate of the neurons, they were stimulated by *in vivo* kainic acid administration, in what constitutes a well-studied mouse model of epilepsy (Broekaart et al., 2021). This enhanced the levels of intracellular TNR, while leaving all other TNR (presumably extracellular) unaffected (**Figure for Reviewers 1a**). This implies that intracellular TNR is modified according to the state of the neurons, in agreement with our general hypothesis. Finally, the enhancement of the intracellular TNR levels was not due to neuronal damage, since another mouse model in which neuronal damage is prominent, the Alzheimer Disease 5xFAD model, showed no effects on intracellular TNR accumulation (**Figure for Reviewers 1b**).

Second, having verified that intracellular TNR can be found in the hippocampus *in vivo*, we turned to the question of whether these molecules are newly synthesized, or are older molecules that the cells have endocytosed from the ECM. To analyze this, we used a technique we introduced in the past, correlated optical and isotopic nanoscopy (COIN, Saka et al., 2014). Wild-type mice were pulsed with the essential amino acid lysine containing 6 stable ¹³C isotopes, relying on a balanced diet that is easily accepted by the animals. We then obtained hippocampal slices from the animals, and immunostained them for TNR and for calreticulin, a marker of the endoplasmic reticulum that enabled us to easily detect the cell bodies. After imaging the slices, we analyzed them using nanoscale secondary ion mass spectrometry (nanoSIMS). In nanoSIMS, a primary Cs⁺ beam irradiates the sample and causes the sputtering of secondary particles from the sample surface. These particles are partly ionized and are then identified by mass spectrometry. This

reveals the ^{13}C isotopes, which enabled us to test whether the TNR-containing spots consisted of newly-synthesized proteins (*i.e.* rich in ^{13}C isotopes), or whether they contained older proteins (poor in ^{13}C isotopes). As indicated in **Figure for Reviewers 2**, the intracellular (somatic) TNR spots were substantially older than the rest of the cell: older than the endoplasmic reticulum marked by calreticulin, and also older than cellular nuclei, which were detected directly in nanoSIMS. This implies that they are not newly synthesized, and therefore are most likely molecules that the cell has endocytosed from the ECM, in line with the model proposed by the rest of our work.

Third, we also sought to verify that TNR endocytosis is also seen in acute brain slices from rats, and that increased neuronal activity enhances the internalization of TNR. To test this, we applied a tagged, recombinant version of TNR to the slices, and followed its endocytosis by fluorescence imaging. This molecule, which is known to integrate into the ECM (Morawski et al., 2014), was indeed taken up by the cells, and its uptake was enhanced by silencing GABA_A receptors with bicuculline, in line with our general model (**Figure for Reviewers 3**).

Fourth, we were intrigued by the strong effects of dynamin inhibition on TNR recycling (also verified in response to comments from the Reviewer, see the new Extended Data Fig. 15). We therefore tested whether the two molecules interact, directly or indirectly, *in vivo*. We purified synaptosomes from adult rat brains, and immunoprecipitated TNR, relying on the same antibodies used in the rest of this work. This resulted in a measurable co-immunoprecipitation of dynamin (**Figure for Reviewers 4**). While this does not imply that the two molecules interact directly, this observation strongly suggests that the extracellular matrix component TNR is often in compartments that rely on dynamin-mediated membrane scission *in vivo* (*i.e.*, in trafficking compartments).

Figure for Reviewers 1. An analysis of intracellular TNR in acute brain slices from disease models. Hippocampal slices from kainic acid (KA)-induced epilepsy model mice and 5xFAD familial Alzheimer's disease model mice were immunostained with TNR antibodies and with the ER marker calreticulin, to reveal cell bodies clearly and to enable us to identify somatic intracellular TNR molecules. All other TNR was presumed to be extracellular (which is in line with the long lifetime and mainly extracellular location of TNR in the brain in general). The samples were then imaged with confocal microscopy. **a**, Imaged slice regions from mice pre-treated with a microinjection of vehicle (top panel) or KA (bottom panel). An analysis of the TNR fluorescence intensity of intra- or extracellular regions (normalized to the mean of the 'vehicle; extracellular TNR' condition) shows that the proportion of intracellular TNR molecules is increased in KA-treated mice. $N = 3$ mice for each experimental condition, with 60 and 67 regions analyzed for Vehicle and KA, respectively. Statistical significance was evaluated using the Kruskal-Wallis test ($H_3 = 27.93$, $***p < 0.001$), followed by the Dunn multiple comparisons test ('vehicle; extracellular TNR' vs. 'vehicle; intracellular TNR': $p = 0.932$; 'KA; extracellular TNR' vs. 'KA; intracellular TNR': $***p < 0.001$; 'vehicle; intracellular TNR' vs. 'KA; intracellular TNR': $**p = 0.004$). **b**, Imaged slice regions from wildtype (top panel) and 5xFAD mice (bottom panel). An analysis of the TNR fluorescence intensity of intra- or extracellular regions (normalized to the mean of the 'WT; extracellular TNR' condition) shows no significant differences. $N = 3$ mice for each experimental condition, with 68 and 29 regions analyzed for WT and 5xFAD, respectively. Statistical significance was evaluated using the Kruskal-Wallis test ($H_3 = 3.233$, $p = 0.357$). Boxplots show quartiles (boxes) and range (whiskers). Outliers were omitted according to inter-quartile range (IQR) proximity (exceeding $1.5 \times \text{IQR}$). This figure is shown in the manuscript as Fig. 10a,b.

Figure for Reviewers 2. Isotopic imaging suggests intracellular (somatic) TNR in adult mice is not newly synthesized. To accurately measure the turnover of TNR *in vivo*, we performed correlative fluorescence and isotopic imaging (Jähne et al., 2021; Kabatas et al., 2019; Saka et al., 2014a; Truckenbrodt et al., 2018; Vreja et al., 2015) in brain slices of mice pulsed with isotopically stable $^{13}\text{C}_6$ -lysine for 14 or 21 days (previously characterized in Fornasiero et al., 2018). We immunostained the slices for TNR and the ER marker calreticulin, embedded them in LR White, processed them to thin sections (200 nm), and then imaged these in epifluorescence microscopy. Using nanoSIMS, we then measured the enrichment of ^{13}C within the regions marked by TNR and calreticulin, to determine the local protein turnover. **a**, Top panel: a section stained for TNR and calreticulin, imaged in epifluorescence microscopy. Bottom panel: nanoSIMS images of $^{12}\text{C}^{14}\text{N}^-$ (left) and $^{13}\text{C}^{14}\text{N}^-$ (middle) secondary ions, respectively. The $^{13}\text{C}^{14}\text{N}^- / ^{12}\text{C}^{14}\text{N}^-$ ratio image (right) indicates the enrichment of ^{13}C , and thus the extent of protein turnover. Scale bar = 4 μm . **b**, Zoom of the square regions indicated in panel a. The circular ROIs, showing TNR or calreticulin-enriched areas, are indicated in the $^{13}\text{C}^{14}\text{N}^- / ^{12}\text{C}^{14}\text{N}^-$ ratio image. ^{13}C enrichment is lower in the TNR-enriched region. Scale bar = 500 nm. The regions chosen are in the immediate vicinity of the cell nucleus (visualized as a bright area in the $^{12}\text{C}^{14}\text{N}^-$ and $^{13}\text{C}^{14}\text{N}^-$ images), and are therefore intracellular. **c**, A quantification of the $^{13}\text{C}^{14}\text{N}^- / ^{12}\text{C}^{14}\text{N}^-$ ratio as fold over the natural abundance level, for nuclear, ER-enriched and TNR-enriched intracellular regions. The TNR-enriched areas exhibit the lowest ^{13}C enrichment in these cells. Statistical significance was evaluated using the Kruskal-Wallis test ($H_2 = 167.2$, $***p < 0.001$), followed by the Dunn multiple comparisons test ($***p < 0.001$ for the comparison between all groups, for both 14 and 21 day-long $^{13}\text{C}_6$ -lysine pulses). $N = 3$ mice for both the 14 and 21 day conditions, with 6 sections imaged per mouse. Boxplots show quartiles (boxes) and range (whiskers). Outliers were omitted according to inter-quartile range (IQR) proximity (exceeding $1.5 \times \text{IQR}$). This figure is shown in the manuscript as Fig. 10c-e.

Figure for Reviewers 3. His-tagged recombinant TNR is internalized in acute hippocampal slices from rats. To assess whether acute hippocampal slices are capable of internalizing TNR, we incubated slices from 3 week-old rats with His-tagged recombinant TNR (rTNR). Afterwards, the rTNR that remained at the surface of the slices was stripped by incubation with proteinase K. The slices were then fixed and immunostained with anti-His-tag antibodies (to reveal internalized rTNR) and anti-VGlut1 nanobodies (to reveal synapses), and were imaged with epifluorescence microscopy. We compared control slices to slices treated with bicuculline (40 μ M). To assess the total amount of rTNR and the background fluorescence, respectively, we omitted the incubation with proteinase K (top row) or the incubation with rTNR (bottom row). Scale bar = 15 μ m. We analyzed the pixel-by-pixel correlation between the rTNR and VGlut1 images as a measure for the presence of rTNR in synaptic regions. Unsurprisingly, high amounts of rTNR are found before stripping (as expected from the literature; Morawski et al., 2014). rTNR was also found in synaptic regions after stripping, and this was enhanced by bicuculline stimulation. N = 3 mice, with 20 images acquired per condition. Statistical significance was evaluated using the Kruskal-Wallis test ($H_3 = 179$, $***p < 0.001$), followed by the Dunn multiple comparisons test (all groups were significantly different from each other: $p = 0.034$ for the comparison between '-rTNR' and 'Untreated - stripped', and $***p < 0.001$ for all other comparisons). Boxplots show quartiles (boxes) and range (whiskers). Outliers were omitted according to inter-quartile range (IQR) proximity (exceeding $1.5 \times \text{IQR}$). This figure is shown in the manuscript as Extended Data Fig. 19.

Figure for Reviewers 4. Dynamin co-immunoprecipitates with TNR. We prepared synaptosomes from the cortices of 5 to 6-week-old rats, using a previously established protocol (Rizzoli et al., 2006). The synaptosomes were then subjected to a conventional immunoprecipitation procedure, using TNR antibodies. The left panel shows the immunoprecipitation of TNR, while the right panel shows a similar blot, revealing dynamin (using a pan-dynamin antibody). The experiment was performed 4 times, with similar results. The complete lanes are shown in the blots. The IP lanes correspond to 15% of the IP material. For the synaptosome lane, used as a positive control, only 0.05% of the material was run. Filled arrowheads show the bands of the antibodies using for the IP. This figure is shown in the manuscript as Extended Data Fig. 15h.

2. The authors suggest that TNR recycling occurs at synaptic sites and perturbing recycling disrupts SYT1-mediated vesicular release. To increase the functional significance of their findings, more in vitro experiments to assess synapse function and/or plasticity would strengthen these conclusions and increase the impact. This could be done by electrophysiology, calcium imaging, and/or assessment of structural synapse changes upon manipulation of TNR recycling.

We performed a series of experiments to address these points.

First, we perturbed TNR by tagging the recycling molecules with large antibody clusters, using an assay we described in the original manuscript (see also Richter et al., 2019). The addition of the antibody clusters perturbed synaptic vesicle exo- and endocytosis, as shown in the original manuscript. It did not affect the stimulus-induced rise in intracellular Ca^{2+} upon electrical stimulation, as observed by calcium imaging (**Figure for Reviewers 5a,b**). However, an interesting perturbation of neuronal activity became evident when we examined the spontaneous Ca^{2+} activity in the absence of stimulation. The addition of antibody aggregates raised substantially the spontaneous firing rate (**Figure for Reviewers 5c-e**), despite the inhibition of vesicular trafficking. As TNR has previously been shown to interact with voltage-gated Na^+ channels (Na_v s) (Xiao et al., 1999), this effect may be due to the TNR perturbation affecting Na_v activity, which may, in turn, increase the excitability of the neurons.

Second, we tested the effects of these aggregates on structural synapse changes, both in dissociated hippocampal cultures and in organotypic slices. Binding large antibody clusters to the newly-emerged TNR epitopes resulted in a significant reduction of spine head sizes both in culture and in slices (**Figure for**

Reviewers 6). As spine head size is an important reporter for synaptic strength (Humeau and Choquet, 2019), this implies that the TNR dynamics are linked to synaptic plasticity. Importantly, the antibody aggregates had no effect when they were attached to the surface-resident, non-recycling TNR epitopes.

This observation, which mirrors the effects we observed on synaptic vesicle exo- and endocytosis, allows us to conclude that the recycling TNR molecules are important in synaptic dynamics, both in the short term (*e.g.* vesicle dynamics) and in the long term (*e.g.* spine head plasticity).

Figure for Reviewers 5. Perturbing the recycling pool of TNR molecules affects spontaneous, but not evoked Ca^{2+} activity. To assess the effects of the perturbation of the TNR recycling pool on Ca^{2+} dynamics, we incubated dissociated hippocampal cultures with antibody aggregates that bound specifically to newly-emerged TNR. To test whether the TNR pool has an influence on Ca^{2+} dynamics, we measured the Ca^{2+} influx using a neuron-specific fluorescent indicator (red) in live neurons under basal conditions, or following a brief stimulation. **a**, The images show neurons under basal conditions, either at rest (left), or during bursting activity (right). Scale bar = 20 μm. **b**, The Ca^{2+} influx in response to a 1-second, 20-Hz stimulus is not affected by perturbing the TNR pool. N = 4 independent experiments, with 30 and 29 neurons imaged for the 'Ctrl' and 'Aggregates' conditions, respectively. Data show mean (lines) ± SEM (shaded regions). **c-e**, Typical traces from control and aggregate-treated neurons indicate that under basal conditions, perturbing the TNR pool increases the spontaneous firing rate and the total Ca^{2+} influx over time. N = 4 independent experiments, with 74 and 75 neurons imaged for the 'Ctrl' and 'Aggregates' conditions, respectively. Statistical significance was evaluated using Mann-Whitney U-tests, $U_{145} = 717.5$, $***p < 0.001$ and $U_{56} = 147.5$, $***p < 0.001$). Data represent the mean ± SEM. This figure is shown in the manuscript as Extended Data Fig. 16.

Figure for Reviewers 6. Perturbing the recycling pool of TNR molecules leads to a reduction in spine head size. To assess the effects of the perturbation of the TNR recycling pool on synapse structure, we incubated dissociated hippocampal cultures and organotypic hippocampal slice cultures with antibody aggregates for 12 hours, and visualized the plasma membranes with DiO (cultured neurons) or by infection with AAV vectors expressing eGFP under the synapsin 1 (hSyn) promoter (organotypic slices). We imaged individual spines in confocal microscopy, and then aligned these to the same orientation, in order to visualize the average spine under each treatment condition. Representative single spines are shown in the insets. The average spine in cultures in which newly-emerged, recycling TNR epitopes are bound to antibody aggregates appears smaller than the average spine in untreated cultures. This is not the case for cultures in which the non-recycling TNR epitopes ('all other epitopes') are targeted. This observation is confirmed by an analysis of the spine head area. Scale bar = 300 nm (dissociated cultures) and 500 nm (organotypic slices). $N = 3$ independent experiments, with > 80 synapses and > 60 synapses analyzed per condition for the measurements in cultured neurons and organotypic slices, respectively. Statistical significance was evaluated using one-way ANOVA (cultured neurons: $F_{2,6} = 5.269$, $*p = 0.05$) or repeated measures ANOVA (organotypic slices: $F_{1,041,2,083} = 20.76$, $*p = 0.042$), followed by Fisher's LSD test (cultured neurons: $**p = 0.005$ and $p = 0.418$; organotypic slices: $*p = 0.025$ and $p = 0.16$, for the comparisons between 'all other epitopes'/'new epitopes' and 'all other epitopes'/'Tyrode', respectively. Data represent the mean \pm SEM, with dots indicating individual experiments. This figure is shown in the manuscript as Fig. 9f,g.

3. More controls are warranted. To rule out the possibility of incomplete glutathione cleavage, the authors should include as a control a surface protein which is stable on the surface and is not endocytosed. Although appreciated that some controls are included to rule out non-recycled TNR degradation, the inclusion of another protein control that is endocytosed by not recycled back would bolster these findings.

We now include these controls. We analyzed the lysosomal marker LAMP1, as a molecule that is retrieved from the plasma membrane, but is not readily recycled, and myelin basic protein (MBP), as a molecule that is stable on the cellular plasma membrane. Both showed the expected behavior (**Figure for Reviewers 7**). LAMP1 was internalized, albeit not as extensively as synaptotagmin 1 or TNR, which is expected from LAMP's limited dynamics. Afterwards, no LAMP1 recycling could be detected. MBP was entirely lost by glutathione cleavage, indicating that it is not endocytosed under our experimental conditions.

Figure for Reviewers 7. The cell-surface biotinylation assay applied to non-recycling and non-endocytosing proteins. As additional controls, we repeated our cell-surface biotinylation assay for the lysosomal marker LAMP1, which is endocytosed, but undergoes little recycling, and myelin basic protein (MBP), which is not expected to endocytose. **a**, Example images of beads collecting the surface, endocytosed and non-recycled pools, immunostained for LAMP1 and MBP. Scale bar = 1 μm . **b**, An analysis of the fluorescence intensity of the beads, normalized to the mean of the 'surface' condition in the corresponding experiment. LAMP1 is detected in the endocytosed pool, and is present in similar amounts in the non-recycled pool, suggesting that it does not return to the plasma membrane. MBP is detected almost exclusively in the surface pool, which indicates that it is not endocytosed. N = 3 individual experiments, at least 50 beads imaged per experiment. Statistical significance was evaluated using repeated measures one-way ANOVA ($F_{1.293, 2.585} = 19.6$, $*p = 0.028$; $F_{1.52, 3.041} = 28337$, $***p < 0.001$), followed by the Holm-Sidak multiple comparisons test ($*p = 0.045$ and $p = 0.162$; $***p < 0.001$ and $p = 0.068$), for LAMP1 and MBP, respectively. Data represent the mean \pm SEM, with dots indicating individual experiments. This figure is shown in the manuscript as Fig. 1d,e.

4. In Extended Fig 1b, the authors show the percentage of total TNR that co-localizes with VGlut1 and VGAT. It would also be informative to indicate the percentage of VGlut1+ and VGAT+ puncta that are TNR positive.

We have measured this, from the same experiments shown in the original Extended Fig. 1b. The results, shown in **Figure for Reviewers 8**, indicate that 60% of the VGlut1-positive synapses contain TNR. This is also true for 39% of the VGAT-positive synapses.

Figure for Reviewers 8. A higher percentage of excitatory boutons colocalize with TNR, compared to inhibitory boutons. The images shown in the original Extended Fig. 1b were analyzed to determine the proportion of synapses that overlap with TNR signal. We found that the percentage of TNR+/VGlut1+ synapses (~60%) is significantly higher than the percentage of TNR+/VGAT+ synapses (~40%). N = 3 experiments, at least 10 neurons imaged per experiment. Statistical significance was evaluated using a paired *t*-test: $t = 3.312$, $**p = 0.003$). Boxplots show quartiles (boxes) and range (whiskers). Outliers were omitted according to inter-quartile range (IQR) proximity (exceeding $1.5 \times \text{IQR}$). This figure is shown in the manuscript as Extended Data Fig. 1d.

5. While there seems to be more recycled TNR at synapses compared to non-recycled, it doesn't seem from the images that the recycled TNR is preferentially at synapses vs. other parts of the neuron in Fig 2d or in Fig 3. The authors should clarify this in the text and also perform measurements to show that the recycled TNR is enriched at dendrites with an associated bouton vs. the rest of the dendrite or neuron.

We performed the required analysis, relying on the same dataset as in the original Fig. 2d. We averaged the TNR signals along dendrites or axons (labeled using the lipophilic tracer DiO), moving from the synapses (marked by VGlut1) towards the rest of the neurites (dendrites or axons). A substantial enrichment at synapses can be found for newly-emerged epitopes, which is much higher than that observed for all TNR epitopes (**Figure for Reviewers 9**). This indicates that newly-emerged TNR epitopes are indeed preferentially found at synapses.

Figure for Reviewers 9. New TNR epitopes are enriched at synapses. The images shown in the original Fig. 2d were analyzed to quantify the mean TNR signal in relation to distance from synapses (from the center of VGlut1-marked boutons), measured along dendrites or axons marked with DiO. Newly-emerged TNR epitopes are substantially more enriched in the synaptic area. The lines show means \pm SEM from 3 independent experiments, with more than 100 synapses analyzed for each experiment. The enrichment of newly-emerged epitopes in the synaptic region is significantly different from that of all epitopes (repeated measures ANOVA on log-transformed data: $F_{1.977, 3.954} = 24.13$, $**p = 0.006$, followed by Fisher's LSD test: $*p = 0.024$ and $*p = 0.036$ for the comparison of 'new epitopes' to 'all epitopes' at distance = 0, for dendrites and axons respectively). This figure is shown in the manuscript as Fig. 2e.

If recycled TNR is not highly enriched in the synapse and also exists at other places along the neuron, what would this mean for the neuron?

Newly-emerged TNR epitopes are also found along the neurites, although in this situation they are mainly endocytosed, and are therefore found in intracellular organelles. This idea was presented in the original Fig. 3. The organelles involved in recycling TNR molecules are now presented more extensively in **Figure for Reviewers 21** (shown in the manuscript as Fig. 6b-g).

For the “other” TNR epitopes, we found them to be less enriched at synapses, and to be far less dynamic, which implies that they may be structural elements that ensure neuronal stability.

6. Can the authors comment on the TNR shown outside the DiO-labeled cell (e.g. Fig 2d top right panel)? Is this secreted TNR?

We used a sparse DiO labeling here, implying that only a handful of cells are labeled on each coverslip. This enables us to analyze the neurites of individual cells very efficiently, but also leaves the neurites of other cells invisible. TNR outside of DiO-labeled cells is probably on the surfaces of, or endocytosed in, other cells in the culture, which are not labeled with DiO.

7. Fig 3c is very intriguing and a highlight of this paper. Inclusion of a protein that does not recycle and a synaptic protein such as Syt1 would be useful comparisons.

We have now included such proteins. This experiment requires good affinity tools for the extracellular epitopes of the molecules, which limits our choice. However, we were able to use Syt1, as a molecule that does recycle, and the EGF receptor, as a molecule that is endocytosed, but does not recycle readily. Both molecules were on the surface at “time 0”, and were endocytosed after 15 minutes. A proportion of the Syt1 molecules returned to the surface after another 45 minutes, while no EGF receptors returned (**Figure for Reviewers 10**).

The amount of Syt1 that resurfaces may appear small, at a first view. However, one needs to take into account that the surface pool of Syt1 molecules balances the recycling pool of vesicles in these synapses. The surface molecules are taken up by endocytosis, and are replaced on the surface by exocytosed vesicles from the recycling pool. The two pools of molecules intermix on a time scale of minutes (Truckenbrodt et al., 2018; Wienisch and Klingauf, 2006). This implies that we cannot expect all of the surface pool molecules to come back to the neuronal surface simultaneously. As the two pools are almost equal in size (Truckenbrodt et al., 2018), only ~50% of the initial surface molecules are expected to be back on the surface at any given time point, which is roughly the value we observed here.

This observation makes the highly synchronized dynamics of TNR even more remarkable, especially as they take place over longer time periods.

Figure for Reviewers 10. Measurement of Syt1 and EGF using the surface antibody stripping assay. As additional controls for the experiment shown in original Fig. 3c, we applied the same surface-stripping assay to two proteins: synaptotagmin 1 (Syt1), which is well known to undergo recycling, and the EGF receptor, which is endocytosed but does not recycle readily. We incubated the neurons with fluorophore-conjugated antibodies directed against the luminal domain of Syt1, or with fluorophore-conjugated EGF, and then imaged the neurons in epifluorescence before and after stripping the surface molecules with proteinase K. Immediately after the labeling, the staining was strongly reduced by the stripping, causing neurites to become virtually invisible for both proteins. After 15 minutes, the staining is similar before and after stripping, for both proteins, indicating that they have been endocytosed. Following a longer time period (60 minutes for Syt1; 4 hours for EGF, to account for any long-term recycling of this molecule), a loss of Syt1 from the neurites was observed after stripping, indicating that a portion of Syt1 had resurfaced. This was not the case in EGF-labeled cultures, indicating that they remained inside the cells. We quantified this by reporting the fluorescence ratio between the images taken before and after stripping (normalized to the first time point), as in the assay we devised initially for TNR. Scale bar = 10 μm . The graph demonstrates the diverging trends for the two proteins at the latest time point. $N = 3$ independent experiments, 5 images for each condition per data point. Statistical significance was evaluated using the Student's t -test ($t = 3.782$, $*p = 0.019$). Data represent the mean (lines) \pm SEM (shaded regions); dots indicate the individual experiments. This figure is shown in the manuscript as Extended Data Fig. 13.

8. The authors should test whether the cyclical surface recycling of TNR can be blocked by inhibition of dynamin with Dyngo (like in Extended Fig 7), either throughout the experiment or between days 2 and 3.

This is indeed the case. Dyngo application prevents the endocytosis of TNR, so that most of it is still on the surface at day 1. Some internalization is apparent at day 3, implying that the TNR endocytosis has been substantially slowed down (**Figure for Reviewers 11**).

Figure for Reviewers 11. The endocytosis of recycling TNR epitopes is slowed down by pharmacological inhibition of dynamin. We repeated the experiment shown in the original Fig. 3c in the presence of 30 μM Dyngo® 4a, to inhibit dynamin-dependent endocytosis. As expected, stripping significantly decreased the TNR staining immediately after labeling, indicating that the majority of the epitopes are present at the surface. However, this was also apparent at 1 and 3 days after labeling. Scale bar = 10 μm. The graph shows the fluorescence ratio between the images taken before and after stripping (normalized to the first time point). A similar ratio across all days suggests that TNR endocytosis is significantly slowed as a result of the drug treatment. N = 3 independent experiments, 5 images for each condition per data point. Statistical significance was evaluated using the Kruskal-Wallis test ($H_2 = 4.782$, $p = 0.104$). Data represent the mean (lines) ± SEM (shaded regions); dots indicate the individual experiments. This figure is shown in the manuscript as Extended Data Fig. 15c.

9. Could the authors speculate on why the TNR recycling is on a 3 day cycle in Fig. 3c and, perhaps, show some data supporting this? Could this give some functional insight?

To support this observation, we added a tagged, recombinant version of TNR to the sections. TNR added to preparations is known to integrate into the ECM (Morawski et al., 2014). We then followed recombinant TNR using a similar assay as in the previous two figures. The results, presented below in **Figure for Reviewers 26**, in reply to a specific question from Reviewer 2, indicate that this molecule recycles quantitatively on a 3-day cycle, with virtually all molecules coming back to the surface on the 3-day time point.

To provide functional insight in this direction, we examined the localization of the newly-emerged, and later internalized, TNR epitopes inside the cell. We then immunostained the cells for an assortment of intracellular targets, and searched for a colocalization with internalized TNR. We observed that a high proportion of the TNR molecules reached the Golgi apparatus, including dendritic Golgi outposts, as well as the endoplasmic reticulum (ER). The results are presented below in **Figure for Reviewers 21**, in reply to a specific question from Reviewer 2.

TNR recycling through the Golgi/ER appears rather unusual. Such a pathway would be needed, however, if the surface-exposed TNR molecules suffer modifications to their sugar moieties, and therefore require re-glycosylation, and/or a specific glycosylation pattern is needed to recycle perisynaptic TNR. This type of Golgi/ER recycling and re-glycosylation pathway has been less investigated than many other trafficking pathways, but has been demonstrated for several cell surface glycoproteins, especially in liver cells (Wratil et al., 2016). To test whether TNR follows such a pathway, we labeled newly O-glycosylated proteins by feeding the neurons with azide-modified galactosamine (GalNAz) and glucosamine (GlcNAz), which were

then revealed by tagging with a fluorophore, using a click chemistry reaction (Saka et al., 2014b). We found that the recycling TNR pool colocalized to a significant extent with GalNAz, and less with GlcNAz (**Figure for Reviewers 12**). This in agreement with previous studies that showed that GalNAc (but not GlcNAc) is a dominant component of O-linked carbohydrates on TNR (Woodworth et al., 2002, 2004; Zamze et al., 1999). As a control, we performed the same experiment by feeding the neurons with the methionine analogue azidohomoalanine (AHA), which incorporates into *de novo* synthesized proteins and is then similarly tagged using click chemistry (Dieterich et al., 2010), and we found little colocalization between TNR and AHA, in agreement with the expectation that this is a very long-lived protein (Dörrbaum et al., 2018).

The resolution of the imaging approach we used (two-color STED microscopy) is insufficient to demonstrate whether the recycling TNR molecules are themselves newly glycosylated, or whether their presence in the ER/Golgi compartments simply places them near newly-glycosylated proteins, resulting in the colocalization we measured optically. However, if the latter were true, the TNR molecules would also colocalize with newly-secreted proteins (labeled by AHA), which are abundant in the ER/Golgi compartments. As this was not the case, the overall interpretation of these experiments is that the recycling pool of TNRs consists of molecules that are not metabolically young, and that their trafficking to the ER/Golgi might function as means of re-glycosylation.

Overall, these experiments provide a good explanation for the long time period that TNR requires for recycling, albeit more data will be needed in the future to explore this pathway fully.

Figure for Reviewers 12. TNR recycling possibly relates to TNR re-glycosylation. We labeled newly O-glycosylated proteins by feeding the neurons with azide-modified galactosamine (GalNAz) and/or glucosamine (GlcNAz), which were then revealed by click chemistry. Alternatively, we labeled newly-synthesized proteins by feeding the neurons with azidohomoalanine (AHA), which was also tagged using click chemistry. Newly-emerged TNR epitopes were labeled as before and visualized at the surface. The samples were then imaged in 2-color STED microscopy. Scale bar = 1 μ m. The graph shows an analysis of the colocalization of the signals confirmed that internalized TNR epitopes colocalize significantly with GalNAz or GalNAz+GlcNAz, at levels substantially higher than the minimal AHA colocalization, which is not significantly different from negative controls. N = 4 independent experiments for each condition, at least 10 neurons imaged per data point. Statistical significance was evaluated using the Kruskal-Wallis test ($H_3 = 9.022$, $*p = 0.029$), followed by Fisher's LSD test, ($*p = 0.014$ and $*p = 0.021$ for 'GalNAz' and 'GalNAz+GlcNAz' respectively). Data represent the mean \pm SEM, with dots indicating individual experiments. This figure is shown in the manuscript as Fig. 8.

10. The ordering of figure panels in Fig 2 is confusing. Panels E and F should follow D. Otherwise, it is easy to overlook these panels.

We have performed these changes.

Reviewer #2:

Reviewer's comments relating to "Extracellular matrix remodelling through endocytosis and resurfacing of tenascin-R" by Dankovich et al. The manuscript by Dankovich and colleagues claims that ECM remodelling may not only occur consequent to degradation by proteases, but also by endocytosis and recycling of the scaffolding protein of the extracellular matrix (ECM) tenascin-R. The idea is interesting and the claim is novel.

We thank the Reviewer for the comments.

We replied to all of the Reviewer's comments, as detailed below. Importantly, we would like to thank the reviewer for pointing out that "a direct way to prove the claims of the paper would be to use purified TNR", added to the cultures. We have performed this experiment successfully, thereby strengthening our work substantially (please see our reply to comment #16). We have also replied to all other comments, as detailed below.

However, I am not fully convinced that the work presented supports this statement. I have several comments, as listed below:

1) Tnr is a soluble, secreted ECM molecule; for endocytosis it clearly requires a receptor system. The mechanistic basis of endocytosis is not explained in this study.

The Reviewer is right in pointing out that such a receptor system should be described here. In analyzing this issue, we started from the observation that, in addition to operating as a scaffold, the ECM actively regulates neuronal function by interacting with ECM receptors on the plasma membrane, such as the integrins, which link the ECM to the cell cytoskeleton (Kerrisk et al., 2014). A class of integrin receptors containing the $\beta 1$ subunit is particularly enriched at hippocampal synapses, where they mediate outcomes on dendritic spine motility and LTP development (Huang et al., 2006). A previous work suggests that TNR binds to $\beta 1$ -integrins via its EGFL and/or FN6-8 domains (Liao et al., 2008). Since the endocytosis and recycling of integrins are well-established phenomena (Clegg, 2003), we wondered whether the trafficking of the recycling TNR pool might be related to $\beta 1$ -integrins.

We first examined the colocalization of TNR with $\beta 1$ -integrin at two stages of its recycling pathway (**Figure for Reviewers 13**). First, we found that newly-emerged TNR epitopes colocalize with $\beta 1$ -integrin on the surface. Second, we found that the recycling TNR molecules often colocalized with internalized $\beta 1$ -integrins (**Figure for Reviewers 13**). The colocalization values were higher than those obtained with any other organelle or protein markers we employed (presented in **Figure for Reviewers 21**, below).

To then test integrin binding directly, we used an antibody that blocks $\beta 1$ -integrin (Shi and Sottile, 2008). This reduced profoundly the TNR internalization (**Figure for Reviewers 14**), suggesting that this molecule indeed serves as a receptor to TNR-containing ECM complexes.

Figure for Reviewers 13. TNR recycling is mediated by integrins. We tested the colocalization of the recycling TNR molecules with β 1-integrin. **a**, We labeled newly-emerged TNR epitopes 12 hours post-blocking by applying fluorophore-conjugated antibodies for 1 hour. At the same time, we labeled surface-bound β 1-integrins by applying fluorophore-conjugated antibodies directed against the extracellular domain of the receptors. We then fixed the neurons and imaged these with 2-color STED microscopy. **b**, We labeled newly-emerged TNR epitopes 4 hours post-blocking, concurrently with β 1-integrin. We incubated the neurons a further 12 hours to allow for internalization, and then stripped the surface-bound epitopes using proteinase K. We fixed the neurons and imaged these with confocal microscopy. The images on the right side of each panel show zoomed views of the dashed boxed. Scale bars = 2 μ m (full images) and 500 nm (zoomed images). **c-d**, A quantification of the % of colocalizing TNR signal (for panels a and b, respectively) showed that newly-emerged TNR epitopes colocalize with both cell surface-bound and internalized β 1-integrin receptors. The values are significantly higher than negatives controls, relying on non-specific primary antibodies (the negative controls were imaged in either STED or confocal microscopy for the comparisons to the images in panels a and b, respectively). N = 3 independent experiments for the 'surface β 1-integrin' experiments and the negative controls, and 4 independent experiments for 'internalized β 1-integrin', at least 10 neurons imaged per data point. Statistical significance was evaluated using the Student's *t*-test ('surface integrin' vs. 'neg ctrl': $t = 11.61$, $***p < 0.001$; 'internalized integrin' vs. 'neg ctrl': $t = 3.177$, $*p = 0.025$). Data represent the mean \pm SEM, with dots indicating individual experiments. This figure is shown in the manuscript as Fig. 7a,b.

Figure for Reviewers 14. Blocking β 1-integrin receptors significantly reduces TNR internalization. To directly assess whether β 1-integrin receptors are required for TNR endocytosis, we labeled newly-emerged TNR epitopes 12 hours after blocking, by applying fluorophore-conjugated TNR antibodies for 1 hour, and then immediately incubated the neurons with function-blocking anti- β 1-integrin antibodies for 6 hours. We stripped the neurons by incubation with proteinase K, to remove the surface-bound TNR and reveal the endocytosed molecules. We then fixed the neurons and imaged them with epifluorescence microscopy. The reduction in fluorescence signal is evident in the integrin-blocked cultures. Scale bar = 5

μm . An analysis of the fluorescence intensity confirmed that the amount of internalized TNR is significantly reduced following the blocking of $\beta 1$ -integrin receptors. $N = 3$ independent experiments, at least 15 neurons imaged per data point. Statistical significance was evaluated using the Student's t -test ($t = 3.343$, $*p = 0.029$). Data represent the mean \pm SEM, with dots indicating individual experiments. This figure is shown in the manuscript as Fig. 7c.

To obtain further insight into the endocytosis process, we also strengthened our observations on the involvement of dynamin. Please see comment 8 from Reviewer 1, and comment 3 from Reviewer 3.

Moreover, we have characterized organelles involved in the TNR recycling process (presented below in **Figure for Reviewers 21**, in reply to point #9 of the Reviewer).

Finally, we also provide a hypothesis for the potential role of the recycling process – see comment 9 of Reviewer 1 and **Figure for Reviewers 12**.

2) The study does not distinguish the different modes of Tnr in the extracellular matrix. Thus, Tnr is released into the environment by secretion and thereby part of the interstitial matrix of the CNS. It may bind to axonal surfaces by interaction with the IgSf member F3/F11/contactin. Moreover, it may be associated with perineuronal nets by interaction with other ECM components. So it is unclear which association with the neuronal surface is studied in this work. In fact no proof is given that uptake and recycling of Tnr holds true for the PNN-associated Tnr. To this end co-labelling with a PNN marker such as aggrecan would be required, for example in Fig. 3.

We have now tested whether the uptake and recycling of TNR holds true for the PNN-associated TNR. As indicated in **Figure for Reviewers 15**, this is indeed the case, albeit the proportion of newly-emerged epitopes is $\sim 30\%$ smaller for PNN-associated TNR.

Figure for Reviewers 15. TNR uptake and recycling occurs in both PNN and non-PNN-associated neurons. To assess whether the TNR dynamics we observed also hold true for PNN-associated neurons, we checked whether newly-emerged TNR epitopes could be observed in PNN-associated neurons (identified by a co-staining with WFA). We blocked surface epitopes with unlabeled antibodies against TNR, and then labeled newly-emerged TNR epitopes with fluorophore-conjugated TNR antibodies. We then fixed the neurons and incubated them with secondary anti-mouse antibodies, to label all TNR epitopes, and with WFA to label PNNs, before imaging them in epifluorescence microscopy. Newly-emerged TNR epitopes are visible in both PNN-associated (WFA+, top row) and non-PNN-associated (WFA-, bottom row) neurons. Scale bar = 5 μm . An analysis of the fluorescence ratio between “new” and

“all” TNR epitopes shows that the proportion of newly-emerged epitopes is ~30% smaller for PNN-associated neurons. Nevertheless, this experiment demonstrates that PNN- and non-PNN-associated neurons behave similarly. N = 3 independent experiments, at least 10 neurons imaged per data point. Statistical significance was assessed using a paired *t*-test ($t = 4.663$, $*p = 0.043$). Data represent the mean \pm SEM, with dots indicating individual experiments. This figure is shown in the manuscript as Extended Data Fig. 11.

3) *The observation that Tnr can be documented in PNNs in culture confirms earlier reports. However, there is no direct evidence that the PNNs are remodelled in this system nor that Tnr is pinched off the PNNs. It could as well be taken up from the medium, or axonal surfaces.*

Please see our reply to the previous comment. Ample remodeling of the PNN can be observed in this system.

4) *What is the ratio of PNN-wearing neurons in the culture system used for the study?*

This is documented below, in **Figure for Reviewers 16**. The value is ~11%, which is similar to previously reported numbers (e.g. ~12% in DIV14 mouse dissociated hippocampal cultures, Geissler et al., 2013).

Figure for Reviewers 16. A quantification of the ratio of PNN-associated neurons in dissociated hippocampal cultures. To determine the proportion of PNN-associated neurons at 14 DIV, we fixed neurons and stained them with WFA (magenta), to detect PNNs, and with NeuN antibodies (green), to label all neurons. We then imaged large fields of view with epifluorescence microscopy and counted the number of WFA+ neurons. We determined the proportion of PNN-associated neurons in our cultures to be ~11%. N = 3 individual experiments, 10 images per data point. Data represent the mean \pm SEM, with dots indicating individual experiments. This figure is shown in the manuscript as Extended Data Fig. 2e.

5) *Would the authors comment about the source of Tnr in their culture system? Is Tnr produced by the PNN-positive or -negative neurons, or both?*

We found TNR to be present on all neurons. As indicated in our **Figure for Reviewers 8**, 60% of the glutamatergic synapses and 39% of the GABAergic synapses contain TNR.

We calculated the % of neuronal TNR found in PNN-positive and negative neurons, from the experiment shown in **Figure for Reviewers 15**. While the individual PNN-positive neurons contain more TNR than individual PNN-negative neurons (~2-fold more), there are few such neurons in the cultures (**Figure for Reviewers 16**), implying that, on the whole, the PNN-positive population contains only ~23% of the neuronal TNR, with 77% present in PNN-negative neurons.

At the same time we would like to point out that the rate of production of TNR is not necessarily the most important parameter for our experiments, since it is extremely long-lived (our Extended Data Fig. 1; Dörrbaum et al., 2018; Fornasiero et al., 2018), and low amounts are produced at any one time.

6) In their model, the mouse neurons are cultured in close contact on a monolayer composed of astrocytes. The rat hippocampus is prepared from postnatal animals, at a stage when a significant number of glial cells, in particular astrocytes are already present. The latter can potentially engulf membranes or endocytose macromolecules. In the approach detailed in Fig. 1 the authors rely on a combination of pulse chase using glutathione and subsequent lysis and affinity beads to distinguish surface-associated, endocytosed and non-recycled Tnr. How can they be certain that they are seeing effects in neurons? Glial cell types and astrocytes are probably also part of their culture system.

To test this, we performed two experiments. First, we reproduced the main observation, the appearance of newly-emerged TNR epitopes and their endocytosis, in hippocampal cultures that are grown at a large physical distance above an astrocyte feeder layer (Banker cultures; Kaech and Banker, 2006). Ample TNR epitope emergence and subsequent endocytosis could be observed (**Figure for Reviewers 17**).

Second, we quantified the amounts of TNR in neurons and in astrocytes in the co-cultures used in the rest of our work. Astrocytes only account for ~5% of the internalized TNR, as presented in **Figure for Reviewers 18**.

Figure for Reviewers 17. TNR dynamics are observed in dissociated hippocampal ‘sandwich’ cultures. We repeated our ‘blocking-labeling’ assay in dissociated hippocampal sandwich cultures, in which the neurons are grown at a physical distance above an astrocyte feeder layer. We labeled newly-emerged TNR epitopes 12 hours post-blocking, by applying fluorophore-conjugated antibodies for 1 hour. We then incubated the neurons for an additional 6 hours, to allow for TNR internalization. We stripped the surface-bound epitopes by incubation with proteinase K, and then fixed the neurons and imaged them in epifluorescence microscopy. As controls, we labeled newly-emerged TNR epitopes without stripping the surface-bound molecules, and we also labeled all TNR epitopes by omitting the blocking steps. TNR could be detected in the stripped cultures, suggesting that these molecules are readily endocytosed by the neurons. Scale bar = 5 μm . An analysis of the mean fluorescence intensity (normalized to the ‘all epitopes’ condition) shows that a fraction of the newly-emerged TNR epitopes is internalized within 6 hours. To confirm that the internalization we observe is not due to unspecific uptake in these cultures, we incubated neurons with Atto647N-conjugated mouse secondary antibodies. This value was subtracted from the mean intensities shown in the plot. $N = 3$ independent experiments, at least 10 neurons imaged per data point. Statistical significance was evaluated using a paired t -test ($t = 8.258$, $*p = 0.014$). Data represent the mean \pm SEM, with dots indicating individual experiments. This figure is shown in the manuscript as Extended Data Fig. 12.

Figure for Reviewers 18. A minimal amount of internalized TNR is found in astrocytes. We checked whether recycling TNR epitopes could be detected within astrocytes by co-immunostaining cultures with the astrocytic marker GFAP and the neuronal marker NeuN, using an antibody that recognizes multiple isoforms of the RNA-binding protein FOX-3, localizing to both the nuclei and the cytoplasm (Lind et al., 2004). This staining reveals very strongly neuronal cell bodies and the main neurites of neurons, with only very low background in astrocytes. **a**, We labeled newly-emerged TNR epitopes 12 hours post-blocking, by applying fluorophore-conjugated antibodies for 1 hour. After a further incubation of 12 hours, to allow for internalization, we stripped the surface-bound epitopes using proteinase K. We then fixed the neurons and immunostained them with GFAP and NeuN to distinguish astrocytes and neurons, and imaged these with epifluorescence microscopy. The TNR signal was predominantly present in NeuN+/GFAP- cells, and was significantly weaker in GFAP+ cells. Scale bar = 20 μm. **b**, We quantified the TNR signal in NeuN+/GFAP- and GFAP+ regions of interest as a percentage of the total TNR staining in the images. We found that only ~5% of the internalized TNR could be detected in GFAP+ cells, confirming that this process occurs predominantly in neurons. N = 3 independent experiments, at least 10 images per data point. Statistical significance was evaluated using a paired *t*-test ($t = 29.49$, $**p = 0.001$). Data represent the mean \pm SEM, with dots indicating individual experiments. This figure is shown in the manuscript as Extended Data Fig. 3.

7) The culture system is not characterized using classical cell markers, such as GFAP and O4, the fraction of glial cells after the incubation period of up to 14 DIV was not determined.

We have now performed these experiments (**Figure for Reviewers 19**). Astrocytes are, as expected, abundant in these cultures, while oligodendrocytes are very rare. We observed no signals for microglia markers.

Figure for Reviewers 19. Characterization of the dissociated neuronal cultures used in the study.

To determine the fraction of different cell types in our cultures at 14 DIV, we performed a co-immunostaining with different cell-specific markers, and imaged large fields of view in epifluorescence microscopy. **a-c**, We counted cells overlapping with NeuN, GFAP (a), myelin basic protein (MBP) (b) or Iba1 (c) signals to quantify the number of neurons, astrocytes, oligodendrocytes and microglia, respectively. The cell nuclei were identified using a Hoechst staining. Scale bar = 50 μm. **d**, A quantification of the mean fraction of each cell type in the culture. N = 3 independent experiments. This figure is shown in the manuscript as Extended Data Fig. 2a-d.

8) The label in Fig 3a is interpreted as revealing intracellular Tnr; to prove that the label in fact targets intracellular vesicles electron microscopy combined with immunogold labelling would be required.

While we agree with the Reviewer that this would be an excellent experiment to test this hypothesis, we chose to apply here a much simpler assay, but which provides a very robust and unmistakable answer to

this question. We marked the newly-emerged TNR molecules using antibodies, we allowed them to endocytose, and we then applied LysoTracker to the neurons, which labels virtually all acidic organelles, including synaptic vesicles (as in our previous work, Tischbirek et al., 2012). In the live cells, we observed that ~70% of the TNR spots colocalized with the organelle marker, which provides ample evidence that these molecules had been endocytosed (**Figure for Reviewers 20**). The remaining ~30% of the molecules are probably found in non-acidified organelles, as the ER (see also **Figure for Reviewers 21**).

Figure for Reviewers 20. Internalized TNR is found in acidic organelles. To check whether TNR molecules are endocytosed in intracellular vesicles, we labeled newly-emerged TNR epitopes at 12 hours after blocking, by applying fluorophore-conjugated antibodies for 1 hour, concurrently with the application of LysoTracker™ Green, to label acidic organelles. We incubated the neurons a further 6 hours, to allow for internalization, and then stripped the remaining surface-bound TNR epitopes with proteinase K. We then imaged the neurons live using epifluorescence microscopy. Scale bar = 4 μm. A quantification of the colocalization between the two channels showed that > 70% of the internalized TNR is present in acidic organelles. N = 3 independent experiments, at least 4 neurons imaged per data point. Data represent the mean ± SEM, with dots indicating individual experiments. This figure is shown in the manuscript as Fig. 6a.

9) Alternatively, co-labelling with markers of intracellular vesicles could be attempted, for example markers of the endosome, or of circulating intracellular vesicles, e.g. selected rab proteins.

We immunostained the cells for an assortment of intracellular targets, and searched for a colocalization with internalized TNR. We observed that only a small quantity of TNR molecules was found in Rab5-positive early endosomes and Rab7- or Rab11-positive late or recycling endosomes (**Figure for Reviewers 21**). This, however, does not demonstrate that these organelles do not participate in TNR dynamics, since their relatively slow internalization (hours) implies that only a handful of molecules will be found, at any given time, in compartments involved in rapid molecule sorting, as these endosomes. More importantly, we found that a significant number of molecules colocalized with the Golgi apparatus, including dendritic Golgi outposts (**Figure for Reviewers 21**), and with the endoplasmic reticulum (ER). A proportion of the TNR molecules could also be found in LAMP1-positive organelles, which may represent lysosomes, albeit the slow degradation of TNR implies that this molecule typically escapes degradation in these organelles.

Figure for Reviewers 21. An overview of organelles involved in the trafficking of newly-emerged TNR epitopes. To identify the compartments containing the internalized TNR molecules, we labeled newly-emerged TNR epitopes at 12 hours after blocking, by applying fluorophore-conjugated antibodies for 1 hour. We incubated the neurons a further 6 hours, to allow for internalization, and then stripped the remaining surface-bound TNR epitopes with proteinase K. We then fixed the neurons and immunostained them with several organelle markers. **a-e**, 2 color-STED images of TNR (magenta) and organelle markers (green): caveolin1, Rab11a (recycling endosomes), LAMP1 (lysosomes), TGN38 (trans-Golgi network) and calreticulin (ER). The images on the right side of each panel show zoomed views of the dashed boxes. Colocalizing signals are indicated by the arrowheads. Scale bar = 2 μ m (full images) and 500 nm (zoomed images). **f**, A quantification of the % of TNR spots colocalizing with each marker, compared to a negative control relying on non-specific primary antibodies, showed that TNR colocalized significantly with ER and Golgi markers, as well as with LAMP1, Rab11a and caveolin. N = 3 independent experiments, at least 10 neurons imaged per data point. Statistical significance was evaluated using one-way ANOVA ($F_{8, 18} = 4.284$, $**p = 0.005$), followed by Fisher's LSD test for the comparison of all proteins to the 'Neg ctrl' condition (Caveolin1: $**p = 0.002$; Rab5: $p = 0.099$; Rab7: $p = 0.126$; Rab11a: $*p = 0.017$; Rab11b: $p = 0.169$; LAMP1: $**p = 0.005$; TGN38: $***p < 0.001$; Calreticulin: $***p < 0.001$). Data represent the mean \pm SEM, with dots indicating individual experiments. **g**, A fraction of the newly-emerged TNR localized to dendritic Golgi outposts after endocytosis. We labeled new TNR epitopes (magenta) 4 hours post-blocking and allowed them 12 hours to internalize. To identify dendritic Golgi outposts (Horton and Ehlers, 2003; Pierce et al., 2001) containing the internalized TNR molecules, we immunostained the neurons with the trans-Golgi network marker TGN38. Representative images, taken with a confocal microscope, are shown. Arrowheads denote an overlap between TNR and TGN38. Scale bar = 2 μ m. This figure is shown in the manuscript as Fig. 6b-h.

10) Does Tnr evade degradation in the lysosomes? The authors should show by Western blot developed with avidin peroxidase that the biotinylated Tnr is not (or is?) degraded in their system (which would be required for recycling integral Tnr).

The experiment proposed by the Reviewer is unfortunately impossible, since all surface molecules are biotinylated in this experiment, and not just TNR.

As indicated in our original Extended Data Fig. 1, the TNR antibodies show almost no decrease in signal over 4 days, indicated that this molecule is degraded very slowly. Similar experiments for molecules such as synaptotagmin 1 or VGAT showed losses of ~60% and ~80%, respectively, after 4 days (Truckenbrodt et al., 2018), again indicating that TNR is a remarkably stable protein.

However, to perform the experiment desired by the Reviewer, we added a tagged, recombinant version of TNR to the cultures (Morawski et al., 2014), and we imaged it at a 3-day interval. The result, presented in **Figure for Reviewers 22**, shows that no degradation could be detected. Please also see **Figure for Reviewers 26**, below, for more details on this experiment (including more images and controls).

Figure for Reviewers 22. Application of recombinant TNR in neuronal cultures shows no significant degradation over 3 days. We pulsed the neuronal cultures with a His-tagged recombinant TNR (rTNR), and fixed the neurons immediately after or following a 3-day incubation. We then immunostained the neurons with antibodies against His tag and imaged these with epifluorescence microscopy. Neurons show similar amounts of rTNR at day 0 and day 3 following the pulse. Scale bar = 20 μm . A quantification of the mean fluorescence intensity expressed as fold over background, normalized to the 'day 0' condition, confirmed that there is no significant degradation of rTNR after 3 days. N = 3 independent experiments, 5 images per data point. Statistical significance was evaluated using a paired *t*-test ($t = 1.126$, $p = 0.377$). Data represent the mean \pm SEM, with dots indicating individual experiments. This figure is shown in the manuscript as Extended Data Fig. 4c.

11) The study is based on the use of one monoclonal antibody to Tnr (clone 619 from synaptic Systems). In fact, they may see degradation fragments of Tnr that carry the epitope. There is no proof that the signal they obtain is due to integral Tnr (see point above).

There is no concern about the degradation of TNR. We have not found any evidence for TNR degradation in our antibody-based assays (Extended Data Fig. 4), or in the assay using the recombinant TNR (please see our answer to the previous comment). In addition, the biotinylation experiment, which the reviewer is referring to, was performed in the presence of lysosomal degradation inhibitors. Finally, all of these observations are in line with the long lifetime of TNR in cultured neurons, which has been thoroughly demonstrated in the past (Dörbaum et al., 2018).

12) The authors claim that they can achieve a 100% saturation of the epitope by adding the Mab clone 619 to Tnr and further that this saturation is stable over time. Therefore, they argue that label obtained with directly labelled 619 can only be explained by the resurfacing of heretofore hidden Tnr epitopes. This is very unlikely because it is highly improbable to achieve complete saturation of 100% of the epitope in the first place. Second, one could assume that the addition of directly labelled antibody would compete with bound antibody, simply because both probes recognize the same epitope. In that case, bound fluorescent probe would not necessarily reflect newly externalized Tnr, but rather the effect of competing off unlabelled antibody.

We have performed a large number of controls for our antibody work. We now present them in detail.

First, we found that the TNR antibodies do not separate from their epitopes in fixed cells at 37 °C, and that the labelled antibodies are unable to compete them off (**Figure for Reviewers 23**).

Second, we also found that the TNR antibodies do not separate from their epitopes in live cells, and no new epitopes appear, when the cell metabolism is reduced by keeping neurons at 4°C (**Figure for Reviewers 24**). Thus, again, the labelled antibodies were unable to compete off the bound antibodies.

Third, we could demonstrate that the labeled TNR molecules are not pre-existing extracellular epitopes that simply become detectable for antibodies through the cellular manipulations that we performed, and also that they are not the result of the cleavage of existing ECM structures (**Figure for Reviewers 25**). This implies that our blocking achieved close to 100% saturation.

We would also like to point out that we are very experienced with this type of experiment, and have performed it for other surface molecules in the past (e.g. Syt1 and VGAT, Truckenbrodt et al., 2018).

Figure for Reviewers 23. The TNR antibodies do not separate from their epitopes in fixed cells at 37 °C. a, We fixed neuronal cultures, and we then blocked their surface TNR epitopes with non-fluorescent antibodies. Immediately afterwards, or after 12 hours, we incubated the neurons with Atto647N-conjugated TNR antibodies. As a control, all TNR epitopes were labeled, by omitting the blocking step. The blocked cultures showed virtually no detectable fluorescence when imaged in epifluorescence microscopy (top panels). In the bottom panels, we enhanced the image contrast, to reveal the outlines of the cells. These are as bright in these images as in negative controls exposed only to Atto647N-conjugated secondary anti-mouse antibodies (leftmost panel). Scale bar = 10 μm. **b,** The analysis of the mean fluorescence intensity confirms that no new TNR epitopes emerge after 12 hours of incubation in fixed neurons, indicating that the

blocking antibodies persist on their epitopes, and do not allow the Atto647N-conjugated TNR antibodies to bind. N = 3 independent experiments, at least 10 neurons imaged per data point. Statistical significance was evaluated using one-way ANOVA ($F_{4, 10} = 120.3$, $***p < 0.001$), followed by the Holm-Sidak multiple comparisons test ($***p < 0.001$ for the comparison of each mean to the 'no blocking' condition). None of the other conditions were significantly different from the autofluorescence negative control. Data represent the mean \pm SEM, with dots indicating individual experiments. This figure is shown in the manuscript as Extended Data Fig. 8.

Figure for Reviewers 24. The TNR antibodies do not separate from their epitopes in live cells at 4 °C.

a, We labeled live neuronal cultures with Atto647N-conjugated TNR antibodies, and imaged them either immediately or following a 12 hour-long incubation at 4°C, in epifluorescence microscopy. Scale bar = 5 μm. **b**, An analysis of the mean fluorescence intensity shows that no significant change in staining is apparent after 12 hours at 4°C, indicating that the antibodies persist on their epitopes. N = 3 independent experiments, at least 10 neurons imaged per data point. Statistical significance was evaluated using a paired *t*-test ($t = 0.286$, $p = 0.802$). **c**, We blocked surface epitopes with unlabeled TNR antibodies, and fixed them either immediately or following a 12 hour-long incubation at 4°C. We then labeled the neurons with Atto647N-conjugated TNR antibodies, to reveal the available TNR epitopes, made available by the putative un-binding of blocking TNR antibodies, or with Atto647N-conjugated mouse secondary antibodies, to reveal the unlabeled blocking TNR antibodies. We then imaged the neurons in epifluorescence microscopy. Scale bar = 5 μm. **d**, An analysis of the fluorescence intensity shows that the staining with TNR antibodies is similar, and extremely low, both immediately and 12 hours after the blocking step. N = 3 independent experiments, at least 10 neurons imaged per data point). Statistical significance was evaluated using one-way ANOVA ($F_{2, 6} = 69.32$, $p^{***} < 0.001$) followed by the Holm-Sidak multiple comparisons test ($p = 0.945$ for the comparison between 'TNR 0 h' and 'TNR 12 h', and $p^{***} < 0.001$ for both comparisons to the 'anti-mouse' condition). Data represent the mean \pm SEM, with dots indicating individual experiments. This figure is shown in the manuscript as Extended Data Fig. 9.

Figure for Reviewers 25. The labeled TNRs are not pre-existing extracellular epitopes or the result of cleavage of existing ECM structures. **a**, To test whether the newly-emerged epitopes observed after TNR blocking could represent pre-existing epitopes that simply become available to antibody binding, we blocked surface epitopes with unlabeled antibodies, and then subjected them to several treatments that severely modify the cell surface. The treatments were fixation with 4% PFA or digestion of the surface glycosaminoglycans with chondroitinase ABC. We then added fluorophore-conjugated TNR antibodies to assess the number of epitopes that become available through these procedures. As positive and negative controls we used neurons incubated with TNR antibodies where we omitted the blocking step, or neurons incubated with anti-mouse secondary antibodies, respectively. We then imaged the neurons in epifluorescence microscopy. Scale bar = 10 μm . The graph (bottom panel) indicates that all conditions show substantially less fluorescence than the positive control, and are not distinguishable from the negative control. $N = 2$ independent experiments for the 'chABC' and 'No blocking' conditions, and 3 independent experiments for the 'neg ctrl' and 'PFA-fixed' conditions. At least 10 neurons were imaged per experiment. Statistical significance was evaluated using the Kruskal-Wallis test ($H_3 = 89.06$, $***p < 0.001$), followed by the Dunn multiple comparisons test for comparing each mean to the 'no blocking' condition ($***p < 0.001$). **b**, To test whether the newly-emerged TNRs could represent new epitopes that are exposed through the cleavage of existing ECM structures by secreted proteases, we blocked surface epitopes with unlabeled antibodies, and then treated the cultures with GM6001 to block the activity of matrix metalloproteinases (or with DMSO, as a control). We then added fluorophore-conjugated TNR antibodies to assess the amount of epitopes that become available, and imaged the neurons with epifluorescence microscopy. Scale bar = 10 μm . The graph (bottom panel) indicates that drug-treated cultures do not differ significantly from a negative control where neurons were incubated with anti-mouse secondary antibodies. $N = 3$ independent experiments, at least 10 neurons imaged per experiment. Statistical significance was evaluated using the Kruskal-Wallis test ($H_2 = 53.34$, $***p < 0.001$), followed by the Dunn multiple comparisons test for comparing the 'neg ctrl' condition to 'vehicle' and 'GM6001' ($***p < 0.001$), and 'vehicle' to 'GM6001' ($p = 0.95$). Data represent the mean \pm SEM. This figure is shown in the manuscript as Extended Data Fig. 10.

13) Along the same lines, each equilibrium reaction such as antibody binding has an off-rate. Therefore, one would expect a progressive release of antibody once the incubation with unlabelled antibody is terminated. The more so as in the culture system the medium represents a very large space for off-diffusion of the primary bound unlabelled antibody. In consequence, the bound labelled antibody may reflect an increasing number of free binding sites due to off-rate diffusion.

We agree with the Reviewer. However, the antibodies have two antigen-binding pockets, which will both be bound to TNR epitopes, since the TNR molecules operate as trimers, thereby offering multiple binding sites to each antibody. This increases profoundly the probability that the antibodies stay bound (avidity effect), leading to the situation that we observed in **Figures for Reviewers 23 to 25**, *i.e.* that antibodies do not come readily off their epitopes during the observation time.

Please also note that the Fab fragments of these antibodies also bind TNR very strongly, as discussed below, under comment #15.

14) Thus, the reduced intensity of label in the blocked situation could as well reflect the lower concentration of the labelled probe in relation to the blocking probe.

Our control experiments suggest that this is not the case. Please see our **Figures for Reviewers 23 to 25**.

15) In some case, the authors use Fab fragments for blocking the access to Tnr. It is well known that Fab fragments are monovalent and therefore of lower affinity than the divalent antibodies they are derived from.

The Reviewer probably does not refer to the affinity of the binding pocket for the antigen, since this is the same for antibodies and Fab fragments. The Reviewer probably refers to the fact that the Fab fragments do not benefit from the avidity effect that we mentioned above. We agree that this can be a huge problem for Fab fragments, and one that affects all monovalent immunostainings (as in our own work with nanobodies, *e.g.* Maidorn et al., 2019). However, we were lucky with our choice of antibody, and the resulting Fab fragments do not readily come off their targets (see the original Extended Data Fig. 6, currently Extended Data Fig. 14).

We have worked on aptamers, Fab fragments and nanobodies as replacement tools for antibodies for more than a decade, and we are fully aware of the effects mentioned by the Reviewer. At the same time, our extensive experience with such probes enabled us to find antibodies that perform far above the average antibody behavior, and which can therefore be used in these assays. A plethora of other antibodies we investigated cannot be used in such assays, simply since they bind non-specifically or unbind too readily from their targets. However, the TNR antibody used here is an excellent binder, and can indeed be used in these assays.

Thus directly labelled antibody should the more efficiently compete off the blocking reagent (see point 8).

We performed no experiments combining antibodies and Fab fragments, and therefore we do not see this as a problem.

16) In my view, the evidence of Tnr recycling is based in indirect evidence. A direct way to prove the claims of the paper would be to couple purified Tnr with biotin and add it to the culture system. Because the avidin-biotin interaction is of much (up to 1000-fold, 10-14) higher affinity than average antibody binding I trust that the itinerary of Tnr could reliably be traced using avidin probes.

We have performed the experiment desired by the Reviewer, without even needing to rely on biotin-avidin interactions. Our previous work has demonstrated that exogenous TNR supplied to the culture medium integrated in the ECM and promoted formation of aggrecan-based PNNs (Morawski et al., 2014). Here we used purified TNR that had a His-tag to enable its identification, at any desired time point, by immunostaining for the His tag. Please note that no live experiments are performed using antibodies – these are only added onto fixed and permeabilized cultures, to detect the His tag, which makes this experiment independent from the live use of any affinity probes.

His-tagged TNR bound well onto the cultures, as expected, and was more prominently seen on PNN-exhibiting cells, again as expected (**Figure for Reviewers 26a**).

At “time 0” it was found mostly on the cell surface, and it was mostly internalized after 1 day. The His-tagged TNR returned to the cell surface after 3 days, just as in the antibody-based experiments (**Figure for Reviewers 26b**).

Importantly, His-tagged TNR could also be used in organotypic slices, where we could demonstrate TNR endocytosis, and its dependence on neuronal activity (**Figure for Reviewers 27**).

Figure for Reviewers 26. His-tagged recombinant TNR applied to neuronal cultures recycles over 3 days. We assessed whether the TNR dynamics we observed could be reproduced in an experiment that does not require us to add antibodies to the live neurons, by relying on a recombinant His-tagged TNR (rTNR). **a**, rTNR added to live cultures shows a similar distribution to endogenous TNR. We pulsed neurons with rTNR for 1 hour, and then fixed the cells and stained them with WFA to label PNNs. We imaged the cultures with epifluorescence microscopy. Similarly to endogenous TNR, we observed that rTNR forms dense PNN-like structures at WFA+ neurons. Scale bar = 20 μ m. **b**, To check whether rTNR undergoes similar recycling to endogenous TNR, we performed the following experiment: (1) We pulsed neurons with rTNR for 1 hour, and then incubated the neurons for 0, 1 or 3 days (2), to allow for internalization and recycling of the rTNR. One group of neurons was immediately fixed (3), and the other was first incubated with proteinase K to remove all surface-bound rTNR (3'), and only then fixed. We permeabilized the neurons, and immunostained them with anti-His tag antibodies to reveal all rTNR in the culture (4), or only the internalized rTNR (4'). We then imaged these in epifluorescence microscopy and quantified the fluorescence ratio between the images of non-stripped and surface-stripped cultures, normalized to the first time point. **c**, The results from the experiment described in panel b. Immediately after the rTNR pulse, the staining was strongly reduced by the stripping, causing neurites to become less visible. After 1 day, a similar staining is

observed in stripped and non-stripped cultures, indicating that most rTNR molecules have been endocytosed. After 3 days, rTNR became visible in neurites once again, and decreased after stripping, indicating that a portion of the molecules had resurfaced. Scale bar = 10 μ m. These observations were confirmed by the quantification of the fluorescence ratio between non-stripped and stripped cultures. N = 3 independent experiments, 5 images for each condition per data point. Statistical significance was evaluated using repeated measures one-way ANOVA ($F_{1,044, 2.088} = 28,6$, $*p = 0.03$), followed by Fisher's LSD test ('day 0' vs. 'day 1': $**p = 0.002$; 'day 1' vs. 'day 3': $*p = 0.027$; 'day 0' vs. 'day 3': $p = 0.775$). Data represent the mean (lines) \pm SEM (shaded regions), with dots indicating individual experiments. This figure is shown in the manuscript as Fig. 5c-e.

Figure for Reviewers 27. His-tagged recombinant TNR is internalized in organotypic hippocampal slices cultures. We pulsed organotypic hippocampal slices with His-tagged recombinant TNR (rTNR) for 2 hours, and then incubated them a further 6 hours to allow for internalization. Afterwards, the rTNR that remained at the surface of the slices was stripped by incubation with proteinase K. The slices were then fixed and immunostained with anti-His-tag antibodies (to reveal internalized rTNR), and imaged with epifluorescence microscopy. We compared control slices to slices treated with bicuculline (40 μ M). To assess the total amount of rTNR and the background fluorescence, we omitted the incubation with proteinase K (no stripping) or the incubation with rTNR (- rTNR), respectively. Scale bar = 20 μ m. An analysis of the fluorescence intensity revealed that the amount of rTNR in stripped organotypic slices is significantly higher than the background fluorescence, suggesting that rTNR is internalized in these slices, and that this amount is increased in the presence of bicuculline. In the absence of proteinase K, the amount of rTNR is higher still, showing that a population of the molecules remains at the surface. N = 3 independent experiments, 15 images per condition. Statistical significance was evaluated using the Kruskal-Wallis test ($H_3 = 184.1$, $***p < 0.001$), followed by the Dunn multiple comparisons test (all groups were significantly different from each other: $***p < 0.001$, with the exception of the comparison between the 'No stripping' and 'Bic - stripped' conditions: $p > 0.999$). Boxplots show quartiles (boxes) and range (whiskers). Outliers were omitted according to inter-quartile range (IQR) proximity (exceeding $1.5 \times \text{IQR}$). This figure is shown in the manuscript as Extended Data Fig. 18c.

Reviewer #3:

In this short manuscript Dankovich et al report on the unexpected observation that the ECM molecule tenascin-R (TNR) and possibly other ECM factors are endocytosed over many years and recycling by primary CNS neurons, often near synapses. The very slow reappearance (days) of TNR molecules on the neuronal surface appeared to be facilitated by neuronal activity as suggested by pharmacological blockers or enhancers of GABAA or glutamate receptor function. Additional pharmacological experiments using the partially non-specific drugs Dyngo (to block dynamin), BFA (to block Golgi traffic) or monensin, a ionophore that alters Na/ H homeostasis, interfered with TNR recycling to the surface. Finally, the authors conduct experiments in which they add anti-TNR antibody aggregates to neurons and observe decreased activity-induced exo-endocytosis of the SV marker synaptotagmin 1. Based on these experiments in cultured neurons and a few experiments in slices, a model is proposed according to which the remodeling of synapses is accompanied by the endocytosis and recycling of ECM molecules. The observation that neurons are capable of slowly internalizing parts of the ECM is quite interesting and novel. The authors use a number of technical controls and variations of their assays ascertain that they are truly looking at TNR endocytosis and recycling. I concur with this conclusion.

However, the paper lacks compelling functional data and mechanistic insights into the endocytic recycling process. These would be expected for a publication at this level and to leave a lasting impact on the field. The present form of the Ms seems premature for publication.

We thank the Reviewer for the comments.

We would first like to point out that our intention was to introduce the concept of TNR endocytosis and recycling, which is a previously unknown mechanism for ECM dynamics. This view is fully supported by the statements of our other three reviewers:

- Reviewer 1: *These findings are very interesting and **challenge existing assumptions** about ECM plasticity and the role of ECM in synaptic plasticity.*
- Reviewer 2: *The idea is interesting and **the claim is novel**.*
- Reviewer 4: *This is an interesting topic and a well-designed study, which **sheds new light on the dynamics of extracellular matrix** components.*

Nevertheless, we agree with the Reviewer that adding more mechanistic insight would be important. We therefore performed many new experiments, which fully support the original model that we proposed, and also offer new insight into the following:

- We describe β 1-integrin as a potential receptor for TNR recycling (**Figures for Reviewers 13 and 14**)
- We describe organelles that participate in TNR recycling (**Figures for Reviewers 20 and 21**)
- We verify the involvement of dynamin in TNR dynamics (**Figures for Reviewers 11 and 28**)
- We provide a clue for the potential need for TNR endocytosis and recycling, by pointing out that it may become re-glycosylated during recycling (**Figure for Reviewers 12**)
- We show that newly-emerged TNR integrates into the ECM at synapses (**Figure for Reviewers 31**).

We hope that this is sufficient to enable the publication of our work in Nature Communications.

Please also find below our replies to all other points raised by the Reviewer.

Specific questions to be considered before submission elsewhere:

1. It remains unclear how neuronal activity controls TNR endocytosis and/ or recycling and at what level. Such information is required to understand the physiological relevance of the process.

We now describe β 1-integrin as a potential receptor for TNR recycling (**Figures for Reviewers 13 and 14**).

It is well known that, in addition to operating as a scaffold, the ECM actively regulates neuronal function by interacting with ECM receptors on the plasma membrane, such as the integrins, which link the ECM to the cell cytoskeleton (Kerrisk et al., 2014). A class of integrin receptors containing the $\beta 1$ subunit is particularly enriched at hippocampal synapses, where they mediate outcomes on dendritic spine motility and LTP development (Huang et al., 2006). A previous work suggests that TNR binds to $\beta 1$ -integrins via its EGFL and/or FN6-8 domains (Liao et al., 2008). Since the endocytosis and recycling of integrins are well-established phenomena (Clegg, 2003), we feel that our experiments on $\beta 1$ -integrin provide a strong link between TNR recycling and synapse dynamics, explaining how TNR recycling may be controlled by synapses.

Further hypotheses connected to the interactions between TNR, integrins and the presynaptic active zone could be tested in the future (Cohen et al., 2000; Wang et al., 2018). One such possibility is a perturbation of the interaction between integrins, which are known to bind TNR, and presynaptic laminins. Specific laminin isoforms have also been shown to interact with Synaptic Vesicle Protein 2 (SV2), which plays a role in priming synaptic vesicles for Ca^{2+} -triggered release (Chang and Sudhof, 2009; Son et al., 2000).

Overall, we feel that our observations (**Figures for Reviewers 13 and 14**), together with the current knowledge on the relation between integrins and TNR, show a clear path as to how TNR endocytosis could be controlled, through the involvement of integrins. However, the detailed control of the newly-discovered TNR recycling pathway is beyond the efforts of a first publication on this subject, as hopefully the Reviewer agrees.

Especially, experiments aimed at addressing the claimed function of TNR recycling for synapse remodeling in vitro and, ideally, in functional networks in vivo would be required to raise this study to the level expected from a Nat Commun paper.

In **Figure for Reviewers 5** we present experiments that demonstrate that the perturbation of the recycling TNR molecules affects synaptic remodeling strongly. This effect was observed both in dissociated cultures and in organotypic slices. See also effects of TNR perturbation on Ca^{2+} dynamics in **Figure for Reviewers 5**.

Moreover, significant amounts of TNR molecules can be found intracellularly *in vivo*, and their abundance is affected by the activity of the cells (**Figure for Reviewers 1**). These molecules are not newly synthesized, but are old, presumably recycling molecules (**Figure for Reviewers 2**). Moreover, TNR appears to interact with dynamin, a prominent trafficking molecule, *in vivo*, either directly or indirectly (**Figure for Reviewers 4**).

Please see the respective sections above.

2. Antibody aggregates may elicit non-specific effects on neuronal signaling and, hence, as the authors admit themselves can only serve as a crude door-opener to address the physiological relevance of ECM exo-endocytosis and recycling. More specific manipulations that would require additional mechanistic insight into the uptake and recycling pathway, e.g. using endocytosis defective TNR mutants or its so far unidentified cellular receptor would be a way forward (see also #3).

We agree with the reviewer that antibody aggregates serve as a crude, “brute force” approach to analyze the TNR effects. Nevertheless, they are not unspecific manipulations. The addition of the aggregates on the newly-emerged TNR molecules has profound effects, while their addition on the non-recycling epitopes has no significant effects (**Fig. 9 and Figure for Reviewers 6**). This implies that these applications are remarkably specific in their effects.

At the same time, as the Reviewer surely agrees, it is impossible to find endocytosis-defective mutants for a molecule that was never before described to endocytose, unless we start a major effort that goes far beyond the scope of this manuscript.

Nevertheless, we did find that β 1-integrin acts as a receptor for TNR recycling. Please see our reply to point #1.

3. No information is provided as to the molecular machinery that mediates TNR recycling,

We now describe organelles that participate in TNR recycling. Please see **Figure for Reviewers 21**.

apart from a possible function for dynamin that would need to be substantiated by genetic tools such as available dynamin 1/3 KO mice and neurons or at the very least lentiviral KD of dynamin.

We knocked down dynamin in the neuronal cultures, using a construct described previously by the Haucke laboratory (Kononenko et al., 2014). This reduced substantially the amounts of internalized TNR (**Figure for Reviewers 28**).

Figure for Reviewers 28. The endocytosis of recycling TNR epitopes is slowed down by knockdown of dynamin. We performed a triple knockdown of dynamin 1, 2 and 3 using previously described siRNA constructs (Kononenko et al., 2014) and assessed the effects on TNR dynamics. **a**, We immunostained control and siRNA-treated neurons with antibodies against dynamin and imaged the cultures with epifluorescence microscopy. The fluorescence signal was visibly reduced in siRNA-treated neurons, demonstrating the effectiveness of the knockdown. Scale bar = 25 μ m. **b**, A quantification of the mean

fluorescence intensity shows that the knockdowns reduced the amount of dynamin by ~ 50%. N = 3 independent experiments, 10 images per data point. Statistical significance was assessed using the Student's *t*-test ($t = 5.259$, $**p = 0.006$). **c**, We labeled newly-emerged TNR epitopes by applying fluorophore-conjugated antibodies 12 hours post-blocking, and then incubated the neurons a further 6 hours to allow for endocytosis. One group of neurons was fixed immediately after, and the other was subjected to a treatment with proteinase K to remove surface-bound TNR epitopes before stripping. **a**, The fluorescence signal was not visibly different when labeling the newly-emerged TNR epitopes in control vs. siRNA-treated cultures (top row). However, the signal when labeling internalized TNR epitopes (following stripping) is significantly reduced in siRNA-treated neurons. Scale bar = 4 μ m. **d**, An analysis of the fluorescence intensity confirms that siRNA knockdown of dynamin significantly reduces the internalization of recycling TNR epitopes after 6 hours. The internalized amount in siRNA-treated neurons is not significantly different from the fluorescence background, determined by measuring internalized secondary antibodies in these neurons. N = 3 independent experiments, 10 images per data point. Statistical significance was evaluated using a one-way ANOVA on log-transformed data ($F_{4,10} = 38.22$, $***p < 0.001$), followed by Tukey's multiple comparisons test (New epitopes 'ctrl' vs. 'siRNA': $p = 0.86$; New epitopes stripped 'ctrl' vs. 'siRNA': $**p = 0.004$; New epitopes stripped 'siRNA' vs. 'autofluorescence': $p = 0.832$). Data represent the mean \pm SEM, with dots indicating individual experiments. This figure is shown in the manuscript as Extended Data Fig. 15d-g.

4. Equally important, it remains undisclosed if and how altered activity in the culture system affects the structure and composition of the ECM that surrounds synapses in the culture system. Such effects would be expected if network activity controls TNR endocytosis and/ or recycling. Such changes could also be looked at in organotypic slice cultures for example.

We have performed the following analyses:

- An analysis of TNR amounts and organization around synapses in relation to synaptic strength, in dissociated cultures (the original Extended Data Fig. 4, now Fig. 3). This showed that active synapses are associated with larger amounts of newly-recycled TNR in their ECM.
- An analysis of the TNR matrix at synapses in dissociated cultures (**Figure for Reviewers 31**), which showed that newly-emerged epitopes are found preferentially at synapses. Importantly, blocking synaptic activity removed this trend, indicating that the addition of recycling TNR molecules at synapses depends on their activity.
- An analysis of the effects of perturbing TNR recycling on synapses in both dissociated cultures and in organotypic slices (**Figure for Reviewers 6**). TNR perturbation affected spine size, and thus synaptic strength.
- Analyses of TNR incorporation in organotypic slices, including slices stimulated by the use of bicuculline, based on the use of antibodies, as in the rest of this manuscript, and on the use of recombinant TNR (**Figures for Reviewers 27 and 30**). The activated slices showed stronger TNR dynamics, as expected from the rest of our work.
- Interestingly, we found that individual neurons had different total levels of TNR epitopes, and that the total TNR levels were similar in the dendrites and the axonal branches of one identified neuron (**Figure for Reviewers 29**, below). These observations are in good agreement with the literature (Brückner et al., 2003; Dityatev et al., 2007). In contrast, there was no such correlation when the newly-emerged TNR epitopes were considered. The dendrites and axonal branches of individual neurons had widely variable fluorescence levels (**Figures for Reviewers 29**). This suggests that the amounts of newly-emerged TNR epitopes are not controlled by the identity or the metabolism of individual neurons, but rather by the local activity conditions of the synapses, as already indicated in the original Extended Data Fig. 4.

Overall, we hope that these experiments address the concerns of the Reviewer.

Figure for Reviewers 29. The total TNR load of individual neurons is similar for their axons and dendrites, but the load of newly-emerged epitopes is not. We stained individual neurons by incubating the cultures with low levels of DiO crystals (green), which label the plasma membranes of only a few neurons on each coverslip. We then labeled the TNR epitopes (magenta) and the VGlut1-positive boutons (blue) as in Figure 1, and we analyzed the average TNR intensity in the dendrites and the axons of individual neurons. Highly branching neurites with clear spine-shaped protrusions were classified as dendrites, whereas thin, continuous neurites with a substantial number of overlapping VGlut1 puncta were classified as axons. **a**, All TNR molecules in the ECM were labeled. A significant correlation is found between the axonal and dendritic TNR loads. N= 3 independent experiments, with 17 individual neurons imaged; Spearman's Rho = 0.745, *** $p < 0.001$). **b**, Only the newly-emerged TNR epitopes were labeled. No significant correlation could be determined. N = 3 independent experiments, with 22 individual neurons imaged; Spearman's Rho = 0.237, $\rho = 0.289$). This figure is shown in the manuscript as Extended Data Fig. 7.

Reviewer #4:

In this study, Dankovich and collaborators investigated the trafficking of Tenascin-R protein in cultured hippocampal neurons. Using several approaches, the authors show that Tenascin-R recycle between the plasma membrane and intracellular pool. Such a cycle will take up to 3 days and is regulated in an activity-dependent manner. This is an interesting topic and a well-designed study, which shed new light on the dynamics of extracellular matrix components. Yet, the study appears to be somehow preliminary and additional experiments are needed to strengthen the conclusion.

We thank the Reviewer for the comments. We performed all of the required experiments, as detailed below.

i) The authors state that recycled TNR molecules are enriched near synapses (e.g. lines 80-84). Yet, the STED illustrations show that TNR are indeed in the proximity of synapses as well as all along the dendritic shaft. The main claim that TNR are specifically trafficked near synapses appear thus overstated. How the authors exactly measured the % of TNR at synapses (Fig. 2e)?

We apologize for the potential confusion here. TNR is present both at synapses and elsewhere, with ~5% of TNR present at GABAergic synapses, and ~8% at glutamatergic synapses (our original Extended Data Fig. 1b). Figure 2e does not show the % of TNR at synapses (since this was presented in the Extended Data Fig. 1b), but the % of synaptic TNR that turns over in 12 hours. This was estimated as follows: we measured the intensity of “total TNR” in synapses, at the synapse ROIs identified by specific immunostaining in Fig. 2. We then measured the “new TNR” in synapses in a similar fashion, in an experiment in which the TNR antibody was applied at 12 hours after a surface blocking procedure. We then calculated the % of the “all TNR” signal that was made up by the “new TNR” signal. We now clarify this in our figure legend.

In addition, as the authors use a cultured system in which astrocytes and neurons are intermingled the exact relevance of the dendritic/spine TNR staining is unclear. Is this influenced by the presence of a nearby glial cells?

To test this, we reproduced the main observation, the appearance of newly-emerged TNR epitopes and their endocytosis, in hippocampal cultures that are grown at a large physical distance above an astrocyte feeder layer (Banker cultures; Kaech and Banker, 2006). Ample epitope appearance and endocytosis could be observed (see **Figure for Reviewers 17**, above).

Additional experiments and analysis describing the whole distribution of TNR along dendrites should be provided.

We have now performed this analysis. Please see **Figure for Reviewers 9**, above.

ii) Interpretation of the data from cultured slices is difficult and thus not convincing. What is exactly the staining observed and how specific it is remain not entirely clear. In addition, one may eventually expect difference between hippocampal layer. The illustrations show different locations within the hippocampus, which further complicate an understanding of the experiment. Clarifications and additional controls would be appreciated.

We agree with the reviewer that the quality of our initial slice experiments was not optimal. We have now performed new experiments, which exhibit a much higher imaging quality. Please see the results in **Figure for Reviewers 30**. We now also include a selection of images of whole slices.

However, we included in our analysis a large selection of random regions from the entire slices, rather than attempting a region-specific investigation.

In addition, we also used a purified TNR version that has a His-tag, and exogenous TNR has been previously demonstrated to integrate into the ECM (Morawski et al., 2014). His-tagged TNR was endocytosed in organotypic slices, in a fashion dependent on neuronal activity (**Figure for Reviewers 27**).

Figure for Reviewers 30. TNR dynamics can also be observed in cultured slices. **a**, Exemplary large field-of-view images of organotypic hippocampal cultured slices, fixed and immunostained for TNR. Scale bar = 50 μm . **b**, To test whether Fab fragments can be used to block TNR surface epitopes in a model that is closer to the *in vivo* morphology of neuronal tissue, we applied our ‘blocking-labeling’ assay to organotypic hippocampal slice cultures. We first blocked TNR surface epitopes by incubating live slices with Fab fragments directed against TNR, applied together with non-fluorescent anti-mouse nanobodies, for 2 hours. Newly-emerged TNR epitopes were labeled with new Fab fragments directed against TNR, applied together with fluorophore-conjugated anti-mouse nanobodies, 12 hours post-blocking. We compared control cultures with cultures treated with bicuculline (40 μM) or with a combination of CNQX (10 μM) and AP5 (50 μM). To test the efficacy of the blocking procedure, we also labeled the newly-emerged TNR epitopes immediately after the blocking step. The blocked slices (‘Block + 0 h’) showed little fluorescence when imaged in confocal microscopy, when compared with a full surface labeling (‘No blocking’). Newly-emerged epitopes can be detected 12 hours post-blocking, and their amounts are visibly higher in the presence of bicuculline, and lower in the presence of CNQX + AP5. Scale bar = 20 μm . An analysis of the mean fluorescence intensity confirms that the amount of newly-emerged TNR epitopes is significantly lower following CNQX + AP5

treatment, and significantly higher following bicuculline treatment. N = 3 independent experiments, 10 images for each condition per experiment. Statistical significance was evaluated using one-way ANOVA ($F_{4, 266} = 115.8$, $***p < 0.001$), followed by Tukey's multiple comparisons test (the comparison between all groups is significant: $***p < 0.001$, with the exception of the comparison between 'Blocked + 12 h + Bic' and 'No blocking': $p = 0.092$). Boxplots show quartiles (boxes) and range (whiskers). Outliers were omitted according to inter-quartile range (IQR) proximity (exceeding $1.5 \times \text{IQR}$). This figure is shown in the manuscript as Extended Data Fig. 18a,b.

iii) Beside these experimental caveats, it remains unclear whether the recycled TNRs after days within neurons are still biologically active or, at least, have the capacity to embed with the ECM. This piece of information will be of great interest for the study.

Interfering with recycling TNR affects presynaptic exo- and endocytosis (original Extended Data Fig. 8), and also affects spine head size, and thereby synaptic strength (**Figure for Reviewers 7**). Importantly, these effects are not reproduced when we interfere with non-recycling TNR molecules, indicating that the recycling ones have a critical and specific role at synapses.

To test the question of the Reviewer directly, we analyzed the location of newly-recycled and older TNR epitopes at synapses, using multicolor super-resolution imaging. We found that the recycling TNR molecules integrate into the perisynaptic ECM, and that this happens preferentially at the spine head (**Figure for Reviewers 31**).

Figure for Reviewers 31. New TNR epitopes integrate into the perisynaptic ECM, preferentially near the spine head. To test whether the recycling TNR molecules integrate into the perisynaptic ECM, we performed 2-color STED super-resolution imaging, to compare the stable and newly-emerged TNR epitopes at the same synapses. In addition, we checked whether the organization of the newly-emerged epitopes is dependent on synaptic activity, by comparing control cultures with cultures treated with bicuculline (40 μ M) or with a combination of CNQX (10 μ M) and AP5 (50 μ M). **a**, An exemplary image of a dendrite. The plasma membrane was visualized by incubating the neurons with DiO. We performed the same ‘blocking-labeling’ assay as in the rest of the study, but used fluorophore-conjugated (rather than unlabeled) antibodies against TNR for the blocking step. “Old” and newly-emerged epitopes are typically in the vicinity of each other, albeit

they do not correlate strongly, as already suggested in Fig. 2. Scale bar = 1 μm . **b**, We imaged many dendrites for each treatment condition and extracted segments containing individual spines. We then aligned these to the same orientation, in order to visualize the average spine under each treatment condition. This visualization showed that the newly-emerged epitopes embed in the ECM, and are especially visible in the vicinity of the spine head. This tendency to localize to the spine head was increased when activity was enhanced with bicuculline, and abolished when activity was reduced with CNQX+AP5. Scale bar = 500 nm. **c**, A quantification of percentage of new TNR epitopes colocalizing with old epitopes showed that the majority of the newly-emerged molecules colocalize with existing molecules, confirming they integrate into the perisynaptic ECM. The remaining epitopes presumably represent endocytosed TNR, found within the dendrites. **d**, To test our observation that the new TNR epitopes preferentially localize to the spine head (panel b), we calculated the fluorescence ratio between the spine head and the dendritic shaft at its base, for both the old and the new epitopes. This analysis confirmed that the newly-emerged TNR epitopes preferentially appear in the spine head, whereas the old epitopes are distributed in both the head and the shaft. As we observed in the average images (b), this bias was slightly increased in bicuculline-treated cultures, and was lost in CNQX+AP5-treated cultures. $N = 3$ independent experiments, with >80 synapses analyzed per condition. Statistical significance was evaluated using a Wilcoxon signed rank test ('Ctrl': $W = -1782$, $***p < 0.001$; 'Bic': $W = -3633$, $***p < 0.001$; 'CNQX+AP5': $W = -1313$, $p = 0.159$). Boxplots show quartiles (boxes) and range (whiskers). Outliers were omitted according to inter-quartile range (IQR) proximity (exceeding $1.5 \cdot \text{IQR}$). This figure is shown in the manuscript as Extended Data Fig. 17.

References

- Broekaart, D.W.M., Bertran, A., Jia, S., Korotkov, A., Senkov, O., Bongaarts, A., Mills, J.D., Anink, J.J., Seco, J., Baayen, J.C., et al. (2021). The matrix metalloproteinase inhibitor IPR-179 has antiseizure and antiepileptogenic effects. *J. Clin. Invest.* 131.
- Brückner, G., Grosche, J., Hartlage-Rübsamen, M., Schmidt, S., and Schachner, M. (2003). Region and lamina-specific distribution of extracellular matrix proteoglycans, hyaluronan and tenascin-R in the mouse hippocampal formation. *J. Chem. Neuroanat.* 26, 37–50.
- Chang, W.-P., and Sudhof, T.C. (2009). SV2 Renders Primed Synaptic Vesicles Competent for Ca²⁺-Induced Exocytosis. *J. Neurosci.* 29, 883–897.
- Clegg, D.O. (2003). Integrins in the development function and dysfunction of the nervous system. *Front. Biosci.* 8, 1020.
- Cohen, M.W., Hoffstrom, B.G., and DeSimone, D.W. (2000). Active Zones on Motor Nerve Terminals Contain $\alpha 3\beta 1$ Integrin. *J. Neurosci.* 20, 4912–4921.
- Dieterich, D.C., Hodas, J.J.L.L., Gouzer, G., Shadrin, I.Y., Ngo, J.T., Triller, A., Tirrell, D.A., and Schuman, E.M. (2010). In situ visualization and dynamics of newly synthesized proteins in rat hippocampal neurons. *Nat. Neurosci.* 13, 897–905.
- Dityatev, A., Brückner, G., Dityateva, G., Grosche, J., Kleene, R., and Schachner, M. (2007). Activity-dependent formation and functions of chondroitin sulfate-rich extracellular matrix of perineuronal nets. *Dev. Neurobiol.* 67, 570–588.
- Dörrbaum, A.R., Kochen, L., Langer, J.D., and Schuman, E.M. (2018). Local and global influences on protein turnover in neurons and glia. *Elife* 7.
- Fornasiero, E.F., Mandad, S., Wildhagen, H., Alevra, M., Rammner, B., Keihani, S., Opazo, F., Urban, I., Ischebeck, T., Sakib, M.S., et al. (2018). Precisely measured protein lifetimes in the mouse brain reveal differences across tissues and subcellular fractions. *Nat. Commun.* 9, 4230.
- Geissler, M., Gottschling, C., Aguado, A., Rauch, U., Wetzel, C.H., Hatt, H., and Faissner, A. (2013). Primary Hippocampal Neurons, Which Lack Four Crucial Extracellular Matrix Molecules, Display Abnormalities of Synaptic Structure and Function and Severe Deficits in Perineuronal Net Formation. *J. Neurosci.* 33, 7742–7755.
- Horton, A.C., and Ehlers, M.D. (2003). Dual Modes of Endoplasmic Reticulum-to-Golgi Transport in Dendrites Revealed by Live-Cell Imaging. *J. Neurosci.* 23, 6188–6199.
- Huang, Z., Shimazu, K., Woo, N.H., Zang, K., Muller, U., Lu, B., and Reichardt, L.F. (2006). Distinct Roles of the beta1-Class Integrins at the Developing and the Mature Hippocampal Excitatory Synapse. *J. Neurosci.* 26, 11208–11219.
- Humeau, Y., and Choquet, D. (2019). The next generation of approaches to investigate the link between synaptic plasticity and learning. *Nat. Neurosci.* 22, 1536–1543.
- Jähne, S., Mikulasch, F., Heuer, H.G.H., Truckenbrodt, S., Agüi-Gonzalez, P., Grewe, K., Vogts, A., Rizzoli, S.O., and Priesemann, V. (2021). Presynaptic activity and protein turnover are correlated at the single-synapse level. *Cell Rep.* 34, 108841.
- Kabatas, S., Agüi-Gonzalez, P., Hinrichs, R., Jähne, S., Opazo, F., Diederichsen, U., Rizzoli, S.O., and Phan, N.T.N. (2019). Fluorinated nanobodies for targeted molecular imaging of biological samples using nanoscale secondary ion mass spectrometry. *J. Anal. At. Spectrom.*
- Kaech, S., and Banker, G. (2006). Culturing hippocampal neurons. *Nat. Protoc.* 1, 2406–2415.
- Kerrisk, M.E., Cingolani, L.A., and Koleske, A.J. (2014). ECM receptors in neuronal structure, synaptic plasticity, and behavior. In *Progress in Brain Research*, pp. 101–131.

Kononenko, N.L., Puchkov, D., Classen, G.A., Walter, A.M., Pechstein, A., Sawade, L., Kaempf, N., Trimbuch, T., Lorenz, D., Rosenmund, C., et al. (2014). Clathrin/AP-2 Mediate Synaptic Vesicle Reformation from Endosome-like Vacuoles but Are Not Essential for Membrane Retrieval at Central Synapses. *Neuron* 82, 981–988.

Liao, H., Huang, W., Schachner, M., Guan, Y., Guo, J., Yan, J., Qin, J., Bai, X., and Zhang, L. (2008). β 1 Integrin-mediated Effects of Tenascin-R Domains EGFL and FN6-8 on Neural Stem/Progenitor Cell Proliferation and Differentiation in Vitro. *J. Biol. Chem.* 283, 27927–27936.

Lind, D., Franken, S., Kappler, J., Jankowski, J., and Schilling, K. (2004). Characterization of the neuronal marker NeuN as a multiply phosphorylated antigen with discrete subcellular localization. *J. Neurosci. Res.* 79, 295–302.

Maidorn, M., Olichon, A., Rizzoli, S.O., and Opazo, F. (2019). Nanobodies reveal an extra-synaptic population of SNAP-25 and Syntaxin 1A in hippocampal neurons. *MABs*.

Morawski, M., Dityatev, A., Hartlage-Rübsamen, M., Blosa, M., Holzer, M., Flach, K., Pavlica, S., Dityateva, G., Grosche, J., Brückner, G., et al. (2014). Tenascin-R promotes assembly of the extracellular matrix of perineuronal nets via clustering of aggrecan. *Philos. Trans. R. Soc. Lond. B. Biol. Sci.* 369, 20140046.

Pierce, J.P., Mayer, T., and McCarthy, J.B. (2001). Evidence for a satellite secretory pathway in neuronal dendritic spines. *Curr. Biol.* 11, 351–355.

Richter, K.N., Patzelt, C., Phan, N.T.N., and Rizzoli, S.O. (2019). Antibody-driven capture of synaptic vesicle proteins on the plasma membrane enables the analysis of their interactions with other synaptic proteins. *Sci. Rep.* 9, 9231.

Rizzoli, S.O., Bethani, I., Zwilling, D., Wenzel, D., Siddiqui, T.J., Brandhorst, D., and Jahn, R. (2006). Evidence for early endosome-like fusion of recently endocytosed synaptic vesicles. *Traffic*.

Saka, S.K., Vogts, A., Kröhnert, K., Hillion, F., Rizzoli, S.O., and Wessels, J.T. (2014a). Correlated optical and isotopic nanoscopy. *Nat. Commun.*

Saka, S.K., Honigmann, A., Eggeling, C., Hell, S.W., Lang, T., and Rizzoli, S.O. (2014b). Multi-protein assemblies underlie the mesoscale organization of the plasma membrane. *Nat. Commun.* 5, 4509.

Shi, F., and Sottile, J. (2008). Caveolin-1-dependent β 1 integrin endocytosis is a critical regulator of fibronectin turnover. *J. Cell Sci.* 121, 2360–2371.

Son, Y.-J., Scranton, T.W., Sunderland, W.J., Baek, S.J., Miner, J.H., Sanes, J.R., and Carlson, S.S. (2000). The Synaptic Vesicle Protein SV2 Is Complexed with an α 5-Containing Laminin on the Nerve Terminal Surface. *J. Biol. Chem.* 275, 451–460.

Tischbirek, C.H., Wenzel, E.M., Zheng, F., Huth, T., Amato, D., Trapp, S., Denker, A., Welzel, O., Lueke, K., Svetlitchny, A., et al. (2012). Use-Dependent Inhibition of Synaptic Transmission by the Secretion of Intravesicularly Accumulated Antipsychotic Drugs. *Neuron* 74, 830–844.

Truckenbrodt, S., Viplav, A., Jähne, S., Vogts, A., Denker, A., Wildhagen, H., Fornasiero, E.F., and Rizzoli, S.O. (2018). Newly produced synaptic vesicle proteins are preferentially used in synaptic transmission. *EMBO J.* 37, 1–24.

Vreja, I.C., Kabatas, S., Saka, S.K., Kröhnert, K., Höschen, C., Opazo, F., Diederichsen, U., Rizzoli, S.O., Kröhnert, K., Höschen, C., et al. (2015). Secondary-ion mass spectrometry of genetically encoded targets. *Angew. Chemie - Int. Ed.* 54, 5784–5788.

Wang, Q., Han, T.H., Nguyen, P., Jarnik, M., and Serpe, M. (2018). Tenectin recruits integrin to stabilize bouton architecture and regulate vesicle release at the *Drosophila* neuromuscular junction. *Elife* 7.

Wienisch, M., and Klingauf, J. (2006). Vesicular proteins exocytosed and subsequently retrieved by compensatory endocytosis are nonidentical. *Nat. Neurosci.*

Woodworth, A., Fiete, D., and Baenziger, J.U. (2002). Spatial and Temporal Regulation of Tenascin-R Glycosylation in the Cerebellum. *J. Biol. Chem.* 277, 50941–50947.

Woodworth, A., Pesheva, P., Fiete, D., and Baenziger, J.U. (2004). Neuronal-specific Synthesis and Glycosylation of Tenascin-R. *J. Biol. Chem.* 279, 10413–10421.

Wratil, P.R., Horstkorte, R., and Reutter, W. (2016). Metabolic Glycoengineering with N -Acyl Side Chain Modified Mannosamines. *Angew. Chemie Int. Ed.* 55, 9482–9512.

Xiao, Z.-C., Ragsdale, D.S., Malhotra, J.D., Mattei, L.N., Braun, P.E., Schachner, M., and Isom, L.L. (1999). Tenascin-R Is a Functional Modulator of Sodium Channel β Subunits. *J. Biol. Chem.* 274, 26511–26517.

Zamze, S., Harvey, D.J., Pesheva, P., Mattu, T.S., Schachner, M., Dwek, R.A., and Wing, D.R. (1999). Glycosylation of a CNS-specific extracellular matrix glycoprotein, tenascin-R, is dominated by O-linked sialylated glycans and “brain-type” neutral N-glycans. *Glycobiology* 9, 823–831.

REVIEWER COMMENTS

Reviewer #1 (Remarks to the Author):

The authors have done a beautiful job with the revision and the manuscript has significantly improved. There are no remaining concerns.

Reviewer #2 (Remarks to the Author):

Comments concerning the revised version of „Extracellular matrix remodeling through endocytosis and resurfacing of Tenascin-R“ by Dankovich and colleagues.

The authors have substantially revised their manuscript and included impressive sets of novel data. Thereby, they have waived several, but not all of my concerns.

1) My former point 5 asked about the cellular source of Tnr in their system. They base their reply on localization of Tnr to neurons by immunocytochemistry and argue that 77% of the signal is attributable to PNN-negative and 23% to PNN-positive neurons. This, however, does not reveal the source, as Tnr can be taken up by endocytosis, as the authors argue. Thus, Tnr could be provided by a subpopulation of neurons, or astrocytes (the latter have been proposed as source of Tnr in the literature, Wanner et al, GLIA 56:1691-1709 (2008)). Hence it remains open whether we look at a paracrine or an autocrine mode of Tnr presentation to neuronal surfaces.

2) My former point 10 concerned the potential degradation of Tnr. Because the authors base many conclusions of their study on the use of one singular monoclonal antibody (clone 619) and fluorescence signals obtained therewith, they cannot exclude that they consider merely a fragment of Tnr that carries the epitope. The degradation of Tnr has not been examined, or excluded in their study so far. The authors deliberately misunderstood my suggestion, because I did not ask for Western blot analysis of biotinylated cell cultures, which would be senseless, as they rightly point out, but the analysis of biotinylated Tnr at the beginning and the end of its hypothesized journey through the cell. Alternatively, they can as well use the poly-His-tagged Tnr to perform the experiment, that is expose the culture to recombinant His-tagged Tnr and investigate the protein after 2 hours, 3 days and 6 days of incubation using Western blot and detection by antibodies to the His-tag. According to the assumed recycling pathway, one should see a slow but minor reduction of overall His-tagged Tnr with largely unchanged Mr over the observation period. I consider this

experiment necessary because the authors underline forcefully that integral Tnr is recycled over a period of days, but biochemical evidence that Tnr is not degraded to fragments is missing so far.

3) Former points 13-15 related to the antibody binding affinities and potential competition effects. Obviously, the binding pockets of divalent IgGs and the monovalent fragments derived therefrom are the same, but the determination of dissociation constants will generally reveal that divalent IgGs have a higher binding affinity (and lower Kd) to target than the individual Fab obtained from the same antibody. It is therefore unexpected that the intact antibody cannot compete off the Fab fragment. As the authors argue, it seems “an excellent binder” indeed. Because the antibody is of such importance for the study presented, credit should be given by citing the original paper that described the clone 619.

4) The authors should comment that the recombinant rTNR they use is truncated and does not comprise the amino terminus; therefore, it is probably not forming the trimers typical of native Tnr, which may change its binding properties to various interaction partners.

Reviewer #3 (Remarks to the Author):

The authors have done a beautiful job addressing all my previous questions and concerns and those of the other referees. I therefore enthusiastically endorse publication of the paper in Nat Commun.

Reviewer #4 (Remarks to the Author):

The authors have fully addressed my comments. The study is of great interest and will be an important piece of data for the field.

Replies to Reviewer Comments

Reviewer #1

The authors have done a beautiful job with the revision and the manuscript has significantly improved. There are no remaining concerns.

We thank the Reviewer for the comments.

Reviewer #2

Comments concerning the revised version of „Extracellular matrix remodeling through endocytosis and resurfacing of Tenascin-R“ by Dankovich and colleagues. The authors have substantially revised their manuscript and included impressive sets of novel data. Thereby, they have waived several, but not all of my concerns.

We thank the Reviewer for the comments.

1) My former point 5 asked about the cellular source of Tnr in their system. They base their reply on localization of Tnr to neurons by immunocytochemistry and argue that 77% of the signal is attributable to PNN-negative and 23% to PNN-positive neurons. This, however, does not reveal the source, as Tnr can be taken up by endocytosis, as the authors argue. Thus, Tnr could be provided by a subpopulation of neurons, or astrocytes (the latter have been proposed as source of Tnr in the literature, Wanner et al, GLIA 56:1691-1709 (2008)). Hence it remains open whether we look at a paracrine or an autocrine mode of Tnr presentation to neuronal surfaces.

As we mentioned in our original Reply to Reviewers under point 5, the rate of turnover (production and degradation) of TNR is limited, and therefore most of the TNR that we analyzed is not actively produced during our investigation time. It is very difficult to trace back the source of the TNR that has been produced before the start of the investigations, which necessarily leaves this question open. All we can state with certainty is that the TNR dynamics we have observed take place in both PNN-positive and PNN-negative neurons, and not in astrocytes, which contain negligible amounts of TNR in our cultures. We now state this clearly in the manuscript (page 3), and we cite the paper suggested by the Reviewer.

2) My former point 10 concerned the potential degradation of Tnr. Because the authors base many conclusions of their study on the use of one singular monoclonal antibody (clone 619) and fluorescence signals obtained therewith, they cannot exclude that they consider merely a fragment of Tnr that carries the epitope. The degradation of Tnr has not been examined, or excluded in their study so far. The authors deliberately misunderstood my suggestion, because I did not ask for Western blot analysis of biotinylated cell cultures, which would be senseless, as they rightly point out, but the analysis of biotinylated Tnr at the beginning and the end of its hypothesized journey through the cell. Alternatively, they can as well use the poly-His-tagged Tnr to perform the experiment, that is expose the culture to recombinant His-tagged Tnr and investigate the protein after 2 hours, 3 days and 6 days of incubation using Western blot and detection by antibodies to the His-tag. According to the assumed recycling pathway, one should see a slow but minor reduction of overall His-tagged Tnr with largely unchanged Mr over the

observation period. I consider this experiment necessary because the authors underline forcefully that integral Tnr is recycled over a period of days, but biochemical evidence that Tnr is not degraded to fragments is missing so far.

We politely protest against the phrasing “deliberately misunderstood”, as this was not the case. However, we are happy that the Reviewer now suggested an experiment that is entirely feasible, using the recombinant TNR. We have performed this experiment exactly as described by the Reviewer. Only a minor reduction of overall His-tagged TNR can be observed, which is well in line with the long lifetime of this protein in culture (half-life of ~7 days, Dörrbaum et al., 2018). The results are shown below, in **Figure for Reviewers 1**.

Figure for Reviewers 1. Recombinant TNR is degraded slowly in neuronal cultures. **a**, We assessed the amounts of rTNR after 2 hours, 3 days and 6 days of incubation with neuronal cultures using Western blotting. A typical example is shown here. **b**, A quantification of the rTNR band intensities (normalized), demonstrates only a minor reduction of rTNR. This is in line with the long lifetime of TNR in cultures (half-life of ~7 days: Dörrbaum et al., 2018). N = 3 independent experiments. Statistical significance was assessed using repeated measures one-way ANOVA ($F_{1,045, 2.09} = 7.565$, $p = 0.106$). Data represent the mean (lines) \pm SEM (shaded regions); dots indicate the individual experiments. The black arrowhead indicates the large non-specific band induced by the presence of albumin, which is extremely abundant in the cultures (similar to the situation in serum, which has been well described in the literature: Aroca-Aguilar et al., 2018; Bottenus et al., 2011; Chart et al., 1998). This figure is shown in the manuscript as Supplementary Fig. 4d,e.

3) Former points 13-15 related to the antibody binding affinities and potential competition effects. Obviously, the binding pockets of divalent IgGs and the monovalent fragments derived therefrom are the same, but the determination of dissociation constants will generally reveal that divalent IgGs have a higher binding affinity (and lower Kd) to target than the individual Fab obtained from the same antibody. It is therefore unexpected that the intact antibody cannot compete off the Fab fragment. As the authors argue, it seems “an excellent binder” indeed.

As we pointed out in the previous Reply to Reviewers, we did not perform any experiments combining the Fab fragment and the antibodies, and therefore this point is not relevant to our work.

We agree that the antibody should compete off the Fab fragments. However, we never tested this, since combining Fab fragments and antibodies is not a good, well-controlled experiment, in our opinion.

Because the antibody is of such importance for the study presented, credit should be given by citing the original paper that described the clone 619.

We have now added this citation to our Results and Methods sections (pages 3 and 29).

4) The authors should comment that the recombinant rTNR they use is truncated and does not comprise the amino terminus; therefore, it is probably not forming the trimers typical of native Tnr, which may change its binding properties to various interaction partners.

The recombinant TNR we use is not truncated in the fashion that the Reviewer mentions. It only lacks the signal peptide that is typical for proteins like TNR, which are targeted to the endoplasmic reticulum and are later secreted. The signal peptide, comprising the first ~30 amino acids, is expected to be removed in the mature protein, and therefore should not be included in the recombinant protein that we purchased, which is expected to be virtually identical to the secreted wild-type human protein.

The cysteine-rich N-terminal area involved in trimer formation (e.g. Nörenberg et al., 1995) is located further along, around amino acids 150 to 180 in the human isoform, and is definitely included in our recombinant protein. Therefore, this protein is expected to behave in a similar fashion to the native protein.

Reviewer #3

The authors have done a beautiful job addressing all my previous questions and concerns and those of the other referees. I therefore enthusiastically endorse publication of the paper in Nat Commun.

We thank the Reviewer for the comments.

Reviewer #4

The authors have fully addressed my comments. The study is of great interest and will be an important piece of data for the field.

We thank the Reviewer for the comments.

References

Aroca-Aguilar, J.-D., Fernández-Navarro, A., Ontañón, J., Coca-Prados, M., and Escribano, J. (2018). Identification of myocilin as a blood plasma protein and analysis of its role in leukocyte adhesion to endothelial cell monolayers. *PLoS One* *13*, e0209364.

Bottenus, D., Hossan, M.R., Ouyang, Y., Dong, W.-J., Dutta, P., and Ivory, C.F. (2011). Preconcentration and detection of the phosphorylated forms of cardiac troponin I in a cascade microchip by cationic isotachopheresis. *Lab Chip* *11*, 3793.

Chart, H., Evans, J., Chalmers, R.M., and Salmon, R.L. (1998). Escherichia coli O157 serology: false-positive ELISA results caused by human antibodies binding to bovine serum albumin. *Lett. Appl. Microbiol.* *27*, 76–78.

Dörrbaum, A.R., Kochen, L., Langer, J.D., and Schuman, E.M. (2018). Local and global influences on protein turnover in neurons and glia. *Elife* *7*.

Nörenberg, U., Hubert, M., Brümmendorf, T., Tárnok, A., and Rathjen, F.G. (1995). Characterization of functional domains of the tenascin-R (restrictin) polypeptide: cell attachment site, binding with F11, and enhancement of F11-mediated neurite outgrowth by tenascin-R. *J. Cell Biol.* *130*, 473–484.

REVIEWERS' COMMENTS

Reviewer #2 (Remarks to the Author):

The authors have performed additional experiments and clarified the issues I pointed at in my comments. All my concerns have been waived.